# Learning Representations from Heterogeneous Data for Robust Heart Rate Modeling

## Abstract

Heart rate prediction is vital for personalized health monitoring and fitness, while it frequently faces a critical challenge in real-world deployment: *data heterogeneity*. We classify it in two key dimensions: *source heterogeneity* from fragmented device markets with varying feature sets, and *user heterogeneity* reflecting distinct physiological patterns across individuals and activities. Existing methods either discard device-specific information, or fail to model user-specific differences, limiting their real-world performance. To address this, we propose a framework that learns latent representations agnostic to both heterogeneity, enabling downstream predictors to work consistently under heterogeneous data patterns. Specifically, we introduce a random feature dropout strategy to handle source heterogeneity, making the model robust to various feature sets. To manage user heterogeneity, we employ a time-aware attention module to capture long-term physiological traits and use a contrastive learning objective to build a discriminative representation space. To reflect the heterogeneous nature of real-world data, we created and publicly released a new benchmark dataset, ParroTao. Evaluations on both Parro-Tao and the public FitRec dataset show that our model significantly outperforms existing baselines by 17.5% and 10.4% in terms of MSE, respectively. Furthermore, analysis of the learned representations demonstrates their strong discriminative power, and one downstream application task confirm the practical value of our model.

## 1 Introduction

Heart rate is a key marker of cardiorespiratory health and physical performance. Accurate heart rate prediction helps individuals and clinicians monitor workload, detect risk, and plan activities with greater confidence. For fitness enthusiasts, it links training plans to physiological response by estimating how the heart will respond to a chosen pace or interval structure, which helps users target appropriate zones, adjust intensity in real time, and avoid overreaching (Nazaret et al., 2023; Sumida et al., 2013). For professional athletes, predicted heart rate profiles support evaluation of training effectiveness across sessions and inform recovery management, reducing the chance of injury (Zadeh et al., 2021; Arnold & Sade, 2017). For older adults and patients in rehabilitation, heart rate prediction can flag abnormal fluctuations in advance and guide timely intervention, which improves safety during daily activities and exercise (Ballinger et al., 2018; Dunn et al., 2021).

Early heart-rate prediction models were primarily physiology-based, employing coupled ordinary differential equations to represent cardiovascular dynamics (Stirling et al., 2008; Zakynthinaki, 2015; Mazzoleni et al., 2016; 2018; Engelen et al., 1996). While these models offer interpretability and perform well in controlled settings, they are typically developed on small, homogeneous cohorts under strict laboratory protocols. This reliance on controlled conditions limits their robustness and generalization to free-living populations (Nazaret et al., 2023). The recent proliferation of wearable devices has enabled the collection of large-scale, real-world data, paving the way for data-driven approaches. Consequently, researchers have increasingly applied machine learning methods, from statistical models (Oyeleye et al., 2022; Fang et al., 2021) to advanced architectures like recurrent and attention-based neural networks (Ni et al., 2024; Alharbi et al., 2021; Lin et al., 2023). These data-driven techniques capture complex temporal patterns, model nonlinear relationships, and integrate multiple modalities, leading to more accurate and robust predictions in real-world conditions.

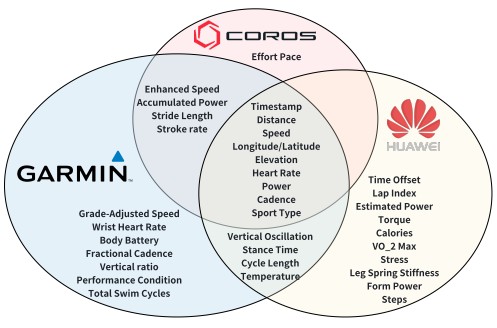 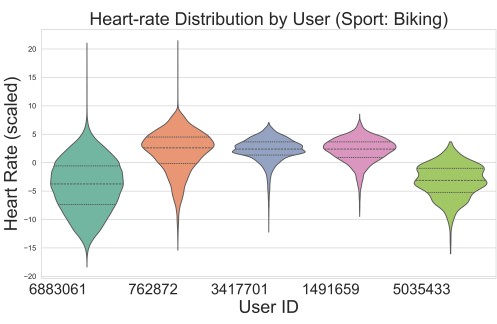

(a) Device-level differences in feature sets.

(b) Heart rate distributions during biking for five users from FitRec dataset (Ni et al., 2019).

Figure 1: Data heterogeneity in wearable data. (a) Three popular wearable devices—Garmin Fore-runner 255, Coros Pace 2, and Huawei GT 2—capture different feature sets, indicating source heterogeneity. (b) Users show distinct heart rate distributions under the same activity, highlighting user heterogeneity.

Despite this progress, *data heterogeneity* remains a significant challenge for real-world deployment. This challenge stems from the necessity for systems to process data from diverse devices and users, leading to two primary forms of heterogeneity: *source heterogeneity* and *user heterogeneity*. *Source heterogeneity* arises from a fragmented device market. Data from different devices or datasets often provide different feature sets and are sampled at different temporal resolutions, even for the same activity (Figure 1a). *User heterogeneity* reflects physiological and behavioral diversity. Even with the same device, sensed features can vary across activity types, and individuals exhibit distinct physiological profiles that lead to different heart rate distributions for the same activity (Figure 1b). Most existing methods treat such heterogeneous data as if they were homogeneous. Many are trained and evaluated on a single dataset, which limits their applicability when integrating multiple sources. When datasets are combined, a common practice is to keep only the intersection of available features and to unify the temporal resolution to the coarsest sampling rate. This discards device-specific signals and can degrade predictive accuracy. In addition, current methods seldom leverage full historical information to model differences across users and activities, which limits personalization.

To address these challenges of data heterogeneity, we propose to learn a *unified representation space* for heterogeneous heart rate time series, where a robust predictive model can be trained. For this purpose, we need to project disparate data points—originating from heterogeneous devices, users, and activities—into a common latent space. To address *source heterogeneity*, we employ a flexible encoder with a random feature dropout strategy. This technique enables the model to map data with potentially different feature sets into a shared representation space. To tackle *user heterogeneity*, we introduce two components to enhance personalization and discriminative power. First, a time-aware attention module captures user-specific physiological traits by selectively weighing historical data. Second, we design a contrastive loss to learn discriminative representations. This forces the representations of different users and activities to be distinct while ensuring that representations for the same user and activity remain similar. By combining these elements, our approach generates a powerful, personalized, and robust embedding for more accurate heart rate prediction.

Apart from the proposed method, although data heterogeneity is common in real-world scenarios, most existing public benchmarks rarely account for cross-source and cross-user variations. To bridge this gap, we have constructed and publicly released PARROTAO—a large-scale, multi-device, multi-activity dataset that reflects the heterogeneous nature of real-world data. This dataset comprises recorded exercise segments from recreational athletes, captured by devices from diverse manufacturers during a wide range of athletic activities. Unlike common practices that unify datasets by retaining only the intersection of features, we deliberately preserve the distinct, device-specific feature sets. This design, combined with the variety of activities, ensures our dataset more faithfully represents the challenges of real-world data, making it a more rigorous benchmark for evaluating a model's performance. On both the public FitRec dataset (Ni et al., 2019) and our new PARROTAO dataset, our experiments demonstrate that the proposed approach significantly outperforms exist-

ing baselines by 17.5% and 10.4% in terms of MSE, respectively, while also achieving superior performance across fine-grained sport categories. Furthermore, analyses of the learned representations confirm that our framework effectively models both source and user heterogeneity. Finally, we demonstrate the practical versatility of our framework by applying it to one downstream task.

The contributions of this paper are twofold. First, we propose a simple yet effective architecture that jointly leverages current and historical multi-channel wearable signals under missing and non-uniform feature patterns, and we systematically validate its design on two datasets. Second, we construct and will publicly release PARROTAO, a large-scale, multi-sport, multi-device dataset that preserves device-specific feature sets and cross-user variation, offering a realistic testbed for modeling heterogeneity in real-world wearable data.

The rest of the paper is organized as follows. Section 2 formulates the problem. Section 3 elaborates on our method. Section 4 presents our dataset, validates our approach, and provides an application example. Finally, Section 5 provides conclusion. Related work can be found in Appendix A. Source code can be found in supplementary material.

## 2 PROBLEM FORMULATION

Our goal is to predict a user's heart rate sequence for a given workout session based on their historical data and the characteristics of the current session. A workout session refers to a continuous period of exercise, such as a 30-minute run or a one-hour cycling trip. This task mirrors a practical scenario where, for instance, a runner plans a route with a specific elevation profile and target pace and uses the model to forecast their physiological response, helping them select a suitable training plan.

We represent each workout session as a multivariate time series. For a user $u$, their history $\mathcal{H}_u$ consists of their past $N_u$ workout sessions:

$$\mathcal{H}_u = \{(\mathbf{X}^{(i)}, \mathbf{y}^{(i)}, \mathbf{a}^{(i)}, \Delta\tau^{(i)})\}_{i=1}^{N_u}. \tag{1}$$

Here, for the $i$-th historical session: $\mathbf{X}^{(i)} = (\mathbf{x}_1^{(i)}, \ldots, \mathbf{x}_{T_i}^{(i)})$ is the sequence of multivariate features read from one sensor, where $T_i$ is the duration of the session. Each vector $\mathbf{x}_t^{(i)} \in \mathbb{R}^D$ contains sensor data at time $t$, such as pace, elevation, and cadence. We consider a global set of $D$ possible feature dimensions across all devices. $\mathbf{y}^{(i)} = (y_1^{(i)}, \ldots, y_{T_i}^{(i)})$ is the corresponding sequence of observed heart rate values. $\mathbf{a}^{(i)} \in \mathbb{R}^m$ is a vector of static attributes for the session, such as activity type (e.g., running, cycling) and user ID. $\Delta\tau^{(i)}$ is the time interval between the end of session $i-1$ and the start of session $i$, capturing the temporal gap between workouts. Due to the diversity of wearable devices, a session from a particular device may only provide a subset of the $D$ global features. For any feature dimension $d \in \{1, \ldots, D\}$ that is unobserved in a session, its value is set to zero.

The prediction task is defined as follows: given a user's history $\mathcal{H}_u$ and the features of current planned training workout, $\mathbf{X}^{(\mathrm{cur})} \in \mathbb{R}^{D \times T}$, we aim to predict the heart rate sequence $\hat{\mathbf{y}}^{(\mathrm{cur})} = (\hat{y}_1, \ldots, \hat{y}_T)$. This is achieved by learning a model $f_\theta$ with parameters $\theta$:

$$\hat{\mathbf{y}}^{(\mathrm{cur})} = f_\theta(\mathbf{X}^{(\mathrm{cur})}, \mathcal{H}_u).$$

The model parameters $\theta$ are optimized by minimizing a loss function over a training set of $N$ examples. Each sample consists of a target session and its corresponding history. The objective is to minimize the mean squared error (MSE) between the predicted and true heart rate sequences:

$$\mathcal{L}(\theta) = \frac{1}{N} \sum_{j=1}^{N} \frac{1}{T_j} \sum_{t=1}^{T_j} (\hat{y}_t^{(j)} - y_t^{(j)})^2.$$

This formulation explicitly addresses the challenge of data heterogeneity, distinguishing our approach from traditional ones that often hold the following simplified assumptions:

**(1) Source Heterogeneity:** Traditional methods typically assume a fixed and complete feature space, meaning that every input $\mathbf{X}$ has the same set of features. Our formulation relaxes this by defining a global feature set of dimension $D$, where any given session's input $\mathbf{X}^{(i)}$ may contain only a sparse subset of these features. The model $f_\theta$ must therefore learn to be robust to varied feature availability across different devices and data sources.

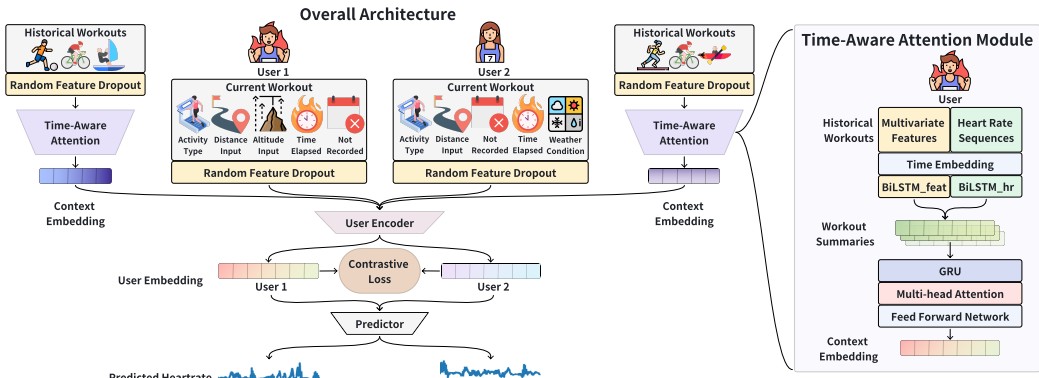

Figure 2: Architecture of the proposed framework. Random feature dropout acts on both historical and current inputs to alleviate source heterogeneity. A time-aware attention module compresses the historical record $\mathcal{H}_u$ into a context embedding $\mathbf{u}_u$, which is concatenated with the feature matrix of the current workout plan $\mathbf{X}^{(\text{cur})}$ and fed to a user encoder. Finally, a joint objective combines mean-squared error with an InfoNCE contrastive loss that aligns semantically similar embeddings.

**(2) User Heterogeneity:** Most existing methods implicitly assume that the data-generating process $P(\mathbf{y}|\mathbf{X}, u)$ remains stationary for each user, regardless of their training history. In contrast, our model conditions the predictive distribution on the user's historical data, and current activity type, i.e., $P(\mathbf{y}^{(\text{cur})}|\mathbf{X}^{(\text{cur})}, \mathcal{H}_u)$. This allows our model to capture user-specific and context-dependent variations in the relationship between sensor inputs and heart rate. As a result, the model learns a user-aware and context-sensitive mapping.

## 3 METHOD

To jointly address both *source* heterogeneity and *user* heterogeneity, we design a framework that learns a *unified representation space* for heterogeneous heart rate time series. As illustrated in Figure 2, our framework adopts a multi-stage architecture that explicitly incorporates solutions for data heterogeneity into the representation learning process, resulting in robust user embeddings for precise heart rate prediction.

The process begins by applying a *random-feature dropout* module to both the historical and current workout plan across different devices to ensure device-agnostic learning. Subsequently, the model operates in two main encoding stages. First, a *time-aware attention module* processes the user's workout history $\mathcal{H}_u$ to generate a **context embedding**, $\mathbf{u}_u$, which encapsulates long-term physiological traits. Second, this context embedding is concatenated with the features of the current workout plan, $\mathbf{X}^{(\text{cur})}$, and fed into a *user encoder* to produce a final **user embedding**, $\mathbf{z}_u$. This final embedding, which integrates historical context with current session dynamics, is the cornerstone of our approach. It is regularized using a *contrastive loss* to enforce physiological consistency and passed to a predictor to forecast the heart rate sequence.

### 3.1 RANDOM FEATURE DROPOUT

To mitigate source heterogeneity and force the model to learn from diverse feature subsets, we design a random feature dropout module that operates on the feature dimension. This is applied during training to all input features, i.e., both the current session's $\mathbf{X}^{(\text{cur})}$ and all historical $\mathbf{X}^{(i)}$.

For any given feature matrix $\mathbf{X}^{(j)} \in \mathbb{R}^{D \times T_j}$ (where $j$ can be "cur" or a historical index $i$), we generate a sample-specific binary mask $\mathbf{m}^{(j)} \in \{0, 1\}^D$. This mask determines which feature channels are kept ($m_d^{(j)} = 1$) or dropped ($m_d^{(j)} = 0$) for the entire time sequence $T_j$. The probability $p$ of dropping a feature follows a curriculum strategy (Bengio et al., 2009). It gradually increases

from a minimum value $p_{\min}$ to a maximum $p_{\max}$ over the first $E$ epochs, governed by the equation:

$$p(e) = p_{\min} + (p_{\max} - p_{\min}) \cdot \min\left(\frac{e}{E}, 1.0\right),$$

where $e$ is the current epoch number. This strategy allows the model to first learn from richer data before adapting to more challenging, sparse inputs.

To ensure model stability, we enforce two constraints. First, a predefined set of essential feature can be designated as "main features", which are always protected from being dropped. During implementation, speed and altitude are designated as the main features, given their availability across all sensors. Second, it is guaranteed that at least $K$ features are retained for every sample. This dropout strategy ensures that the model does not become reliant on any specific sensor's feature set, thereby enhancing its robustness and generalizability.

## 3.2 TIME-AWARE ATTENTION MODULE FOR CONTEXT EMBEDDING

Figure 2 depicts the proposed Time-Aware Attention Module. The user's workout history, defined in Eq. 1 , provides essential information about their baseline fitness and long-term trends. We distill this history into a fixed-size context embedding $\mathbf{u}_u$. The dynamic components ($\mathbf{X}^{(i)}$, $\mathbf{y}^{(i)}$, $\Delta\tau^{(i)}$) are processed by a hierarchical encoder, while the static attributes $\mathbf{a}^{(i)}$ (e.g., user ID, activity type) are utilized later for contrastive learning (see Section 3.3).

**1) Intra-Workout Encoding.** We first model the temporal dynamics within each historical workout. The time interval $\Delta\tau^{(i)}$ is mapped to a learned time embedding $\mathbf{e}_\tau^{(i)} \in \mathbb{R}^{D_t}$. This embedding is shared by all time steps and concatenated with both the feature and heart rate sequences. Two separate Bi-directional LSTMs (BiLSTMs) (Hochreiter & Schmidhuber, 1997) then encode these augmented sequences:

$$\mathbf{w}_{\text{feat}}^{(i)} = \text{BiLSTM}_{\text{feat}}\left(\left[\mathbf{X}^{(i)}; \mathbf{e}_\tau^{(i)}\mathbf{1}^\top\right]\right), \quad \mathbf{w}_{\text{hr}}^{(i)} = \text{BiLSTM}_{\text{hr}}\left(\left[\mathbf{y}^{(i)}; \mathbf{e}_\tau^{(i)}\mathbf{1}^\top\right]\right).$$

The final hidden states of the BiLSTMs are concatenated to form a summary vector for each workout: $\mathbf{w}^{(i)} = [\mathbf{w}_{\text{feat}}^{(i)}; \mathbf{w}_{\text{hr}}^{(i)}] \in \mathbb{R}^{2H}$.

**2) Inter-Workout Modeling.** To capture the user's physiological evolution, the sequence of workout summaries $\{\mathbf{w}^{(i)}\}_{i=1}^{N_u}$ is processed by a Gated Recurrent Unit (GRU) (Cho et al., 2014):

$$\{\mathbf{c}^{(i)}\}_{i=1}^{N_u} = \text{GRU}\left(\{\mathbf{w}^{(i)}\}_{i=1}^{N_u}\right).$$

This produces a sequence of context vectors $\mathbf{c}^{(i)} \in \mathbb{R}^H$, each conditioned on all preceding workouts.

**3) Attention and Fusion.** Finally, a multi-head attention mechanism identifies the most relevant historical sessions. The attention output $\mathbf{a}$ is computed using the context vector of the most recent historical workout $\mathbf{c}^{(N_u)}$ as the query, and the full sequence $\{\mathbf{c}^{(i)}\}_{i=1}^{N_u}$ as keys and values, and the final **context embedding** $\mathbf{u}_u \in \mathbb{R}^{H_c}$ is produced by fusing the most recent context vector with the attention output via a feed-forward network (FFN):

$$\mathbf{a} = \text{Attention}\left(\text{query} = \mathbf{c}^{(N_u)}, \text{key/value} = \{\mathbf{c}^{(i)}\}_{i=1}^{N_u}\right), \quad \mathbf{u}_u = \text{FFN}\left(\left[\mathbf{c}^{(N_u)}; \mathbf{a}\right]\right).$$

## 3.3 CONTRASTIVE REPRESENTATION LEARNING

While $\mathbf{u}_u$ captures historical patterns, the final representation must also incorporate the current workout plan. We broadcast $\mathbf{u}_u$ along the time dimension of the current workout plan's feature matrix, $\mathbf{X}^{(\text{cur})} \in \mathbb{R}^{D \times T}$, and concatenate them to form an enriched input $[\mathbf{X}^{(\text{cur})}; \mathbf{u}_u\mathbf{1}^\top] \in \mathbb{R}^{(D+H_c) \times T}$. This combined representation is fed into a **user encoder**. The output of this encoder is taken as the definitive **user embedding**, $\mathbf{z}_u \in \mathbb{R}^{H_z}$. This embedding is a holistic representation, conditioned on both long-term history and immediate task characteristics.

To ensure the user embeddings $\mathbf{z}_u$ are physiologically meaningful, we regularize the embedding space using an InfoNCE contrastive loss (Oord et al., 2018). For a mini-batch of $B$ samples, we form pairs of embeddings and their group labels $\{(\mathbf{z}_b, g_b)\}_{b=1}^B$. The group label $g_b$ is derived from

the static attributes $\mathbf{a}_b^{(\mathrm{cur})}$ of the corresponding sample (e.g., user ID or activity type). We define $\mathcal{P}$ as the set of all positive pairs of indices. The loss is:

$$\mathcal{L}_{\mathrm{CL}} \;=\; -\frac{1}{|\mathcal{P}|} \sum_{(b,c)\in\mathcal{P}} \log \frac{\exp(\mathbf{z}_b^\top \mathbf{z}_c / \tau)}{\sum_{k=1, k\neq b}^{B} \exp(\mathbf{z}_b^\top \mathbf{z}_k / \tau)}, \quad \mathcal{P} = \{(b,c) \mid g_b = g_c, \; b \neq c\}.$$

This loss, applied after $\ell_2$ normalization with temperature $\tau$, pulls embeddings from the same group together, enforcing a consistent structure on the representation space.

## 3.4 Training Objective

The final user embedding $\mathbf{z}_u$ is passed to a feed-forward predictor network to generate the heart rate forecast, $\hat{\mathbf{y}}$. The model is trained end-to-end by minimizing a combined loss function over a mini-batch of $B$ samples:

$$\mathcal{L} \;=\; \frac{1}{B} \sum_{b=1}^{B} \underbrace{\left( \frac{1}{T_b} \sum_{t=1}^{T_b} (y_t^{(b)} - \hat{y}_t^{(b)})^2 \right)}_{\mathcal{L}_{\mathrm{MSE}}^{(b)}} + \lambda\, \mathcal{L}_{\mathrm{CL}},$$

where $\mathcal{L}_{\mathrm{MSE}}^{(b)}$ is the mean squared error for the $b$-th sample of duration $T_b$, and $\mathcal{L}_{\mathrm{CL}}$ is the contrastive loss. The hyperparameter $\lambda$ balances point-wise accuracy against representation coherence.

## 4 Experiments

To comprehensively evaluate our proposed method, we design four corresponding groups of experiments: **(Group 1)** We evaluate the overall and per-sport predictive performance of our model against 8 baselines on the FitRec and PARROTAO datasets. **(Group 2)** We conduct comprehensive ablation studies on both datasets to analyze the individual impact of our three proposed modules. **(Group 3)** We use t-SNE to visualize the learned user embeddings, qualitatively assessing their ability to form distinct clusters for different users, different sports, and different activities from the same user. **(Group 4)** We demonstrate the model's utility in Route Recommendation.

### 4.1 Dataset Description

To comprehensively evaluate our model, we utilize two distinct datasets. We first use FitRec, a widely used open-source benchmark, to assess performance under large-scale and relatively uniform conditions. To further examine our model's robustness in the presence of real-world heterogeneity, we introduce PARROTAO—a device-diverse dataset collected from multiple wearable brands.

**FitRec.** The FitRec corpus, released by Ni et al. (2019), contains 167,373 workout sessions from 956 users across 40 sport categories, scraped from the public platform *Endomondo*. Each session is a multivariate time series recording core metrics alongside contextual metadata. As raw speed values can be unreliable, we follow the original study and use a *derived speed*, computed from distance and time intervals. For modeling, each session is partitioned into non-overlapping windows of 450 time steps. This results in an average of over 175 windows per user, spanning approximately two years.

**ParroTao.** To address the absence of publicly available, device-diverse corpora, we developed PARROTAO, a comprehensive dataset containing 42,576 workout sessions from 113 recreational athletes. The data were collected using three leading wearable brands (*Coros*, *Garmin*, and *Huawei*) over a three-year period from 2022 to 2025. PARROTAO is intentionally heterogeneous, with partially overlapping sensor suites and different data units across vendors, reflecting real-world conditions.

Table 1: Counts of recorded feature categories in the FitRec and PARROTAO datasets.

| Channel | FitRec | ParroTao | | |
|---|---|---|---|---|
| | | Coros | Garmin | Huawei |
| Core Spatiotemporal | 3 | 5 | 5 | 3 |
| Cardiorespiratory | 1 | 3 | 6 | 4 |
| Running dynamics | 0 | 2 | 7 | 3 |
| Sport-specific extras | 0 | 1 | 2 | 1 |
| Contextual Metadata | 3 | 5 | 5 | 5 |

Table 2: Overall performance comparison on the FitRec and PARROTAO datasets. We report the Mean ± Std for MSE and MAE estimated over 200 bootstraps. The best results are highlighted in **bold**. Lower values indicate better performance.

| Model | FitRec | | ParroTao | |
|---|---|---|---|---|
| | MSE ↓ | MAE ↓ | MSE ↓ | MAE ↓ |
| User Mean | $570.94 \pm 10.42$ | $18.06 \pm 0.10$ | $629.89 \pm 17.26$ | $19.82 \pm 0.27$ |
| Smartphone $VO_2$-Driven | $355.62 \pm 6.21$ | $14.03 \pm 0.07$ | $311.53 \pm 8.11$ | $13.08 \pm 0.16$ |
| Hybrid ODE–Neural | $346.31 \pm 5.82$ | $13.81 \pm 0.07$ | $387.24 \pm 9.18$ | $15.27 \pm 0.18$ |
| MLP | $370.26 \pm 6.00$ | $14.35 \pm 0.07$ | $284.62 \pm 7.34$ | $12.36 \pm 0.16$ |
| STM-BiLSTM-Att | $278.20 \pm 3.06$ | $12.50 \pm 0.06$ | $174.92 \pm 6.07$ | $9.70 \pm 0.15$ |
| FitRec | $248.02 \pm 3.53$ | $11.69 \pm 0.06$ | $139.53 \pm 4.86$ | $8.24 \pm 0.11$ |
| TCN | $254.93 \pm 3.37$ | $11.91 \pm 0.06$ | $160.36 \pm 5.31$ | $8.61 \pm 0.13$ |
| Transformer | $253.70 \pm 3.16$ | $11.79 \pm 0.05$ | $148.37 \pm 5.36$ | $8.24 \pm 0.13$ |
| **Ours** | $\mathbf{204.63 \pm 3.41}$ | $\mathbf{10.28 \pm 0.05}$ | $\mathbf{125.04 \pm 4.57}$ | $\mathbf{7.53 \pm 0.11}$ |

Table 1 provides a comprehensive, side-by-side comparison of the variables available in both datasets. This setup allows us to test our method's robustness against both relatively uniform data (FitRec) and highly heterogeneous data (PARROTAO). More details about the datasets can be found in Appendix C.

## 4.2 BASELINES AND IMPLEMENTATIONS

**Baselines.** We compare our method against state-of-the-art baselines spanning two main categories: physiology-based methods and data-driven methods. (1) The physiology-based approaches include the Smartphone $VO_2$-Driven model (Sumida et al., 2013) and the Hybrid ODE–Neural model (Nazaret et al., 2023). (2) The data-driven methods include FitRec (Ni et al., 2019), STM-BiLSTM-Att (Lin et al., 2023), MLP (Popescu et al., 2009), TCN (Lea et al., 2017), and Transformer (Vaswani et al., 2017). We strictly followed the original implementation of the baseline for experiment. We also report the User mean baseline, which predicts each time step with the user's training-set average. Details of the baselines can be found in Appendix D.

**Implementations.** Our model employs a two-layer LSTM backbone encoder with 128-dimensional hidden states, followed by dropout (p=0.2) and GELU activation. The time-aware attention module processes the 10 most recent workouts using a 64-dimensional BiLSTM, 128-dimensional cross-session GRU, and 4-head multi-head attention. We split the dataset into 80%, 10%, and 10% for training, validation, and testing, selecting the best model based on validation performance. We evaluate using Mean Square Error (MSE) and Mean Absolute Error (MAE) on the test set, reporting mean and standard deviation across 200 bootstrap iterations, with statistical significance assessed using one-sided Wilcoxon signed-rank tests and Friedman tests.

## 4.3 GROUP 1: PREDICTIVE PERFORMANCE

**Overall Performance:** Table 2 compares our model with the baselines on FitRec and PARROTAO. Our approach achieves the lowest MSE and MAE on both datasets, reducing MSE by 17.5% on FitRec and 10.4% on PARROTAO relative to the strongest competitors. For each baseline, Wilcoxon signed rank tests consistently yield $p < 0.001$ with absolute Cohen's $d > 6$, demonstrating that our improvements are statistically significant and substantial. Detailed statistics are reported in Appendix Table 7.

**Per-Sport Analysis:** To provide a granular assessment of our model's capabilities, we evaluated its performance on individual sport types. Our model consistently achieves top-1 performance across individual sports on both FitRec and PARROTAO datasets, obtaining 13 first-place rankings for MSE on FitRec and 6 on PARROTAO as shown in Table 4. The model achieves the lowest MSE and MAE for prevalent activities such as running and cycling. It also demonstrates remarkable robustness in data-scarce scenarios (e.g. lowest MSE of 16.28 for Elliptical training despite only 153 samples).

Table 3: Pairwise Wilcoxon signed-rank test results (win–draw–loss) of our model against baselines.

| Comparison (Ours vs.) | FitRec | | ParroTao | |
|---|---|---|---|---|
| | MSE | MAE | MSE | MAE |
| User Mean | 21-0-4 | 22-0-3 | 12-0-1 | 12-0-1 |
| Smartphone VO2 | 20-0-5 | 19-0-6 | 10-0-3 | 10-0-3 |
| Hybrid ODE–Neural | 21-0-4 | 21-0-4 | 12-0-1 | 12-0-1 |
| MLP | 20-0-5 | 21-0-4 | 9-0-4 | 9-0-4 |
| STM-BiLSTM-Att | 19-0-6 | 20-0-5 | 8-0-5 | 9-0-4 |
| FitRec | 17-0-8 | 17-0-8 | 9-0-4 | 9-0-4 |
| TCN | 18-0-7 | 19-0-6 | 9-0-4 | 7-0-6 |
| Transformer | 19-0-6 | 17-0-8 | 7-0-6 | 7-0-6 |

Table 4: Number of first-place rankings in MSE and MAE across models on the FitRec and PAR-ROTAO datasets.

| Dataset | Metric | Mean | Smart. | Hybrid. | MLP | STM. | TCN | FitRec | Trans. | Ours |
|---|---|---|---|---|---|---|---|---|---|---|
| FitRec | MSE | 2 | 0 | 0 | 1 | 1 | 2 | 3 | 3 | 13 |
| | MAE | 2 | 1 | 0 | 1 | 1 | 1 | 3 | 4 | 12 |
| PARROTAO | MSE | 0 | 1 | 0 | 1 | 0 | 0 | 2 | 5 | 4 |
| | MAE | 0 | 1 | 0 | 1 | 0 | 0 | 1 | 5 | 5 |

Table 5: Ablation results on the FitRec and PARROTAO datasets. We report the Mean ± Std for MSE and MAE estimated over 200 bootstraps. "w/o" denotes "without". The best results are highlighted in **bold**.

| Model Variant | FitRec | | ParroTao | |
|---|---|---|---|---|
| | MSE ↓ | MAE ↓ | MSE ↓ | MAE ↓ |
| Ours (Full) | **204.63 ± 3.41** | **10.28 ± 0.05** | **125.04 ± 4.57** | **7.53 ± 0.11** |
| w/o Dropout | 213.40 ± 3.15 | 10.62 ± 0.05 | 127.42 ± 5.04 | 7.75 ± 0.11 |
| w/o TAT | 209.52 ± 2.93 | 10.61 ± 0.05 | 144.34 ± 5.45 | 8.40 ± 0.13 |
| w/o Contrastive | 246.08 ± 4.33 | 11.50 ± 0.06 | 142.48 ± 5.57 | 8.21 ± 0.13 |

Detailed per-sport result can be found in Appendix Tabels 8 and 9. Statistical validation confirms our model significantly outperforms all baselines across fine-grained sport categories on both datasets, as shown in Table 3 ($p < 0.05$). Detailed results are reported in Appendix Tables 10and 11.

## 4.4 GROUP 2: ABLATION STUDIES

To validate the effectiveness of our approach, we conducted ablation studies by systematically removing key components: random feature dropout (w/o Dropout), time-aware attention (w/o TAT), and contrastive loss (w/o Contrastive). Table 5 presents the ablation results, demonstrating that each component contributes positively to model performance. The full model consistently achieves the lowest MSE and MAE on both datasets, underscoring the synergy of its architectural design. Specifically, removing contrastive learning produces the steepest decline in accuracy, with MSE rising by 20.3% on FitRec and 35.9% on PARROTAO, which confirms the effectiveness of contrastive learning in producing discriminative representations for accurate prediction. Excluding the time aware attention module also harms performance, especially on the more challenging PARROTAO dataset where the error climbs by 17.0%, thus validating its role in modelling long range temporal dependencies. Eliminating random feature dropout leads to a smaller yet consistent increase in both MSE and MAE, highlighting its contribution as a regulariser in mitigating device overfitting.

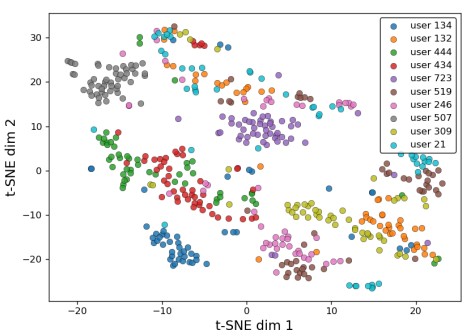 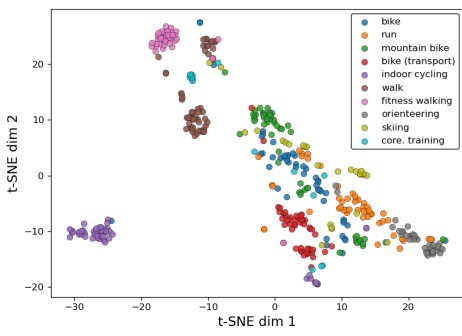

(a) t-SNE of top 10 users by workout sessions.      (b) t-SNE of the top 10 sports.

Figure 3: t-SNE visualizations of learned feature representations, colored by (a) user identity and (b) sport category. All plots show clear separation between different groups, indicating the effectiveness of the contrastive learning strategy in representation learning.

## 4.5 GROUP 3: ANALYSIS OF USER EMBEDDINGS

To evaluate the effectiveness of learned representations, we visualize user embeddings using t-SNE across different user identities and sports categories. Both Figure 3a and Figure 3b demonstrate well-separated clusters for the top 10 users and sports categories, respectively. These visualizations confirm that the learned representations are highly discriminative across both users and sports, which is essential for personalized heart rate modeling.

## 4.6 GROUP 4: DOWNSTREAM APPLICATION

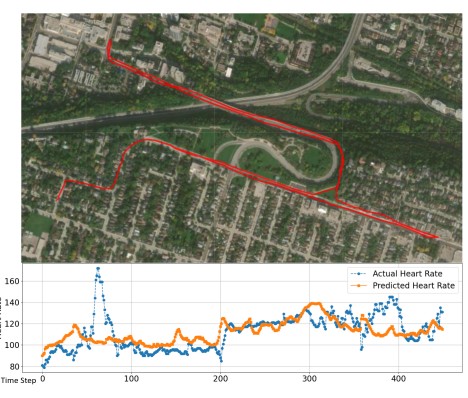 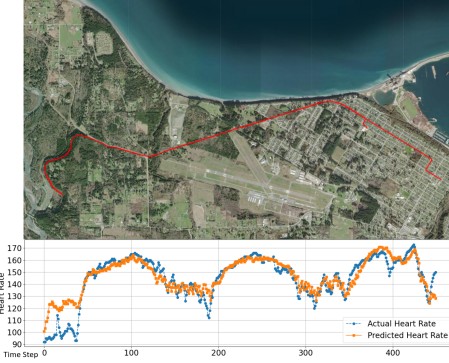

(a) Candidate route A and Corresponding HR.      (b) Candidate route B and Corresponding HR.

Figure 4: Route recommendation example. (a, b) Topographical profiles and corresponding heart rate responses for two candidate routes, A and B. The close agreement validates the model's effectiveness in forecasting physiological demands and supporting personalized route selection.

Our model's accurate heart-rate prediction capabilities unlock several valuable downstream applications. We demonstrate one major application: **Personalized Route Recommendation**. Choosing a route with the appropriate difficulty is key to effective training. Our model can serve as a planning tool by forecasting the physiological demands of different routes based on topographical data and intended pace. Athletes can input their planned routes and receive full heart rate curve predictions from our model, enabling them to compare physiological costs before starting their run.

Figure 4 shows an example of route selection in cross-country running. Two candidate routes, A and B, are considered. Route A lies on flat terrain, whereas route B climbs through mountainous ground. Our model predicts that route B induces higher average heart rate and more pronounced fluctuations compared to route A. Subsequent measurements confirm that the actual heart rate curves

closely match these predictions, demonstrating the model's effectiveness for route recommendation. Athletes can rely on the predicted heart rate to choose the route that best matches their training goals.

## 5 CONCLUSION

In this work, we propose a unified representation learning framework to address data heterogeneity in heart rate prediction, proposing multiple components to mitigate device dependencies and enhance representation quality. We introduce the PARROTAO dataset to better demonstrate real-world heterogeneity challenges. Experiments on both FitRec and PARROTAO datasets show consistent improvements across diverse devices, users, and sports scenarios, establishing a foundation for robust heart rate modeling in real-world environments.

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

CONTENTS OF APPENDIX

## A  RELATED WORK

**Heart Rate Prediction.**  Early investigations into heart rate dynamics relied on physiological knowledge to develop predictive models (Sumida et al., 2013; Lanjewar et al., 2021; Nazaret et al., 2023). For instance, researchers estimated heart rate through oxygen consumption modeling (Lanjewar et al., 2021). While these physiologically-grounded models offer strong interpretability and demonstrate robust performance in controlled environments, they were typically validated on small, homogeneous participant cohorts under stringent laboratory conditions.

As larger wearable device datasets became available, researchers transitioned toward data-driven approaches. Initial studies primarily explored classical time series models and traditional machine learning techniques, including ARIMA, k-nearest neighbors, XGBoost and so on(Oyeleye et al., 2022; Fang et al., 2021; Thoonen et al., 2022; Våge et al., 2023). However, statistical and conventional machine learning models exhibited inherent limitations when modeling complex physiological sequences such as heart rate patterns. For example, while ARIMA achieved strong accuracy within short temporal windows, it failed to capture long-range nonlinear dependencies.

Subsequent research investigated the effectiveness of deep neural networks for heart rate prediction tasks. Studies explored recurrent neural networks that leverage gating mechanisms to model intricate temporal dependencies and integrate multimodal inputs (Ni et al., 2019; Fan; Alharbi et al., 2021; Firdaus et al., 2023; Våge et al., 2023). Most recently, self-attention architectures have emerged as promising solutions. Their superior capability for capturing global contextual information has motivated researchers to explore these methods for heart rate prediction applications (Ni et al., 2024; Popovska & Georgieva-Tsaneva, 2024). Despite achieving steady performance improvements, these attention-enhanced frameworks remain trained on homogeneous data sources and do not explicitly address inter-device and inter-user heterogeneity. This limitation leaves substantial room for developing robust solutions that generalize effectively across diverse real-world deployment scenarios.

**Representation Learning for Heterogeneous Wearable Data.**  Data heterogeneity poses a significant challenge in wearable data applications. Recent methods have focused on learning unified representation spaces to integrate multi-stream and multi-modal wearable data. For example, Ali & Chong (2019) utilized semantic annotation and deep representation learning to align ontologies and facilitate the integration of heterogeneous healthcare data. Addressing the issue of unlabeled or partially labeled data, Sheng & Huber (2020) employed weakly supervised Siamese networks to learn activity- and user-aware embeddings from incomplete labels. In addition, Spathis et al. (2020) showed that large-scale, self-supervised pre-training can produce physiology-aware features that are transferable across diverse health tasks.

Privacy-preserving scenarios have prompted the adoption of federated learning as a key paradigm. For instance, Li et al. (2021) introduced Meta-HAR, which meta-trains a Reptile-style encoder and then personalizes it on-device. To address challenges such as extreme non-IID distributions and data scarcity, Zhang et al. (2023) developed FR-HMP, a method that combines client clustering with graph-based knowledge transfer.

In multimodal representation learning, research has increasingly focused on encoder fusion techniques to integrate signals into robust representations. Notable approaches include conditional attention mechanisms (Samyoun et al., 2022), stacked autoencoders (Ross et al., 2023), and transformer-based self-supervision (Wu et al., 2023). Synthesizing these advancements, a recent survey by Ye et al. (2024) systematically maps the challenges of modality, streaming, and subject heterogeneity to established machine learning paradigms, including transfer, multi-view, continual, and federated learning.

## B  LLM USAGE

LLM use was limited to grammar and phrasing refinement applied to the entire manuscript after completion of all technical content. No conceptual ideas, experimental design, data processing or result interpretation were generated by LLMs. All outputs were manually verified, and responsibility for all content rests with the authors.

## C  ADDITIONAL DETAILS ON PARROTAO DATASET

Table 6: Comparison of variables and their units across the FitRec and PARROTAO datasets. A hyphen (–) indicates that a channel is not recorded for the corresponding source dataset, while a checkmark (✓) indicates a unitless variable. The last column summarises the role of each channel in the cleaned PARROTAO dataset.

| Channel | FitRec | Coros | Garmin | Huawei | Role in PARROTAO |
|---|---|---|---|---|---|
| **Core spatiotemporal** | | | | | |
| Timestamp | Unix time | s | s | s$^{\dagger}$ | input |
| Distance | mi | m | m | m, mi | input |
| Speed | mi·h$^{-1}$, km·h$^{-1}$ | m/s | m/s | m/s, mph | input |
| Enhanced speed | – | m/s | m/s | – | input |
| Grade-adjusted speed | – | – | m/s | – | input |
| Effort pace | – | m/s | – | – | input |
| Longitude / Latitude | deg | deg | deg | deg | input |
| Altitude / Elevation | m | m | m | m, ft | input |
| **Cardiorespiratory & power** | | | | | |
| Heart rate | bpm | bpm | bpm | bpm | target |
| Wrist heart rate | – | – | bpm | – | derived-only |
| Power | – | W | W | W | input |
| Accumulated power | – | W | W | – | input |
| Estimated power | – | – | – | W | input |
| Calories | – | – | – | kcal | derived-only |
| Body battery | – | – | ✓ | – | derived-only |
| Respiration rate | – | – | brpm | – | derived-only |
| **Running / motion dynamics** | | | | | |
| Cadence | – | spm | spm | spm | input |
| Stride length | – | mm | mm | – | input |
| Stance time | – | – | ms | ms | input |
| Fractional cadence | – | – | ✓ | – | input |
| Vertical ratio | – | – | % | – | input |
| Cycle length | – | – | m | m | input |
| Performance condition | – | – | ✓ | – | derived-only |
| **Sport-specific extras** | | | | | |
| Stroke / Stroke rate | – | spm | spm | – | input |
| Temperature | – | – | °C | °C, F | input |
| **Contextual metadata** | | | | | |
| User ID | ID | ID | ID | ID | metadata |
| Sport category | category | category | category | category | metadata |
| Gender | binary | binary | binary | binary | metadata |
| Age | – | category | category | category | metadata |
| Fitness Level | – | category | category | category | metadata |

**Units:** $m$: metre, $mi$: mile, $km$: kilometre, $h$: hour, $s$: second, $deg$: degree, $ft$: foot, $bpm$: beats per minute, $W$: watt, $N{\cdot}m$: newton-metre, $kcal$: kilocalorie, $kJ$: kilojoule, $mL$: millilitre, $kg$: kilogram, $min$: minute, $spm$: steps/strokes per minute, $mm$: millimetre, $ms$: millisecond, $kN/m$: kilonewton per metre, $brpm$: breaths per minute.
$^{\dagger}$ Huawei samples timestamps every $5\,s$. Coros and Garmin sample at $1\,s$.

## D  ADDITIONAL DETAILS ON BASELINES

We compare our method against the following baselines, covering naïve personalization, physiologically grounded models, and modern deep learning architectures.

**User mean.** For each workout from a user $u$, the target variable at every time step is predicted as the mean value of that variable across all of $u$'s sequences in the training set. This serves as a simple yet important personalized baseline.

**Multilayer Perceptron (MLP) (Popescu et al., 2009).** A standard Multilayer Perceptron with two hidden layers. It processes a flattened concatenation of kinematic features, user attribute embeddings, and contextual inputs (e.g., features from a previous workout) to produce a prediction for each time step.

**Smartphone VO$_2$-Driven model (Sumida et al., 2013).** This physiology-grounded model first computes an oxygen demand proxy from resting metabolism, speed, and gradient. This proxy is personalized using a user-specific scaling factor and resting heart rate normalization. Finally, the proxy and other kinematic features are fed into a feed-forward network to predict heart rate.

**Hybrid ODE–neural model (Nazaret et al., 2023).** This model uses a system of ordinary differential equations (ODEs) to represent heart rate dynamics. Personalization is achieved by making the ODE parameters learned functions of a user embedding. An encoder generates this embedding from a user's workout history, and a decoder solves the resulting personalized ODEs—which account for workout intensity, environment, and fatigue—to forecast the heart rate trajectory.

**FitRec (Ni et al., 2019).** A sequential recommendation-style model that uses user embeddings and recurrent encoders to jointly model user state and session dynamics. We adapt it to map input sequences and user IDs to per-timestep heart-rate predictions.

**STM-BiLSTM-Att (Lin et al., 2023).** An attention-augmented recurrent architecture. It utilizes an LSTM layer for long-range temporal feature extraction, a BiLSTM layer to capture both forward and backward dependencies, and a final attention mechanism to re-weight the BiLSTM hidden states before making a prediction. We train it end-to-end on windowed inputs as per the original design.

**Temporal Convolutional Network (TCN) (Lea et al., 2017).** A Temporal Convolutional Network that applies dilated 1D convolutions over the sequence of concatenated features. Its residual blocks are designed to capture temporal patterns across a large receptive field to make per-timestep predictions.

**Transformer (Vaswani et al., 2017).** A multi-layer Transformer encoder that processes the full input sequence. It uses self-attention mechanisms to model complex, long-range dependencies between all time steps, followed by a position-wise head to generate the output sequence.

## E    ADDITIONAL DETAILS ON IMPLEMENTATION

### E.1    MODEL STRUCTURE

The backbone encoder for the current workout is a two-layer LSTM with a 128-dimensional hidden state, followed by a dropout layer (ratio 0.2) and GELU activation. To encode historical data, we use a time-aware attention module that processes the 10 most recent workouts of a user. This module comprises a time encoding component, a 64-dimension BiLSTM, a 128-dimension cross-session GRU, and a 4-head multi-head attention mechanism (hidden dimension 128). User ID, sport category, and gender are represented as learned embeddings. The final input to the backbone encoder is a concatenation of the current workout's features, the output from the historical encoder, and all relevant context embeddings.

### E.2    TRAINING DETAILS

We randomly partition the data into 80% for training, 10% for validation, and 10% for testing. In all experiments, this 80/10/10 split is performed at the participant level: all sessions from a given user are assigned to exactly one split, so no user appears in more than one set. Because users contribute different numbers of workouts, we construct the splits so that the total number of workouts in each split is approximately in an 8:1:1 ratio for train, validation, and test. The model is trained using the RMSProp optimizer with a batch size of 64 and a learning rate of 0.01. We incorporate a contrastive loss with a weight of 0.1 and a cosine similarity temperature of 0.1. To ensure stability, we apply gradient clipping at a threshold of 2.0. We employ early stopping with a patience of 10 epochs, selecting the best model based on its validation set performance for final evaluation. We note that hyperparameters were not extensively tuned.

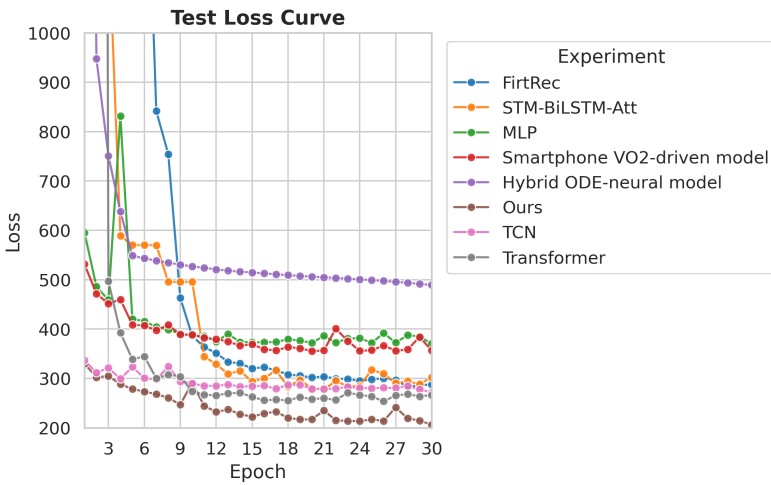

Figure 5: Models' Convergence Rate on the FitRec Dataset

### E.3 Introduction on BiLSTM

Bidirectional Long Short-Term Memory (BiLSTM) is an extension of the standard LSTM (Siami-Namini et al., 2019). It consists of two independent LSTM layers that process the sequence in opposite directions. The forward LSTM captures dependencies from past to future, while the backward LSTM captures dependencies from future to past. Their outputs are combined, either by concatenation or weighted fusion, to form the final representation. This structure enables BiLSTM to exploit both past and future context, leading to more comprehensive sequence modeling.

## F Additional Experiments

### F.1 Convergence Rate

Figure 5 demonstrates that our model converges markedly faster than all baselines. Specifically, the test loss of *Ours* drops below 250 within the first ten epochs and stabilizes around 200 by epoch 15, whereas the best competing methods STM-BiLSTM-Att, TCN and the Transformer require more than 20 epochs to reach comparable loss levels and never fall below 280 in the observed range. Other baselines such as FirtRec and the Hybrid ODE-Neural model exhibit even slower convergence. The consistently lower trajectory of *Ours* model underscores the efficiency and robustness of the proposed optimization scheme.

### F.2 Additional Results of Performance Experiment

#### F.2.1 Methodology Details

We evaluate performance using Mean Squared Error (MSE) and Mean Absolute Error (MAE) on the held-out test set. To estimate uncertainty, we report the mean and standard deviation for each metric across 200 bootstrap iterations, with each iteration sampling 80% of the test set with replacement.

For statistical evaluation, we use the one-sided Wilcoxon signed-rank test to compare our model with each baseline, applying the Benjamini–Hochberg FDR correction ($q = 0.05$) for multiple comparisons. For per-sport analysis, we use the Friedman test to compare model ranks across sports, followed by pairwise Wilcoxon tests with FDR correction. We also report Cohen's d to quantify effect sizes.

#### F.2.2 Results

Table 7: Statistical analysis of our model against baselines using a one-sided Wilcoxon signed-rank test. We report the relative improvement (Imp. %) and Cohen's d effect size. For all comparisons, the FDR-corrected p-value ($p_{adj}$) is less than 0.001, indicating statistically significant improvements.

| Baseline Compared | MSE | | MAE | |
|---|---|---|---|---|
| | Imp. (%) ↑ | Cohen's d | Imp. (%) ↑ | Cohen's d |
| **FitRec Dataset** | | | | |
| User Mean | 64.16 | -35.06 | 43.05 | -81.26 |
| MLP | 44.73 | -28.11 | 28.35 | -65.01 |
| Smartphone VO$_2$-Driven | 42.46 | -24.81 | 26.70 | -60.82 |
| Hybrid ODE–Neural | 40.91 | -24.84 | 25.50 | -56.15 |
| STM-BiLSTM-Att | 26.44 | -28.87 | 17.70 | -48.57 |
| FitRec | 17.49 | -12.53 | 12.00 | -32.18 |
| TCN | 19.73 | -14.38 | 13.66 | -35.31 |
| Transformer | 19.34 | -15.72 | 12.79 | -36.00 |
| **ParroTao Dataset** | | | | |
| User Mean | 80.15 | -30.62 | 62.04 | -48.38 |
| MLP | 56.07 | -25.28 | 39.09 | -35.94 |
| Smartphone VO$_2$-Driven | 59.86 | -26.75 | 42.49 | -40.09 |
| Hybrid ODE–Neural | 67.71 | -30.12 | 50.73 | -43.30 |
| STM-BiLSTM-Att | 28.52 | -10.48 | 22.43 | -17.98 |
| FitRec | 10.38 | -5.21 | 8.65 | -11.05 |
| TCN | 22.03 | -10.73 | 12.64 | -13.79 |
| Transformer | 15.73 | -5.81 | 8.70 | -7.99 |

Table 8: Per-sport performance comparison on the FitRec dataset. We report MSE and MAE for each model. $N_{samp}$ denotes the total number of data samples for each sport. Best result in each row is in **bold**. Model names are abbreviated: Mean (User Mean), Smart. (Smartphone VO2-Driven), Hybrid (Hybrid ODE-Neural), STM (STM-BiLSTM-Att), FitRec (FitRec), Trans. (Transformer).

| Sport | $N_{samp}$ | Metric | Mean | Smart. | Hybrid | MLP | STM | TCN | FitRec | Trans. | Ours |
|---|---|---|---|---|---|---|---|---|---|---|---|
| Badminton | 139 | MSE | 268.00 | 359.02 | 248.93 | 259.64 | 237.65 | 219.21 | **214.38** | 476.84 | 276.78 |
| | | MAE | 13.29 | 13.98 | 13.11 | 13.12 | 12.33 | 12.11 | **11.82** | 18.95 | 13.98 |
| Biking | 1.2M | MSE | 586.01 | 354.78 | 369.68 | 385.40 | 300.73 | 274.38 | 262.63 | 290.81 | **225.83** |
| | | MAE | 18.46 | 14.64 | 14.79 | 15.31 | 13.17 | 12.64 | 12.26 | 13.06 | **11.09** |
| Biking (transport) | 81k | MSE | 1500.35 | 973.78 | 903.22 | 948.31 | 233.28 | 209.18 | 188.23 | 221.59 | **156.19** |
| | | MAE | 25.63 | 18.37 | 17.66 | 18.61 | 11.46 | 10.59 | 10.09 | 11.18 | **8.93** |
| Circuit Training | 813 | MSE | 1233.44 | 1485.00 | 1217.04 | 1795.67 | 1031.26 | 1521.38 | **772.04** | 1654.04 | 1073.43 |
| | | MAE | 28.74 | 33.49 | 28.71 | 37.80 | 26.64 | 34.83 | **24.00** | 35.92 | 26.53 |
| Core Stability | 9.9k | MSE | 707.01 | 612.02 | 631.17 | 700.65 | 816.02 | 599.90 | **536.63** | 737.89 | 733.84 |
| | | MAE | 21.56 | 20.30 | 19.60 | 21.80 | 23.03 | 19.66 | **18.31** | 20.89 | 21.14 |
| Country Skiing | 4.5k | MSE | 408.63 | 256.83 | 265.05 | 273.10 | 265.95 | 219.98 | 187.39 | 197.80 | **153.19** |
| | | MAE | 16.36 | 12.31 | 12.64 | 13.02 | 13.19 | 11.70 | 10.83 | 11.16 | **9.50** |
| Downhill Skiing | 1.3k | MSE | 2137.17 | 809.17 | 1407.15 | **540.92** | 734.23 | 945.31 | 825.25 | 1091.66 | 974.62 |
| | | MAE | 40.79 | 24.74 | 32.36 | **19.74** | 23.17 | 23.64 | 21.56 | 25.82 | 23.87 |
| Elliptical | 153 | MSE | 2558.73 | 1190.26 | 975.20 | 812.55 | 1883.87 | 2360.76 | 1008.95 | 1629.16 | **16.28** |
| | | MAE | 50.47 | 34.35 | 30.80 | 28.30 | 42.58 | 48.12 | 30.48 | 37.54 | **3.20** |
| Fitness Walking | 5.8k | MSE | 1375.57 | 428.07 | 401.28 | 491.48 | 428.84 | 399.07 | 412.04 | 327.94 | **309.01** |
| | | MAE | 30.54 | 15.59 | 15.46 | 17.06 | 15.57 | 15.13 | 14.88 | **13.30** | 13.38 |
| Golf | 342 | MSE | 2118.83 | 1328.68 | 1306.66 | 701.39 | 759.43 | 843.03 | 1045.83 | 1015.27 | **375.84** |
| | | MAE | 40.19 | 29.47 | 35.08 | 25.25 | 25.55 | 28.18 | 29.62 | 30.82 | **17.76** |
| Hiking | 4.6k | MSE | 1147.07 | 434.73 | 584.55 | 435.50 | 452.86 | 451.39 | 474.04 | **400.21** | 515.55 |
| | | MAE | 29.98 | 16.55 | 19.44 | 16.49 | 16.82 | 17.05 | 17.17 | **16.04** | 17.29 |
| Horseback Riding | 301 | MSE | **290.71** | 741.39 | 1714.80 | 813.90 | 738.86 | 724.74 | 415.21 | 1209.31 | 475.45 |
| | | MAE | **14.18** | 25.41 | 39.60 | 26.54 | 25.22 | 25.19 | 18.32 | 32.51 | 19.69 |

*Continued on next page*

| Sport | $N_{samp}$ | Metric | Mean | Smart. | Hybrid | MLP | STM | TCN | FitRec | Trans. | Ours |
|---|---|---|---|---|---|---|---|---|---|---|---|
| Indoor Cycling | 28k | MSE | 883.41 | 563.06 | 545.07 | 534.79 | **409.94** | 566.33 | 738.03 | 448.94 | 637.04 |
|  |  | MAE | 24.60 | 19.86 | 19.32 | 19.73 | **16.55** | 19.15 | 21.66 | 17.31 | 20.30 |
| Kayaking | 1.5k | MSE | 1668.17 | 1343.88 | 847.70 | 1391.05 | 769.91 | **532.75** | 895.31 | 683.94 | 712.02 |
|  |  | MAE | 34.56 | 32.12 | 24.09 | 32.76 | 23.01 | **17.91** | 24.69 | 21.65 | 22.88 |
| Mountain Biking | 149k | MSE | 549.66 | 377.16 | 396.45 | 420.48 | 330.65 | 306.38 | 294.83 | 336.60 | **238.88** |
|  |  | MAE | 18.67 | 15.18 | 15.64 | 16.23 | 13.95 | 13.51 | 13.16 | 14.30 | **11.79** |
| Orienteering | 12k | MSE | 503.81 | 382.87 | 332.85 | 375.91 | 290.54 | 243.63 | 247.38 | 250.09 | **203.67** |
|  |  | MAE | 17.78 | 14.75 | 13.18 | 14.49 | 12.38 | 11.29 | 11.70 | 11.70 | **10.38** |
| Roller Skiing | 6.9k | MSE | 570.16 | 376.52 | 326.24 | 448.84 | 483.11 | 334.41 | 327.54 | 306.27 | **172.87** |
|  |  | MAE | 19.08 | 15.62 | 14.47 | 16.73 | 17.40 | 14.34 | 14.24 | 14.14 | **10.57** |
| Rowing | 1k | MSE | 1273.99 | 863.27 | 618.96 | 765.00 | 580.90 | **472.35** | 542.79 | 562.52 | 495.87 |
|  |  | MAE | 28.02 | 24.41 | 20.84 | 23.68 | 19.19 | 18.42 | 19.39 | 19.12 | **16.57** |
| Running | 1.2M | MSE | 461.52 | 300.54 | 266.86 | 298.47 | 237.71 | 216.21 | 212.30 | 194.52 | **162.84** |
|  |  | MAE | 16.47 | 12.69 | 12.04 | 12.58 | 11.36 | 10.71 | 10.62 | 9.92 | **8.92** |
| Skating | 3.7k | MSE | 788.48 | 401.55 | 455.27 | 448.02 | 330.47 | 338.99 | 332.12 | 391.13 | **271.61** |
|  |  | MAE | 22.25 | 16.27 | 16.95 | 16.85 | 15.11 | 14.49 | 14.69 | 16.30 | **13.03** |
| Tennis | 150 | MSE | 111.54 | 115.92 | 1342.97 | 352.04 | 574.16 | 255.07 | 556.56 | 152.48 | **93.90** |
|  |  | MAE | 8.80 | **7.93** | 34.41 | 16.87 | 21.89 | 14.26 | 21.31 | 10.35 | 8.05 |
| Treadmill Running | 155 | MSE | 689.49 | 730.61 | 1124.49 | 683.95 | 2605.79 | 1064.97 | 534.19 | **382.90** | 682.22 |
|  |  | MAE | 25.92 | 26.71 | 32.93 | 25.82 | 50.66 | 32.08 | 22.69 | **18.78** | 25.70 |
| Treadmill Walking | 161 | MSE | **2378.23** | 4392.97 | 4787.53 | 7168.30 | 4153.26 | 6565.12 | 5468.72 | 3547.43 | 5579.53 |
|  |  | MAE | **46.52** | 64.55 | 67.44 | 83.31 | 62.12 | 79.31 | 72.02 | 57.11 | 72.74 |
| Walking | 13k | MSE | 1937.33 | 281.32 | 286.38 | 267.51 | 335.30 | 243.57 | 228.76 | **209.03** | 221.26 |
|  |  | MAE | 38.71 | 13.01 | 12.93 | 12.66 | 14.83 | 12.42 | 11.73 | **11.19** | 11.50 |
| Weight Training | 655 | MSE | 515.46 | 599.99 | 725.32 | 647.03 | 327.59 | 557.73 | 633.32 | 755.56 | **242.69** |
|  |  | MAE | 18.21 | 20.91 | 23.35 | 21.87 | 15.03 | 19.53 | 21.43 | 23.85 | **12.54** |

Table 9: Per-Sport performance comparison on the PARROTAO dataset. Best result in each row is in **bold**. Model names are abbreviated as in the previous table.

| Sport | $N_{samp}$ | Metric | Mean | Smart. | Hybrid | MLP | STM | TCN | FitRec | Trans. | Ours |
|---|---|---|---|---|---|---|---|---|---|---|---|
| Running | 451k | MSE | 544.14 | 282.81 | 353.92 | 260.01 | 151.42 | 139.69 | 117.97 | 129.84 | **103.00** |
|  |  | MAE | 18.59 | 12.49 | 14.72 | 11.81 | 9.13 | 8.07 | 7.71 | 7.76 | **6.93** |
| Cross-Country Run | 14k | MSE | 849.64 | 550.25 | 545.32 | 465.64 | 343.64 | 314.47 | 226.37 | 243.62 | **221.82** |
|  |  | MAE | 24.24 | 19.01 | 18.41 | 17.23 | 14.10 | 13.38 | 11.22 | 11.89 | **11.21** |
| Cycling | 18k | MSE | 1625.61 | 641.13 | 815.63 | 566.70 | 434.18 | 355.16 | **323.12** | 277.79 | 330.91 |
|  |  | MAE | 32.24 | 19.20 | 21.60 | 17.82 | 15.29 | 13.41 | 12.34 | **12.10** | 12.67 |
| Open Water Swimming | 434 | MSE | 2746.05 | 108.26 | 683.41 | 62.72 | 68.80 | 90.88 | 61.50 | **48.47** | 64.33 |
|  |  | MAE | 51.85 | 8.22 | 24.90 | 6.42 | 6.94 | 8.30 | 6.35 | **5.77** | 6.93 |
| Treadmill Running | 3.8k | MSE | 750.02 | 373.93 | 480.54 | 311.19 | 198.98 | 187.37 | 253.45 | **155.91** | 303.32 |
|  |  | MAE | 21.61 | 15.14 | 16.06 | 13.66 | 10.39 | 9.29 | 11.35 | **7.78** | 11.05 |
| Indoor Aerobics | 2.1k | MSE | 1739.97 | 823.53 | 812.75 | 677.60 | 667.35 | 805.86 | 928.91 | 1105.09 | **891.14** |
|  |  | MAE | 37.20 | 20.05 | 21.62 | 18.77 | 18.13 | 21.31 | 22.09 | 22.98 | **22.50** |
| Customized Exercise | 627 | MSE | 651.60 | **359.16** | 566.87 | 637.42 | 417.85 | 494.27 | 504.43 | 830.64 | 419.78 |
|  |  | MAE | 19.12 | **15.78** | 20.76 | 20.26 | 17.16 | 16.72 | 18.02 | 25.32 | 17.01 |
| Rope Skipping | 3.1k | MSE | 2446.99 | 249.07 | 546.74 | 267.67 | 224.92 | 156.95 | 153.49 | **128.03** | 138.04 |
|  |  | MAE | 45.91 | 12.73 | 20.27 | 13.38 | 12.69 | 9.21 | 9.43 | **8.49** | 9.07 |
| Walking | 307 | MSE | 1633.25 | 703.04 | 718.77 | 595.47 | 557.26 | 536.95 | 1248.97 | 684.72 | **394.75** |
|  |  | MAE | 32.43 | 22.42 | 22.28 | 19.37 | 19.59 | 19.48 | 29.86 | 22.09 | **15.91** |
| Rowing Machine | 1.4k | MSE | 2461.51 | 438.82 | 677.75 | 444.05 | 430.41 | 422.15 | **402.67** | 345.11 | 486.08 |
|  |  | MAE | 44.81 | 16.10 | 20.51 | 15.97 | 15.55 | 15.39 | **14.94** | 14.03 | 17.00 |
| Strength Training | 6.1k | MSE | 1256.10 | 680.15 | 754.51 | 613.33 | 397.77 | 373.42 | 394.35 | **296.10** | 362.11 |
|  |  | MAE | 30.76 | 20.10 | 20.61 | 19.21 | 16.11 | 13.92 | 14.49 | **11.98** | 14.18 |
| Hiking | 796 | MSE | 82.29 | 435.73 | 1075.88 | **71.25** | 78.79 | 169.65 | 356.85 | 1003.80 | 242.15 |
|  |  | MAE | 7.39 | 19.17 | 31.63 | **6.96** | 7.30 | 10.14 | 14.08 | 20.51 | 11.76 |
| Pool Swimming | 2.5k | MSE | 1747.09 | 381.96 | 419.96 | 491.55 | 430.31 | 307.96 | 360.84 | **291.79** | 289.32 |
|  |  | MAE | 38.39 | 14.87 | 16.00 | 18.25 | 16.57 | 13.62 | 14.24 | **13.55** | 12.70 |

Table 10: Friedman test average ranks for each model on both datasets (lower is better). The p-values confirm a significant difference among models.

| Model | FitRec | | ParroTao | |
|---|---|---|---|---|
| | MSE Rank | MAE Rank | MSE Rank | MAE Rank |
| User Mean | 7.28 | 7.36 | 8.46 | 8.31 |
| Smartphone VO2-Driven | 5.76 | 5.76 | 6.00 | 5.92 |
| Hybrid ODE–Neural | 6.08 | 6.16 | 7.31 | 7.46 |
| MLP | 6.00 | 6.16 | 5.23 | 5.00 |
| STM-BiLSTM-Att | 4.96 | 5.04 | 4.00 | 4.23 |
| TCN | 4.12 | 4.00 | 3.54 | 3.46 |
| FitRec | 3.68 | 3.56 | 3.92 | 4.00 |
| Transformer | 4.32 | 4.20 | 3.38 | 3.38 |
| **Ours** | **2.80** | **2.76** | **3.15** | **3.23** |
| **Friedman p-value** | 1.32e-08 | 3.55e-09 | 6.87e-08 | 1.10e-07 |

Table 11: Pairwise Wilcoxon signed-rank test results of our model against baselines. We report the FDR-Corrected p-value and the win-draw-loss (w-d-l) statistics.

| Dataset | Comparison (Ours vs.) | MSE | | MAE | |
|---|---|---|---|---|---|
| | | p-value | w-d-l | p-value | w-d-l |
| FitRec | FitRec | 2.20e-01 | 17-0-8 | 1.01e-01 | 17-0-8 |
| | User Mean | 3.29e-04 | 21-0-4 | 1.88e-04 | 22-0-3 |
| | MLP | 2.87e-04 | 20-0-5 | 3.81e-05 | 21-0-4 |
| | TCN | 9.64e-03 | 18-0-7 | 1.03e-03 | 19-0-6 |
| | Transformer | 2.55e-02 | 19-0-6 | 1.73e-02 | 17-0-8 |
| | STM-BiLSTM-Att | 8.07e-03 | 19-0-6 | 2.03e-03 | 20-0-5 |
| | Hybrid ODE–Neural | 1.62e-04 | 21-0-4 | 5.39e-05 | 21-0-4 |
| | Smartphone VO2-Driven | 3.42e-03 | 20-0-5 | 2.50e-04 | 19-0-6 |
| PARROTAO | FitRec | 1.10e-01 | 9-0-4 | 2.44e-01 | 9-0-4 |
| | Mean | 4.88e-04 | 12-0-1 | 7.32e-04 | 12-0-1 |
| | MLP | 6.81e-02 | 9-0-4 | 4.79e-02 | 9-0-4 |
| | TCN | 5.88e-01 | 9-0-4 | 8.39e-01 | 7-0-6 |
| | Transformer | 4.97e-01 | 7-0-6 | 6.85e-01 | 7-0-6 |
| | STM-BiLSTM-Att | 6.85e-01 | 8-0-5 | 3.76e-01 | 9-0-4 |
| | Hybrid ODE–Neural | 4.88e-04 | 12-0-1 | 4.88e-04 | 12-0-1 |
| | Smartphone VO2-Driven | 8.06e-03 | 10-0-3 | 6.10e-03 | 10-0-3 |

## G  DESIGN CHOICES IN THE TIME-AWARE ATTENTION MODULE

To assess the impact of our architectural choices in the time-aware attention module, we conducted additional experiments in which (i) we kept the GRU fixed and replaced the BiLSTM with a GRU, a unidirectional LSTM, or a Transformer encoder, and (ii) we kept the BiLSTM fixed and replaced the GRU with a BiLSTM, a unidirectional LSTM, or a Transformer encoder. We evaluated all variants on the FitRec and PARROTAO datasets, and report the results in Table 12. Across both datasets, these substitutions lead to only small fluctuations in MSE and MAE, suggesting that the performance gains primarily come from the overall design of the time-aware attention module rather than from any particular choice of recurrent encoder.

## H  IMPACT OF IMPUTATION STRATEGY ON MODEL PERFORMANCE

We conducted additional experiments in which unobserved features were imputed using the feature-wise mean, median, or majority value estimated from the training, validation, and test sets, and we

Table 12: Encoder choice in the time-aware attention module. We report mean $\pm$ std of MSE and MAE on the FitRec and PARROTAO datasets when replacing either the first (default: BiLSTM) or second (default: GRU) encoder slot with alternative architectures.

| Dataset | Modification | MSE $\downarrow$ | MAE $\downarrow$ |
|---|---|---|---|
| **FitRec** | Original | **204.63** $\pm$ **3.41** | $10.28 \pm 0.05$ |
| | BiLSTM $\rightarrow$ GRU | $248.84 \pm 3.16$ | $11.78 \pm 0.06$ |
| | BiLSTM $\rightarrow$ LSTM | $206.20 \pm 4.09$ | **10.27** $\pm$ **0.06** |
| | BiLSTM $\rightarrow$ Transformer | $206.30 \pm 3.48$ | $10.30 \pm 0.06$ |
| | GRU $\rightarrow$ BiLSTM | $212.42 \pm 3.49$ | $10.56 \pm 0.06$ |
| | GRU $\rightarrow$ LSTM | $210.42 \pm 2.88$ | $10.63 \pm 0.06$ |
| | GRU $\rightarrow$ Transformer | $208.89 \pm 3.50$ | $10.34 \pm 0.06$ |
| **ParroTao** | Original | $125.04 \pm 4.57$ | **7.53** $\pm$ **0.11** |
| | BiLSTM $\rightarrow$ GRU | **124.01** $\pm$ **4.49** | $7.63 \pm 0.11$ |
| | BiLSTM $\rightarrow$ LSTM | $128.53 \pm 4.72$ | $7.80 \pm 0.12$ |
| | BiLSTM $\rightarrow$ Transformer | $127.06 \pm 4.84$ | $7.77 \pm 0.11$ |
| | GRU $\rightarrow$ BiLSTM | $125.57 \pm 4.79$ | $7.65 \pm 0.11$ |
| | GRU $\rightarrow$ LSTM | $125.29 \pm 4.86$ | $7.54 \pm 0.12$ |
| | GRU $\rightarrow$ Transformer | $127.71 \pm 4.80$ | $7.79 \pm 0.11$ |

Table 13: Heart-rate forecasting performance (MSE and MAE) on the PARROTAO dataset under different imputation strategies.

| Imputation method | MSE $\downarrow$ | MAE $\downarrow$ |
|---|---|---|
| Avg | $429.95 \pm 10.17$ | $16.34 \pm 0.18$ |
| Majority | $400.15 \pm 10.83$ | $15.48 \pm 0.20$ |
| Median | $389.15 \pm 10.78$ | $15.15 \pm 0.19$ |
| Zero-Imputation | **125.04** $\pm$ **4.57** | **7.53** $\pm$ **0.11** |

compared these variants with our original zero-imputation scheme. The result is shown in Table 13. Zero-Imputation consistently yields better predictive performance than these three alternatives. A key reason is that some features are only observed for a relatively small subset of devices or activities. Filling the missing entries of such sparse features with their mean, median, or majority value induces a noticeable distribution shift between genuinely observed and imputed values.

## I  EFFECT OF USER AND ACTIVITY LABELS IN CONTRASTIVE LEARNING

We have compared three variants: using only user IDs as labels, using only activity types as labels, and using both user IDs and activity types jointly (our original implementation). The experimental results are presented in Table 14. The joint-label setting consistently achieves the best performance on both the FitRec and PARROTAO dataset. Intuitively, user-ID supervision encourages the encoder to capture user-specific preferences and physiological patterns across sessions, while activity-type supervision promotes better separation between different sports. Combining both sources of supervision therefore more accurate and robust downstream predictions.

## J  DEMOGRAPHIC CHARACTERISTICS OF THE PARROTAO COHORT

The demographic and fitness characteristics of the PARROTAO cohort are detailed in Table 15. The dataset comprises 113 athletes, with comprehensive metadata on their age, gender, and fitness level.

## K  EFFECT OF HISTORY LENGTH ON TIME-AWARE MODELING

We evaluate history lengths $K \in \{1, 5, 10, 20, 30\}$ on the FITREC dataset and report overall MSE and MAE in Table 16. Moving from $K=1$ to $K=5$ has almost no effect on the aggregate error. In

Table 14: Ablation study of contrastive labels on FitRec and PARROTAO datasets.

| Dataset | Contrastive labels | MSE (mean ± std) | MAE (mean ± std) |
|---------|-------------------|------------------|------------------|
| FitRec | User ID only | $211.16 \pm 3.44$ | $10.50 \pm 0.05$ |
|  | Sport type only | $227.79 \pm 3.26$ | $11.10 \pm 0.06$ |
|  | User ID and Sport type | $\mathbf{204.63 \pm 3.41}$ | $\mathbf{10.28 \pm 0.05}$ |
| ParroTao | User ID only | $128.62 \pm 4.72$ | $7.79 \pm 0.12$ |
|  | Sport type only | $136.79 \pm 4.98$ | $8.19 \pm 0.11$ |
|  | User ID and Sport type | $\mathbf{125.04 \pm 4.57}$ | $\mathbf{7.53 \pm 0.11}$ |

Table 15: Demographic and Fitness Characteristics of the PARROTAO Dataset.

| Characteristic | Count (Athletes) | Percentage |
|----------------|------------------|------------|
| *Age Distribution* | | |
| 18-20 years | 30 | 26.5% |
| 21-23 years | 37 | 32.7% |
| 24-26 years | 26 | 23.0% |
| 27-30 years | 20 | 17.7% |
| *Gender Distribution* | | |
| Male | 79 | 69.9% |
| Female | 34 | 30.1% |
| *Fitness Level Distribution* | | |
| Amateur | 52 | 46.0% |
| Semi-Professional | 39 | 34.5% |
| Professional | 22 | 19.5% |
| **Total Athletes** | **113** | **100.0%** |

contrast, increasing the history length to $K=10$ reduces MSE by about 3.3% and MAE by 0.6% compared to $K=5$. Using $K=20$ workouts yields a very similar performance (slightly lower MSE and MAE), while $K=30$ further improves MAE but starts to increase MSE again. Overall, the range $K \approx 10$–20 appears to be near-optimal at the dataset level, and $K=10$ offers a convenient operating point that clearly outperforms short histories ($K=1, 5$) while avoiding the additional computational cost of $K=20$ or 30.

To examine heterogeneity across sports and users, Table 17 reports, for each $K$, how many sports and users achieve their lowest per-entity MSE at that history length. At the sport level, $8/25$ sports reach their best performance with a short history ($K=1$ or 5), $3/25$ prefer $K=10$, and $14/25$ benefit from longer histories ($K=20$ or 30). At the user level, no single value of $K$ dominates: different users prefer short, intermediate, or long histories, and $K=10$ is competitive (it is the best setting for 22.0% of users) without being extreme in either direction.

Taken together, these results suggest that: (i) using roughly ten past workouts is already sufficient to capture most of the useful historical structure when averaged over the whole dataset, (ii) there is genuine variability across sports and users, with a minority favouring very short or very long histories, and (iii) choosing $K=10$ strikes a reasonable balance between accuracy and computational cost.

## L  ADDITIONAL BASELINES

We expanded our experimental comparison to include two recent self-supervised / transformer-based time-series models: (i) PatchTST, a masked autoencoder for time series (Nie, 2022), and (ii) a transformer with rotary positional embeddings for heterogeneous time series (Rotary TS Transformer) (Su et al., 2024). The results on the FitRec and PARROTAO datasets are summarized in Table 18.

Table 16: Effect of history length $K$ (number of past workouts) on overall prediction error on the FITREC dataset. We report test MSE and MAE for each $K$. $\Delta$MSE and $\Delta$MAE denote the difference with respect to $K=10$ (negative values indicate better performance than $K=10$).

| $K$ (past workouts) | MSE | MAE | $\Delta$MSE vs. $K=10$ | $\Delta$MAE vs. $K=10$ |
|---:|---|---|---|---|
| 1 | 206.47 | 10.52 | +6.82 | +0.26 |
| 5 | 206.40 | 10.32 | +6.75 | +0.06 |
| 10 | 199.65 | 10.26 | 0.00 | 0.00 |
| 20 | 199.50 | 10.24 | -0.15 | -0.02 |
| 30 | 203.24 | 10.17 | +3.59 | -0.09 |

Table 17: Number of sports and users for which a given history length $K$ achieves the lowest per-entity MSE on FITREC. Percentages are relative to 25 sports and 91 users on the test set, respectively, and illustrate that no single history length is uniformly optimal across all entities.

| $K$ | #sports best | % sports | #users best | % users |
|---|---|---|---|---|
| 1 | 3 | 12.0% | 7 | 7.69% |
| 5 | 5 | 20.0% | 23 | 25.3% |
| 10 | 3 | 12.0% | 20 | 22.0% |
| 20 | 9 | 36.0% | 17 | 18.7% |
| 30 | 5 | 20.0% | 24 | 26.4% |

Table 18: Comparison with recent self-supervised / transformer baselines on FitRec and PARROTAO. We report test MSE and MAE (mean $\pm$ standard deviation over 200 bootstrap resamples). Lower values are better.

| Dataset | Model | MSE $\downarrow$ | MAE $\downarrow$ |
|---|---|---|---|
| FitRec | PatchTST (Nie, 2022) | $6633.374 \pm 98.070$ | $68.676 \pm 0.328$ |
| | Rotary TS Transformer (Su et al., 2024) | $331.902 \pm 3.304$ | $14.087 \pm 0.068$ |
| | **Ours** | $\mathbf{204.629 \pm 3.413}$ | $\mathbf{10.285 \pm 0.055}$ |
| PARROTAO | PatchTST (Nie, 2022) | $391.936 \pm 10.871$ | $15.668 \pm 0.203$ |
| | Rotary TS Transformer (Su et al., 2024) | $149.548 \pm 5.397$ | $8.696 \pm 0.128$ |
| | **Ours** | $\mathbf{125.043 \pm 4.569}$ | $\mathbf{7.526 \pm 0.114}$ |

On both datasets, our framework consistently achieves lower MSE and MAE than PatchTST and the Rotary TS Transformer. For the two transformer baselines, we followed their official code for implementation. We used both their original setups and also performed hyperparameter tuning to obtain the best performance for these two models. Nevertheless, their performance did not match that of our method. This suggests that, for the heart-rate prediction task considered in this work, the proposed heterogeneity-handling mechanism is competitive with, and often superior to, more complex recent architectures.

# M ATTENTION WEIGHTS OVER PAST WORKOUTS

To understand how the historical module uses past workouts, we inspect the query attention weights of the time-aware attention module on FitRec when using 10 past workouts ($K=10$). Table 19 reports the average attention weight assigned to each of the ten most recent sessions. The attention mass decays almost monotonically with the age of the workout. Thus, the model puts most emphasis on a small number of recent sessions, rather than uniformly averaging over the entire history.

We also examine sport-specific patterns by averaging attention weights within each sport and identifying, for every sport, the index of the past workout that receives the largest mean attention. As shown in Table 20, in 21 out of 25 sports ($84\%$) the maximum attention lies within the five most recent workouts, while only a small minority of sports peak at longer lags ($t_7-t_{10}$). This indicates that different sports exhibit distinct dependence patterns on historical workouts, with most sports relying primarily on very recent sessions and a few placing more weight on longer-term history.

Table 19: Average query attention weight over the ten most recent workouts on FitRec when the history length is $K{=}10$. "Cum." reports the cumulative attention mass up to $t_k$.

| | $t_1$ | $t_2$ | $t_3$ | $t_4$ | $t_5$ | $t_6$ | $t_7$ | $t_8$ | $t_9$ | $t_{10}$ |
|---|---|---|---|---|---|---|---|---|---|---|
| Mean weight | 0.207 | 0.135 | 0.115 | 0.097 | 0.085 | 0.075 | 0.073 | 0.072 | 0.071 | 0.070 |
| Cum. mass | 0.207 | 0.342 | 0.456 | 0.553 | 0.638 | 0.713 | 0.786 | 0.858 | 0.929 | 1.000 |

Table 20: Location of the maximum query attention weight per sport on FitRec when using 10 past workouts ($K{=}10$). For each lag $t_k$, we report how many sports have their largest mean attention placed on $t_k$.

| $\arg\max_k \text{ attention}(t_k)$ | $t_1$ | $t_2$ | $t_3$ | $t_4$ | $t_5$ | $t_6$ | $t_7$ | $t_8$ | $t_9$ | $t_{10}$ |
|---|---|---|---|---|---|---|---|---|---|---|
| # sports | 13 | 1 | 3 | 2 | 2 | 0 | 1 | 1 | 0 | 2 |
| % of sports | 52.0 | 4.0 | 12.0 | 8.0 | 8.0 | 0.0 | 4.0 | 4.0 | 0.0 | 8.0 |

Table 21: Wilcoxon signed-rank tests for MSE and MAE on FitRec and PARROTAO (one-sided, $H_1$: ours $<$ ablated variant). "Imp" is the relative reduction in error of our full model compared to the ablated version (in %).

| Metric | Comparison | Dataset | $p_{\text{adj}}$ | Imp (%) | $d$ | 95% CI for $d$ |
|---|---|---|---|---|---|---|
| | | | *MSE* | | | |
| MSE | ours vs. no-contrastive | FitRec | $1.00 \times 10^{-34}$ | 16.84 | -9.481 | $[-10.550, -8.678]$ |
| MSE | ours vs. no-dropout | FitRec | $1.00 \times 10^{-34}$ | 4.11 | -2.963 | $[-3.389, -2.634]$ |
| MSE | ours vs. no-TAT | FitRec | $1.32 \times 10^{-33}$ | 2.34 | -1.800 | $[-2.104, -1.557]$ |
| MSE | ours vs. no-contrastive | PARROTAO | $1.31 \times 10^{-34}$ | 12.24 | -2.289 | $[-2.586, -2.054]$ |
| MSE | ours vs. no-dropout | PARROTAO | $7.77 \times 10^{-06}$ | 1.87 | -0.332 | $[-0.470, -0.193]$ |
| MSE | ours vs. no-TAT | PARROTAO | $1.31 \times 10^{-34}$ | 13.37 | -2.551 | $[-2.890, -2.289]$ |
| | | | *MAE* | | | |
| MAE | ours vs. no-contrastive | FitRec | $1.00 \times 10^{-34}$ | 10.55 | -26.001 | $[-28.766, -23.839]$ |
| MAE | ours vs. no-dropout | FitRec | $1.00 \times 10^{-34}$ | 3.13 | -9.956 | $[-10.978, -9.085]$ |
| MAE | ours vs. no-TAT | FitRec | $1.00 \times 10^{-34}$ | 3.10 | -9.941 | $[-10.963, -9.145]$ |
| MAE | ours vs. no-contrastive | PARROTAO | $1.31 \times 10^{-34}$ | 8.30 | -3.829 | $[-4.341, -3.413]$ |
| MAE | ours vs. no-dropout | PARROTAO | $1.44 \times 10^{-30}$ | 2.92 | -1.318 | $[-1.535, -1.127]$ |
| MAE | ours vs. no-TAT | PARROTAO | $1.31 \times 10^{-34}$ | 10.39 | -4.896 | $[-5.630, -4.356]$ |

## N   ADDITIONAL STATISTICAL TEST RESULTS ON THE ABLATION STUDY

We have performed the Wilcoxon signed-rank tests (one-sided, $H_1$: ours $<$ ablated variant) on both the FitRec and PARROTAO dataset. Table 21 summarizes the results for three ablations (*no-contrastive*, *no-dropout*, and *no-TAT*) and two evaluation metrics. We report the adjusted $p$-values after multiple-comparison correction, the relative improvement in error (Imp, in %), the Wilcoxon effect size $d$, and the $95\%$ confidence intervals for $d$. Across both datasets and both metrics, all adjusted $p$-values are below $0.05$, confirming that each of the three components makes a statistically significant contribution to the overall performance.

## O   EVALUATION ON FEATURE IMPORTANCE

We analyse input feature importance at two levels: (i) global feature ablations on FitRec and PARROTAO to see how much performance degrades when individual channels are removed, (ii) sport- and user-level heterogeneity of input channels on both datasets.

### O.1   FEATURE ABLATION ON FITREC

On the FitRec dataset, the full model uses three sequential inputs: distance, altitude, and time elapsed. At inference time, we drop either the distance channel ("w/o distance") or the altitude channel ("w/o altitude"). We report MSE and MAE, averaged per sport and per user on the test set.

Table 22: Global effect of dropping distance or altitude on FitRec. We report test MSE and MAE (mean $\pm$ standard deviation over users and sports) for the full model and its ablated variants.

| Model | Metric | Full model | w/o distance | w/o altitude |
|-------|--------|------------|--------------|--------------|
| Ours | MSE | **204.63 $\pm$ 3.41** | 522.11 $\pm$ 5.60 | 284.68 $\pm$ 3.96 |
| Ours | MAE | **10.28 $\pm$ 0.05** | 17.67 $\pm$ 0.09 | 12.21 $\pm$ 0.07 |

Table 23: Overall effect of dropping each input feature on PARROTAO. For each feature, we report the test error of the ablated model ("$MSE_{drop}$" and "$MAE_{drop}$") and the change relative to the full model ($\Delta MSE$, $\Delta MAE$). Positive $\Delta$ indicates worse performance (higher error).

| Feature | $MSE_{drop}$ | $\Delta MSE$ | $MAE_{drop}$ | $\Delta MAE$ |
|---------|------------|------------|------------|------------|
| speed | 164.13 | 39.09 | 8.76 | 1.24 |
| cadence | 172.30 | 47.25 | 9.09 | 1.56 |
| power | 312.26 | 187.22 | 13.89 | 6.36 |
| stance time | 127.20 | 2.16 | 7.70 | 0.17 |
| temperature | 136.30 | 11.26 | 7.72 | 0.20 |
| enhanced altitude | 130.60 | 5.56 | 7.68 | 0.16 |
| position latitude | 134.96 | 9.92 | 7.86 | 0.33 |
| position longitude | 128.25 | 3.21 | 7.63 | 0.10 |
| step length | 137.88 | 12.83 | 8.09 | 0.56 |
| cycle length 16 | 124.70 | -0.34 | 7.53 | 0.01 |
| vertical oscillation | 125.67 | 0.62 | 7.58 | 0.05 |
| vertical ratio | 129.95 | 4.90 | 7.82 | 0.30 |
| distance in meters | 124.33 | -0.72 | 7.52 | -0.01 |
| elevation in meters | 124.07 | -0.97 | 7.51 | -0.01 |

Table 22 summarizes the global effect of these ablations. Removing distance leads to a substantial degradation in performance compared to the full model, while removing altitude has a smaller but still clearly detrimental effect.

## O.2 FEATURE ABLATION ON PARROTAO

On the PARROTAO dataset, the full model uses 14 sequential input channels. For each feature, we evaluate the model after removing that single channel at inference time on the test set, and we report the resulting MSE and MAE.

As shown in Table 23, dropping most features increases both MSE and MAE, indicating that the model consistently benefits from combining multiple heterogeneous signals. In particular, power, cadence, and speed produce the largest performance degradation when ablated, suggesting that they are globally the most influential features under our normalized representation.

## O.3 FEATURE IMPORTANCE ACROSS SPORTS AND USERS

Beyond these global averages, we also investigate how feature importance varies across sports and users. On FitRec, we compare the relative effect of dropping distance versus altitude at the sport and user levels (Table 24). For $21/25$ sports ($84.0\%$), removing distance increases MSE more than removing altitude. In contrast, $3/25$ sports ($12.0\%$) are more sensitive to altitude, and one sport ($4.0\%$) is largely insensitive to both features. A similar pattern appears across users: for $72/91$ users ($79.1\%$), distance is more important, whereas for $16/91$ users ($18.6\%$) altitude is more critical, and only $3.3\%$ of users are effectively insensitive to both.

On PARROTAO, we summarize the feature importance across sports and users in Table 25. For each feature, we report the average change in MSE when it is dropped and the fraction of sports/users for which the ablation increases MSE. Feature importance is clearly not uniform across all sports and users. Across the 19 users in the test set, power is the most critical feature for the majority of users, but cadence, speed, stance time, temperature, and distance each dominate for at least one user.

Table 24: Heterogeneity of feature importance across sports and users on FitRec. "Distance > altitude" counts sports/users for which dropping distance increases MSE more than dropping altitude, or where only the distance ablation increases MSE. Percentages are relative to 25 sports and 91 users on the test set.

| Pattern | Sports | Users |
|---|---|---|
| Distance > altitude | 21 (84.0%) | 72 (79.1%) |
| Altitude > distance | 3 (12.0%) | 16 (18.6%) |
| Neither clearly important | 1 (4.0%) | 3 (3.3%) |

Table 25: Heterogeneity of feature importance across sports and users on PARROTAO. For each feature, we report the average change in MSE when the feature is dropped (Sport/User mean $\Delta$MSE) and the fraction of sports/users for which the ablation increases MSE ($\Delta$MSE$> 0$). There are 13 sports and 19 users in the test set.

| Feature | Sport mean $\Delta$MSE | Sport $\Delta$MSE$> 0$ | User mean $\Delta$MSE | User $\Delta$MSE$> 0$ |
|---|---|---|---|---|
| speed | 149.1 | 84.6% | 47.8 | 94.7% |
| cadence | 100.3 | 76.9% | 88.1 | 100.0% |
| power | 166.4 | 84.6% | 149.4 | 94.7% |
| stance time | -3.5 | 38.5% | 7.4 | 31.6% |
| temperature | 3.5 | 61.5% | 13.3 | 57.9% |
| enhanced altitude | 5.3 | 53.9% | 13.8 | 73.7% |
| position latitude | 31.6 | 61.5% | 7.1 | 78.9% |
| position longitude | 7.8 | 53.9% | 1.1 | 57.9% |
| step length | 13.7 | 69.2% | 16.6 | 84.2% |
| cycle length 16 | -0.8 | 38.5% | 0.1 | 42.1% |
| vertical oscillation | 5.4 | 69.2% | -0.6 | 36.8% |
| vertical ratio | -2.3 | 38.5% | 6.5 | 52.6% |
| distance in meters | -1.6 | 46.2% | 8.8 | 52.6% |
| elevation in meters | -2.9 | 38.5% | -0.7 | 42.1% |

Table 26: Device-stratified performance on the PARROTAO dataset for the FitRec baseline and our model. Values are macro-averaged test MSE and MAE over all sport IDs for each device.

| Device | FitRec MSE ↓ | FitRec MAE ↓ | Ours MSE ↓ | Ours MAE ↓ |
|---|---|---|---|---|
| Huawei | 148.44 | 8.18 | **120.31** | **7.28** |
| Coros | 142.48 | 8.40 | **132.71** | **7.83** |
| Garmin | 113.54 | 7.50 | **90.19** | **6.28** |

Overall, these results highlight that, the effective importance of each channel is strongly conditioned on sport type and user-specific characteristics.

## P    EVALUATION OF PERFORMANCE STRATIFIED BY PARTICIPANT AND DEVICE

We have added a stratified evaluation to better characterize how performance varies across device types, sports, and participants. Since the FitRec dataset does not contain device labels, we stratify it only by sport and participant. In contrast, the PARROTAO dataset includes three device types (Huawei, Coros, Garmin), which allows us to analyse device- and sport-specific performance.

Table 26 reports macro-averaged MSE and MAE over all sports on PARROTAO, stratified by device type for the two best-performing models (FitRec and Ours). Our model consistently improves over the FitRec baseline across all three device types in both metrics.

To examine variability across participants, Figures 6 and 7 show density plots of per-user test MAE for the FitRec baseline and our model on the FitRec and PARROTAO datasets, respectively. For each dataset, we compute the per-participant MAE for both models and visualise the resulting distributions using normalised histograms with overlaid kernel density estimates. Across almost all users, our model either matches or improves upon the FitRec baseline, while the few extreme outliers in

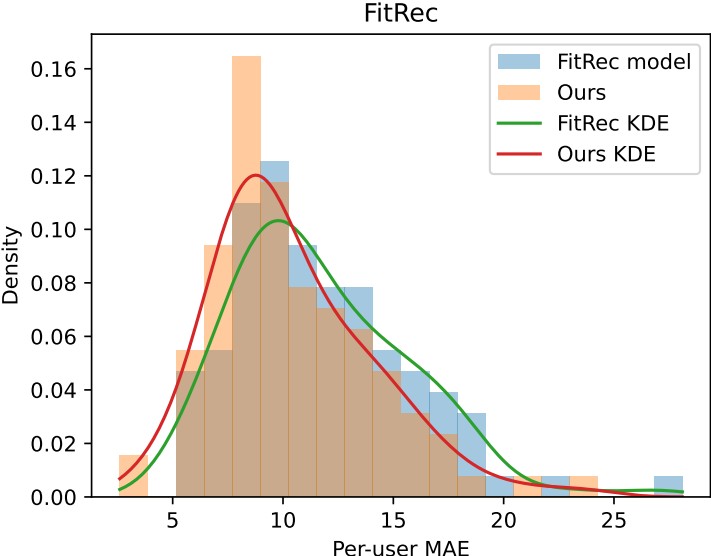

Figure 6: Histogram-like density plot of per-user test MAE for the FitRec baseline and our model on the FitRec dataset. Normalised histograms and kernel density estimates are shown for both methods.

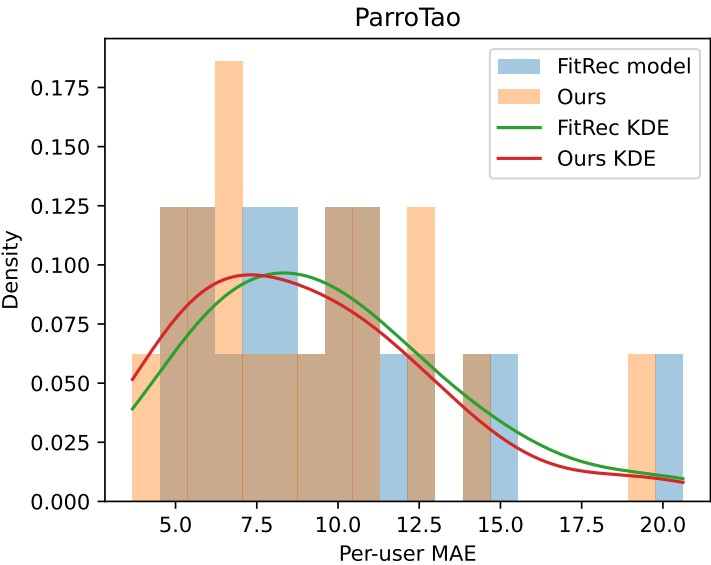

Figure 7: Histogram-like density plot of per-user test MAE for the FitRec baseline and our model on the PARROTAO dataset. Normalised histograms and kernel density estimates are shown for both methods.

the right tail correspond to participants with very few recorded workouts, for whom both models exhibit higher error due to limited data.

Finally, to study the interaction between device type and exercise type on PARROTAO, we report device- and sport-stratified MSEs for four common sports that are recorded by all three devices in Table 27. For most device–sport combinations, our model achieves lower error than the FitRec baseline.

Table 27: Device- and sport-stratified test MSE on PARROTAO for the FitRec baseline and our model. We report results for four sports (sport IDs 0, 1, 2, and 4) that are present on all three device types.

| Device | Sport Type | FitRec MSE | Ours MSE |
|---|---|---|---|
| Huawei | Running | 115.10 | **95.69** |
| Huawei | Cross-Country Run | 149.62 | 167.98 |
| Huawei | Cycling | 316.84 | **238.11** |
| Huawei | Treadmill Running | 56.83 | **24.89** |
| Coros | Running | 121.16 | **108.89** |
| Coros | Cross-Country Run | 211.65 | **210.29** |
| Coros | Cycling | **322.28** | 344.07 |
| Coros | Treadmill Running | **288.09** | 392.96 |
| Garmin | Running | 104.53 | **80.14** |
| Garmin | Cross-Country Run | 335.75 | **312.56** |
| Garmin | Cycling | 216.24 | **211.05** |
| Garmin | Treadmill Running | **442.44** | 588.50 |

Table 28: Representation quality with respect to **sport type** labels. Higher is better for RankMe, kNN, ARI, NMI, ASW, CH, homogeneity, completeness, and V-measure. Lower is better for DBI.

| Dataset | Model | RankMe | kNN@1 | kNN@5 | ARI | NMI | DBI↓ | ASW | CH | Hom. | Compl. | V-meas. |
|---|---|---|---|---|---|---|---|---|---|---|---|---|
| FitRec | FitRec | 40.91 | 0.931 | 0.920 | 0.043 | 0.110 | 1.924 | 0.089 | 2259.19 | 0.193 | 0.077 | 0.110 |
| FitRec | Ours | **61.17** | **0.945** | **0.934** | **0.051** | **0.140** | **1.508** | **0.153** | **3595.47** | **0.245** | **0.098** | **0.140** |
| PARROTAO | FitRec | 37.81 | 0.929 | **0.934** | 0.020 | 0.084 | 1.628 | **0.204** | 967.65 | 0.234 | 0.051 | 0.084 |
| PARROTAO | Ours | **52.33** | **0.934** | 0.926 | **0.024** | **0.089** | **1.420** | 0.168 | **1136.27** | **0.243** | **0.054** | **0.089** |

Table 29: Representation quality with respect to **user** labels. Higher is better for RankMe, kNN, ARI, NMI, ASW, CH, homogeneity, completeness, and V-measure. Lower is better for DBI.

| Dataset | Model | RankMe | kNN@1 | kNN@5 | ARI | NMI | DBI↓ | ASW | CH | Hom. | Compl. | V-meas. |
|---|---|---|---|---|---|---|---|---|---|---|---|---|
| FitRec | FitRec | 40.91 | 0.495 | 0.427 | 0.077 | 0.164 | 1.714 | 0.078 | 164.33 | 0.174 | 0.155 | 0.164 |
| FitRec | Ours | **61.17** | **0.713** | **0.681** | **0.091** | **0.617** | **1.462** | **0.116** | **872.16** | **0.616** | **0.618** | **0.617** |
| PARROTAO | FitRec | 37.81 | 0.369 | 0.319 | **0.097** | 0.170 | 1.614 | 0.149 | 371.51 | 0.181 | 0.160 | 0.170 |
| PARROTAO | Ours | **52.33** | **0.697** | **0.669** | 0.054 | **0.585** | **1.450** | **0.203** | **792.76** | **0.579** | **0.592** | **0.585** |

# Q QUANTITATIVE EVALUATION OF REPRESENTATION DISCRIMINABILITY

To quantitatively assess how well the learned representations separate different labels, we treat the latent vectors from the FitRec baseline and our model as embeddings and evaluate a battery of standard structure-aware metrics on both FitRec and PARROTAO. For each representation and each label space, we measure (i) local label recovery via 1- and 5-NN classification accuracy (kNN@1, kNN@5), (ii) label–representation alignment via Adjusted Rand Index (ARI), Normalized Mutual Information (NMI), homogeneity, completeness, and V-measure, and (iii) cluster quality via Davies–Bouldin Index (DBI, lower is better), average silhouette width (ASW), Calinski–Harabasz score (CH, higher is better), and a global rank-based score (RankMe). We consider three label types: *sport type*, *user identity*, and the combined *user×sport* label. The full results are reported in Tables 28–30.

Across all label granularities and on both datasets, our model consistently achieves better kNN accuracy, stronger label–embedding alignment, and more compact, better-separated clusters than the FitRec baseline. For sport labels, the gains are modest but systematic. For user identity and user×sport labels, the improvements are substantially larger despite the much higher number of classes, indicating that our embeddings differentiate individuals particularly well. This provides quantitative evidence that the proposed architecture learns representations that more cleanly organize sessions by both sport type and user-specific characteristics, which is crucial for downstream personalization and analysis.

Table 30: Representation quality with respect to joint **user×sport** labels. Higher is better for RankMe, kNN, ARI, NMI, ASW, CH, homogeneity, completeness, and V-measure. Lower is better for DBI.

| Dataset | Model | RankMe | kNN@1 | kNN@5 | ARI | NMI | DBI↓ | ASW | CH | Hom. | Compl. | V-meas. |
|---|---|---|---|---|---|---|---|---|---|---|---|---|
| FitRec | FitRec | 40.91 | 0.367 | 0.318 | 0.082 | 0.313 | 1.699 | 0.084 | 116.68 | 0.373 | 0.270 | 0.313 |
| FitRec | Ours | **61.17** | **0.492** | **0.425** | **0.115** | **0.694** | **1.525** | **0.136** | **377.73** | **0.698** | **0.689** | **0.694** |
| PARROTAO | FitRec | 37.81 | 0.437 | 0.383 | **0.088** | 0.326 | 1.570 | 0.116 | 276.04 | 0.388 | 0.280 | 0.326 |
| PARROTAO | Ours | **52.33** | **0.697** | **0.658** | 0.065 | **0.659** | **1.323** | **0.173** | **376.82** | **0.656** | **0.661** | **0.659** |

Table 31: Gender prediction from learned user representations on PARROTAO and FitRec. We report accuracy and F1-score on held-out test sets.

| Dataset | Model | Accuracy ↑ | F1-score ↑ |
|---|---|---|---|
| PARROTAO | FitRec | 0.8959 | 0.8949 |
| PARROTAO | Ours | **0.9765** | **0.9881** |
| FitRec | FitRec | 0.9346 | 0.9666 |
| FitRec | Ours | **0.9580** | **0.9785** |

## R    EVALUATING THE UTILITY OF LEARNED REPRESENTATIONS

To assess whether our learned user representations encode meaningful subject-specific information, we perform a downstream gender prediction task on both the FitRec and PARROTAO datasets. Concretely, we freeze the user embeddings produced by either the FitRec baseline or our model, and train a lightweight classifier on top of these embeddings to predict binary gender. We report accuracy and F1-score on held-out test sets.

As shown in Table 31, both sets of embeddings support strong gender classification, confirming that user identity information is indeed present in the learned representations. Moreover, our model consistently achieves higher accuracy and F1-score on both datasets, indicating that it produces more informative and discriminative user embeddings. Together with the downstream tasks in A7, where we successfully predict user-related physiological quantities, these results provide converging evidence that the proposed framework captures stable, user-specific characteristics that are useful for personalization.

## S    ADDITIONAL EVALUATION OF DOWNSTREAM APPLICATIONS

We have expanded the downstream evaluation beyond route recommendation to further demonstrate the practical utility of the learned representations and the proposed architecture. In addition to the original route-planning scenario, we now consider two additional families of tasks: cross-device feature imputation and heart-rate imputation under missing segments.

### S.1    CROSS-DEVICE FEATURE IMPUTATION ON HUAWEI DEVICES.

We first conduct a cross-device experiment on the PARROTAO subset collected using Huawei wearables, mimicking a deployment scenario where a target device does not expose certain sensor channels that are available on other devices. We train models on Huawei workouts to reconstruct three Huawei-specific targets (calories, stance time, vertical oscillation) from the remaining inputs, and evaluate on a held-out test split.

Table 32 compares the FitRec baseline with our model in terms of MSE and MAE. Across all three targets, our approach substantially reduces both error metrics, showing that it can serve as an effective cross-device imputation module for recovering missing sensor channels.

Table 32: Cross-device feature imputation on the PARROTAO–Huawei subset. We predict three target channels (calories, stance time, vertical oscillation) from the remaining inputs. For each metric we report mean $\pm$ standard deviation.

| Target | Model | MSE $\downarrow$ | MAE $\downarrow$ |
|---|---|---|---|
| Calories | FitRec | $0.03408 \pm 1.20 \times 10^{-4}$ | $0.12439 \pm 6.80 \times 10^{-6}$ |
| | Ours | $\mathbf{0.01275 \pm 3.80 \times 10^{-7}}$ | $\mathbf{0.01607 \pm 1.84 \times 10^{-5}}$ |
| Stance time | FitRec | $0.00929 \pm 2.79 \times 10^{-6}$ | $0.09491 \pm 2.22 \times 10^{-5}$ |
| | Ours | $\mathbf{3.78 \times 10^{-4} \pm 3.63 \times 10^{-7}}$ | $\mathbf{0.01284 \pm 1.53 \times 10^{-5}}$ |
| Vertical oscillation | FitRec | $0.00394 \pm 4.59 \times 10^{-7}$ | $0.05856 \pm 2.65 \times 10^{-6}$ |
| | Ours | $\mathbf{8.29 \times 10^{-4} \pm 8.20 \times 10^{-7}}$ | $\mathbf{0.02495 \pm 1.70 \times 10^{-5}}$ |

Table 33: Cross-device feature imputation on the PARROTAO–Garmin subset. We predict three target channels (body battery, performance condition, vertical ratio) from the remaining inputs. For each metric we report mean $\pm$ standard deviation.

| Target | Model | MSE $\downarrow$ | MAE $\downarrow$ |
|---|---|---|---|
| body_battery | FitRec | $0.00518 \pm 0.00235$ | $0.02208 \pm 0.00146$ |
| | Ours | $\mathbf{7.44 \times 10^{-4} \pm 8.62 \times 10^{-4}}$ | $\mathbf{0.00143 \pm 5.26 \times 10^{-4}}$ |
| performance_condition | FitRec | $1.83 \times 10^{-5} \pm 6.35 \times 10^{-6}$ | $2.34 \times 10^{-3} \pm 7.61 \times 10^{-5}$ |
| | Ours | $\mathbf{1.85 \times 10^{-5} \pm 2.14 \times 10^{-5}}$ | $\mathbf{8.23 \times 10^{-4} \pm 8.11 \times 10^{-5}}$ |
| vertical_ratio | FitRec | $0.0837 \pm 0.0562$ | $0.0147 \pm 0.00629$ |
| | Ours | $\mathbf{0.0574 \pm 0.0492}$ | $\mathbf{0.0187 \pm 0.00493}$ |

### S.2 CROSS-DEVICE FEATURE IMPUTATION ON GARMIN DEVICES.

We further evaluate cross-device imputation on the PARROTAO subset collected with Garmin wearables. Here we reconstruct three Garmin-specific targets (body battery, performance condition, vertical ratio) from the remaining inputs, again training on a subset and evaluating on a held-out test split.

As summarised in Table 33, our model is competitive or superior to the FitRec baseline across these targets, confirming that the proposed architecture can generalise as a flexible imputation module for different device-specific channels.

### S.3 HEART-RATE IMPUTATION.

Finally, we evaluate whether the framework can also support imputing missing heart-rate (HR) values, a common issue for wrist-based sensors. We construct masked variants of the FitRec and PARROTAO datasets (denoted `fitrec-mask` and `parrotao-mask`) by randomly masking short HR segments so that roughly $20\%$ of time steps are missing in each sequence. Given the original covariates, the partially masked HR sequence, and a binary mask indicating missing positions, each model is trained to reconstruct the full HR trajectory, with the loss computed only on masked time steps.

We compare our model against four baselines: (i) the FitRec model, (ii) a Kalman filter, (iii) linear interpolation, and (iv) last-observation-carried-forward (LOCF). Table 34 reports MSE on masked positions for both datasets. Our method achieves the lowest error across all settings, substantially outperforming classical interpolation and filtering methods as well as the FitRec baseline. This shows that the learned representations are not only useful for forecasting, but also for high-quality HR imputation, enabling smoother and more informative post-workout HR profiles.

Table 34: Heart-rate imputation on masked HR sequences (FitRec and PARROTAO). We report mean ± standard deviation of MSE over masked positions (lower is better).

| Dataset | Method | MSE ↓ |
|---------|--------|-------|
| FitRec | FitRec | $69.78 \pm 0.33$ |
| | Kalman | $832.61 \pm 1.77$ |
| | Linear interpolation | $91.11 \pm 0.15$ |
| | LOCF | $174.00 \pm 0.44$ |
| | **Ours** | $\mathbf{40.84 \pm 0.17}$ |
| PARROTAO | FitRec | $19.72 \pm 0.66$ |
| | Kalman | $167.41 \pm 0.48$ |
| | Linear interpolation | $34.81 \pm 0.15$ |
| | LOCF | $81.92 \pm 0.64$ |
| | **Ours** | $\mathbf{7.54 \pm 0.08}$ |

