# OpenReview forum: "Learning Representations from Heterogeneous Data for Robust Heart Rate Modeling"
_ICLR.cc/2026/Conference — ICLR 2026 Conference Withdrawn Submission_

### Official Review · Reviewer_xP3V · 2025-10-15

**Soundness:** 2
**Presentation:** 2
**Contribution:** 2
**Rating:** 4
**Confidence:** 5

**Summary:**

The author introduced a framework that could predict the expected heart rate given a planned activity. A new benchmark dataset is collected and released. The proposed modeling framework contains a dropout layer at the input layer and is trained using contrastive loss to tackle the issue of inter-subject variability. The model achieves the best performance against the baselines.

**Strengths:**

- The problem addressed is meaningful and has strong potential for real-world applications, especially in personalized health monitoring.
- The paper demonstrates a solid applied deep learning effort, combining practical motivation with technical implementation.
- The Group 4 experiment is particularly compelling. It highlights how users can interactively adjust their activity plans based on forecasted heart rate, showing promising translational potential of the model.

**Weaknesses:**

- The selection of baselines could be expanded to include more recent or diverse methods, which would help strengthen the empirical validation. Because the objective is a channel imputation task, there are multiple open sourced and pre-trained time series models that build upon masked autoencoders, for example [1], how is the proposed framework compared to those approaches?
- The way to address heterogeneity is a bit of common sense, and none of the newer and more effective methods is discussed and compared, for example there are varied works proposing modality agnostic model backbone, or channel aware attention, etc, [2] as an example.
- The paper lacks fine-grained analysis on feature contributions. For instance, how much the current input (X_curr) versus historical input (X_hist) influences the model’s predictions. Also the ablation study suggests limited improvement from the proposed modules, which could be discussed more clearly or justified through an additional statistical test result.
- The scope of the contribution (method, dataset, or task formulation) is a bit confusing. It could be articulated more explicitly to help readers better understand the paper’s core advancement.

[1] Nie, Y. "A Time Series is Worth 64Words: Long-term Forecasting with Transformers." arXiv preprint 2022

[2] Su, Jianlin, et al. "Roformer: Enhanced transformer with rotary position embedding." Neurocomputing 2024

**Questions:**

The questions are mostly stated in the weakness above. I've summarized, in terms of the expected improvement, in below:
- How is the proposed framework compared to those self-supervised approach?
- How is the proposed method for handling heterogeneity compared to more up-to-date backbone deep learning module?
- What is the difference of contribution from current input (X_curr) versus historical input (X_hist)? And which feature(s) catch more attention from the trained model during inference time? More analysis on this aspect would provide more insights on the model behaviors.
- Is the contribution lies more on the method or data aspects?

---

> ### Author Response · Authors · 2025-11-20
> **Reply to Reviewer xP3V: Part 1**
>
> We sincerely appreciate the valuable feedback provided by the reviewer. We have carefully considered the reviewer's comments and have made the necessary revisions to our manuscript. Below, we would like to address the reviewer's questions and provide additional information:
>
> **Q1. How does the proposed framework compare with existing self-supervised approaches, and how does the proposed heterogeneity-handling mechanism perform relative to more recent deep learning architectures?**
>
> **A1.**
> We expanded our experimental comparison to include two recent self-supervised / transformer-based time-series models: (i) PatchTST, a masked autoencoder for time series [1], and (ii) a transformer with rotary positional embeddings for heterogeneous time series (Rotary TS Transformer) [2]. The results on the FitRec and *ParroTao* datasets are summarized in Table 1.
>
> On both datasets, our framework consistently achieves lower MSE and MAE than PatchTST and the Rotary TS Transformer. For the two transformer baselines, we followed their official code for implementation. We used both their original setups and also performed hyperparameter tuning to obtain the best performance for these two models. Nevertheless, their performance did not match that of our method. This suggests that, for the heart-rate prediction task considered in this work, the proposed heterogeneity-handling mechanism is competitive with, and often superior to, more complex recent architectures.
>
> These empirical findings are also consistent with prior observations that generic transformer backbones, while strong on broad benchmark suites, may underperform simpler or more tailored architectures on specific forecasting tasks if not carefully adapted [3]. We therefore view the design of transformer-based models that more explicitly exploit the structure of heterogeneous wearable data as an interesting direction for future research, and our framework provides a concrete baseline for such developments.
>
> **Table 1. Comparison with recent self-supervised / transformer baselines on FitRec and *ParroTao*.**
> We report test MSE and MAE (mean ± standard deviation over 200 bootstrap resamples). Lower values are better.
>
> | Dataset                   | Model                     | MSE ↓ | MAE ↓ |
> |---------------------------|---------------------------|-------|-------|
> | FitRec                    | PatchTST [1]              | 6633.374 ± 98.070 | 68.676 ± 0.328 |
> |                           | Rotary TS Transformer [2] | 331.902 ± 3.304 | 14.087 ± 0.068 |
> |                           | **Ours**                  | **204.629 ± 3.413** | **10.285 ± 0.055** |
> | *ParroTao* | PatchTST [1]              | 391.936 ± 10.871 | 15.668 ± 0.203 |
> |                           | Rotary TS Transformer [2] | 149.548 ± 5.397 | 8.696 ± 0.128 |
> |                           | **Ours**                  | **125.043 ± 4.569** | **7.526 ± 0.114** |

---

> ### Author Response · Authors · 2025-11-20
> **Reply to Reviewer xP3V: Part 2**
>
> **Q2. What is the difference between the contribution of the current input ($X_{curr}$) and that of the historical input ($X_{hist}$)? Additionally, which feature(s) are most important?**
>
> **A2.**
> We provide a more fine-grained analysis of how $X_{\text{curr}}$ and $X_{\text{hist}}$ contribute to the predictions, and which features are most influential. The results fall into four parts: (i) ablation of the historical input on both datasets, (ii) analysis of attention weights over past workouts when using a history of 10 sessions, (iii) sensitivity to the number of historical workouts, and (iv) feature importance analysis.
>
> **(i) Contribution of $X_{\text{curr}}$ vs. $X_{\text{hist}}$.**
> We first train a variant that uses only the current-session input $X_{\text{curr}}$ (i.e., no historical module), and compare it to the full model that combines $X_{\text{curr}}$ and $X_{\text{hist}}$. Table 2 reports the results on FitRec and *ParroTao*. Across both datasets, removing $X_{\text{hist}}$ leads to a consistent degradation in MSE and MAE. This shows that $X_{\text{curr}}$ already explains most of the predictive performance, while $X_{\text{hist}}$ provides a complementary gain by capturing user-specific adaptation patterns across workouts.
>
> **Table 2. Ablation of the historical input $X_{\text{hist}}$ on the FitRec and *ParroTao* datasets.**
> We report test MSE and MAE (mean ± standard deviation over 200 bootstrap resamples). “Curr only” denotes the variant that uses only $X_{\text{curr}}$, without the historical module.
>
> | Model Variant | FitRec MSE ↓ | FitRec MAE ↓ | *ParroTao* MSE ↓ | *ParroTao* MAE ↓ |
> |---------------|-------------|--------------|------------------|------------------|
> | Full ($X_{\text{curr}}{+}X_{\text{hist}}$) | **204.63 ± 3.41** | **10.28 ± 0.05** | **125.04 ± 4.57** | **7.53 ± 0.11** |
> | Curr only ($X_{\text{curr}}$ only) | 213.40 ± 3.15 | 10.62 ± 0.05 | 127.42 ± 5.04 | 7.75 ± 0.11 |
>
> **(ii) How the model uses $X_{\text{hist}}$ (attention over past workouts).**
> To understand how the historical module uses past workouts, we inspect the query attention weights of the time-aware attention module on FitRec when using 10 past workouts ($K{=}10$). Table 3 reports the average attention weight assigned to each of the ten most recent sessions. The attention mass decays almost monotonically with the age of the workout. Thus, the model puts most emphasis on a small number of recent sessions, rather than uniformly averaging over the entire history.
>
> **Table 3. Average query attention weight over the ten most recent workouts on FitRec when the history length is $K{=}10$. “Cum.” reports the cumulative attention mass up to $t_k$.**
>
> |                | t₁    | t₂    | t₃    | t₄    | t₅    | t₆    | t₇    | t₈    | t₉    | t₁₀   |
> |----------------|-------|-------|-------|-------|-------|-------|-------|-------|-------|-------|
> | Mean weight    | 0.207 | 0.135 | 0.115 | 0.097 | 0.085 | 0.075 | 0.073 | 0.072 | 0.071 | 0.070 |
> | Cum. mass      | 0.207 | 0.342 | 0.456 | 0.553 | 0.638 | 0.713 | 0.786 | 0.858 | 0.929 | 1.000 |
>
> We also examine sport-specific patterns by averaging attention weights within each sport and identifying, for every sport, the index of the past workout that receives the largest mean attention. As shown in Table 4, in 21 out of 25 sports ($84\%$) the maximum attention lies within the five most recent workouts, while only a small minority of sports peak at longer lags ($t_7$--$t_{10}$). This indicates that different sports exhibit distinct dependence patterns on historical workouts, with most sports relying primarily on very recent sessions and a few placing more weight on longer-term history.
>
> **Table 4. Location of the maximum query attention weight per sport on FitRec when using 10 past workouts ($K = 10$).**
> For each lag $t_k$, we report how many sports have their largest mean attention placed on $t_k$.
>
> | $\arg\max_k \text{ attention}(t_k)$ | $t_1$ | $t_2$ | $t_3$ | $t_4$ | $t_5$ | $t_6$ | $t_7$ | $t_8$ | $t_9$ | $t_{10}$ |
> |--------------------------------------|---------|---------|---------|---------|---------|---------|---------|---------|---------|----------|
> | **# sports**                         | 13      | 1       | 3       | 2       | 2       | 0       | 1       | 1       | 0       | 2        |
> | **% of sports**                      | 52.0    | 4.0     | 12.0    | 8.0     | 8.0     | 0.0     | 4.0     | 4.0     | 0.0     | 8.0      |

---

> ### Author Response · Authors · 2025-11-20
> **Reply to Reviewer xP3V: Part 3**
>
> **(iii) Effect of the history length $K$.**
> Finally, we study how performance changes with the number of historical workouts provided to the module. We evaluate $K \in \{1,5,10,20,30\}$ on the *FitRec* dataset and report overall MSE and MAE in Table 5. Moving from $K{=}1$ to $K{=}5$ has almost no effect on the aggregate error. In contrast, increasing the history length to $K{=}10$ reduces MSE and MAE compared to $K{=}5$. Using $K{=}20$ workouts yields very similar performance (slightly lower MSE and MAE), while $K{=}30$ further improves MAE but starts to increase MSE again. Overall, the range $K \approx 10$--$20$ appears to be near-optimal at the dataset level, and $K{=}10$ offers a convenient operating point that clearly outperforms short histories while avoiding the additional computational cost of $K{=}20$ or $30$.
>
> To examine heterogeneity across sports and users, Table 6 reports, for each $K$, how many sports and users achieve their lowest per-entity MSE at that history length. At the sport level, $8/25$ sports achieve their best performance with a short history ($K{=}1$ or $5$), $3/25$ prefer $K{=}10$, and $14/25$ benefit from longer histories ($K{=}20$ or $30$). At the user level, no single value of $K$ dominates: different users prefer short, intermediate, or long histories, and $K{=}10$ is competitive (it is the best setting for $22.0\%$ of users) without being extreme in either direction. Taken together, these findings suggest that using roughly ten past workouts is already sufficient to capture most of the useful historical structure on average, while still allowing individual sports and users to benefit from longer histories.
>
> **Table 5. Effect of history length $K$ (number of past workouts) on overall prediction error on the *FitRec* dataset.**
> We report test MSE and MAE for each $K$. $Δ$MSE and $Δ$MAE denote the difference with respect to $K = 10$ (negative values indicate better performance than $K = 10$).
>
> | K (past workouts) | MSE ↓ | MAE ↓ | ΔMSE vs. K = 10 | ΔMAE vs. K = 10 |
> |-------------------|-------|-------|-----------------|-----------------|
> | 1  | 206.47 | 10.52 | +6.82 | +0.26 |
> | 5  | 206.40 | 10.32 | +6.75 | +0.06 |
> | 10 | 199.65 | 10.26 | 0.00  | 0.00  |
> | 20 | 199.50 | 10.24 | −0.15 | −0.02 |
> | 30 | 203.24 | 10.17 | +3.59 | −0.09 |
>
> **Table 6. Number of sports and users for which a given history length $K$ achieves the lowest per-entity MSE on *FitRec*.**
> Percentages are relative to 25 sports and 91 users on the test set, respectively, and illustrate that no single history length is uniformly optimal across all entities.
>
> | K | #sports best | % sports | #users best | % users |
> |---|--------------|----------|-------------|---------|
> | 1  | 3 | 12.0 % | 7  | 7.69 % |
> | 5  | 5 | 20.0 % | 23 | 25.3 % |
> | 10 | 3 | 12.0 % | 20 | 22.0 % |
> | 20 | 9 | 36.0 % | 17 | 18.7 % |
> | 30 | 5 | 20.0 % | 24 | 26.4 % |

---

> ### Author Response · Authors · 2025-11-20
> **Reply to Reviewer xP3V: Part 4**
>
> **(iv) Feature importance analysis.**
> We analyse input feature importance at two levels: (i) global feature ablations on FitRec and *ParroTao* to see how much performance degrades when individual channels are removed, (ii) sport- and user-level heterogeneity of input channels on both datasets.
>
> **Feature ablation on FitRec.**
> On the FitRec dataset, the full model uses three sequential inputs: distance, altitude, and time elapsed. At inference time, we drop either the distance channel ("w/o distance") or the altitude channel ("w/o altitude"). We report MSE and MAE, averaged per sport and per user on the test set.
>
> Table 2 summarizes the global effect of these ablations. Removing distance leads to a substantial degradation in performance compared to the full model, while removing altitude has a smaller but still clearly detrimental effect.
>
> **Table 2. Global effect of dropping distance or altitude on FitRec.**
> We report test MSE and MAE (mean ± standard deviation over users and sports) for the full model and its ablated variants.
>
> | Model | Metric | Full model | w/o distance | w/o altitude |
> |-------|--------|-----------:|-------------:|-------------:|
> | Ours  | MSE    | **204.63 ± 3.41** | 522.11 ± 5.60 | 284.68 ± 3.96 |
> | Ours  | MAE    | **10.28 ± 0.05** | 17.67 ± 0.09 | 12.21 ± 0.07 |
>
> **Feature ablation on *ParroTao*.**
> On the *ParroTao* dataset, the full model uses 14 sequential input channels. For each feature, we evaluate the model after removing that single channel at inference time on the test set, and we report the resulting MSE and MAE.
>
> As shown in Table 3, dropping most features increases both MSE and MAE, indicating that the model consistently benefits from combining multiple heterogeneous signals. In particular, power, cadence, and speed produce the largest performance degradation when ablated, suggesting that they are globally the most influential features under our normalized representation.
>
> **Table 3. Overall effect of dropping each input feature on *ParroTao*.**
> For each feature, we report the test error of the ablated model ("$MSE_{drop}$" and “$MAE_{drop}$”) and the change relative to the full model (ΔMSE, ΔMAE). Positive Δ indicates worse performance (higher error).
>
> | Feature              | $MSE_{drop}$ |   ΔMSE | $MAE_{drop}$ |  ΔMAE |
> |----------------------|-------------:|-------:|-------------:|------:|
> | speed                |       164.13 |  39.09 |         8.76 |  1.24 |
> | cadence              |       172.30 |  47.25 |         9.09 |  1.56 |
> | power                |       312.26 | 187.22 |        13.89 |  6.36 |
> | stance time          |       127.20 |   2.16 |         7.70 |  0.17 |
> | temperature          |       136.30 |  11.26 |         7.72 |  0.20 |
> | enhanced altitude    |       130.60 |   5.56 |         7.68 |  0.16 |
> | position latitude    |       134.96 |   9.92 |         7.86 |  0.33 |
> | position longitude   |       128.25 |   3.21 |         7.63 |  0.10 |
> | step length          |       137.88 |  12.83 |         8.09 |  0.56 |
> | cycle length 16      |       124.70 |  -0.34 |         7.53 |  0.01 |
> | vertical oscillation |       125.67 |   0.62 |         7.58 |  0.05 |
> | vertical ratio       |       129.95 |   4.90 |         7.82 |  0.30 |
> | distance in meters   |       124.33 |  -0.72 |         7.52 | -0.01 |
> | elevation in meters  |       124.07 |  -0.97 |         7.51 | -0.01 |

---

> ### Author Response · Authors · 2025-11-20
> **Reply to Reviewer xP3V: Part 5**
>
> **Heterogeneity of feature importance across sports and users.**
> Beyond these global averages, we also investigate how feature importance varies across sports and users. On FitRec, we compare the relative effect of dropping distance versus altitude at the sport and user levels (Table 4). For $21/25$ sports ($84.0%$), removing distance increases MSE more than removing altitude. In contrast, $3/25$ sports ($12.0%$) are more sensitive to altitude, and one sport ($4.0%$) is largely insensitive to both features. A similar pattern appears across users: for $72/91$ users ($79.1%$), distance is more important, whereas for $16/91$ users ($18.6%$) altitude is more critical, and only $3.3%$ of users are effectively insensitive to both.
>
> **Table 4. Heterogeneity of feature importance across sports and users on FitRec.**
> “Distance > altitude” counts sports/users for which dropping distance increases MSE more than dropping altitude, or where only the distance ablation increases MSE. Percentages are relative to 25 sports and 91 users in the test set.
>
> | Pattern                    | Sports | Users |
> |----------------------------|:------:|:-----:|
> | Distance > altitude        | 21 (84.0 %) | 72 (79.1 %) |
> | Altitude > distance        | 3 (12.0 %)  | 16 (18.6 %) |
> | Neither clearly important  | 1 (4.0 %)   | 3 (3.3 %) |
>
> On *ParroTao*, we summarize the feature importance across sports and users in Table 5. For each feature, we report the average change in MSE when it is dropped and the fraction of sports/users for which the ablation increases MSE. Feature importance is clearly not uniform across all sports and users. Across the 19 users in the test set, power is the most critical feature for the majority of users, but cadence, speed, stance time, temperature, and distance each dominate for at least one user. Overall, these results highlight that, the effective importance of each channel is strongly conditioned on sport type and user-specific characteristics.
>
> **Table 5. Heterogeneity of feature importance across sports and users on *ParroTao*.**
> For each feature, we report the average change in MSE when the feature is dropped (Sport/User mean ΔMSE) and the fraction of sports/users for which the ablation increases MSE (ΔMSE > 0). There are 13 sports and 19 users in the test set.
>
> | Feature               | Sport mean ΔMSE | Sport ΔMSE > 0 | User mean ΔMSE | User ΔMSE > 0 |
> |-----------------------|----------------:|---------------:|---------------:|--------------:|
> | speed                 | 149.1 | 84.6 % |  47.8 | 94.7 % |
> | cadence               | 100.3 | 76.9 % |  88.1 |100.0 % |
> | power                 | 166.4 | 84.6 % | 149.4 | 94.7 % |
> | stance time           |  −3.5 | 38.5 % |   7.4 | 31.6 % |
> | temperature           |   3.5 | 61.5 % |  13.3 | 57.9 % |
> | enhanced altitude     |   5.3 | 53.9 % |  13.8 | 73.7 % |
> | position latitude     |  31.6 | 61.5 % |   7.1 | 78.9 % |
> | position longitude    |   7.8 | 53.9 % |   1.1 | 57.9 % |
> | step length           |  13.7 | 69.2 % |  16.6 | 84.2 % |
> | cycle length 16       |  −0.8 | 38.5 % |   0.1 | 42.1 % |
> | vertical oscillation  |   5.4 | 69.2 % | −0.6 | 36.8 % |
> | vertical ratio        |  −2.3 | 38.5 % |   6.5 | 52.6 % |
> | distance in meters    |  −1.6 | 46.2 % |   8.8 | 52.6 % |
> | elevation in meters   |  −2.9 | 38.5 % | −0.7 | 42.1 % |

---

> ### Author Response · Authors · 2025-11-20
> **Reply to Reviewer xP3V: Part 6**
>
> **Q3. Additional statistical test results on the ablation study.**
>
> **A3.**
> We have performed the Wilcoxon signed-rank tests (one-sided, $H_1$: ours $<$ ablated variant) on both the FitRec and *ParroTao* dataset. Table 7 summarizes the results for three ablations (**no-contrastive**, **no-dropout**, and **no-TAT**) and two evaluation metrics. We report the adjusted $p$-values after multiple-comparison correction, the relative improvement in error (Imp, in~\%), the Wilcoxon effect size $d$, and the $95\%$ confidence intervals for $d$. Across both datasets and both metrics, all adjusted $p$-values are below $0.05$, confirming that each of the three components makes a statistically significant contribution to the overall performance.
>
> **Table 7. Wilcoxon signed-rank tests for MSE and MAE on FitRec and *ParroTao* (one-sided, $H_1$: ours < ablated variant). “Imp” is the relative reduction in error of our full model compared to the ablated version (in %).**
>
> | Metric | Comparison | Dataset | p_adj | Imp (%) | d | 95 % CI for d       |
> |--------|------------|---------|-------|---------|----|---------------------|
> | *MSE* |  |  |  |  |  |                     |
> | MSE | ours vs. no-contrastive | FitRec     | $1.00\times 10^{-34}$ | 16.84 | −9.481 | $[−10.550, −8.678]$ |
> | MSE | ours vs. no-dropout     | FitRec     | $1.00\times 10^{-34}$ |  4.11 | −2.963 | $ [−3.389, −2.634]$ |
> | MSE | ours vs. no-TAT         | FitRec     | $1.32\times 10^{-33}$ |  2.34 | −1.800 | $ [−2.104, −1.557]$ |
> | MSE | ours vs. no-contrastive | *ParroTao* | $1.31\times 10^{-34}$ | 12.24 | −2.289 | $ [−2.586, −2.054]$ |
> | MSE | ours vs. no-dropout     | *ParroTao* | $7.77\times 10^{-06}$ |  1.87 | −0.332 | $ [−0.470, −0.193]$   |
> | MSE | ours vs. no-TAT         | *ParroTao* | $1.31\times 10^{-34}$ | 13.37 | −2.551 | $ [−2.890, −2.289]$   |
> | *MAE* |  |  |  |  |  |                     |
> | MAE | ours vs. no-contrastive | FitRec     | $1.00\times 10^{-34}$ | 10.55 | −26.001 | $[−28.766, −23.839]$  |
> | MAE | ours vs. no-dropout     | FitRec     | $1.00\times 10^{-34}$ |  3.13 |  −9.956 | $[−10.978, −9.085]$   |
> | MAE | ours vs. no-TAT         | FitRec     | $1.00\times 10^{-34}$ |  3.10 |  −9.941 | $[−10.963, −9.145]$   |
> | MAE | ours vs. no-contrastive | *ParroTao* | $1.31\times 10^{-34}$ |  8.30 |  −3.829 | $ [−4.341, −3.413]$   |
> | MAE | ours vs. no-dropout     | *ParroTao* | $1.44\times 10^{-30}$ |  2.92 |  −1.318 | $ [−1.535, −1.127]$   |
> | MAE | ours vs. no-TAT         | *ParroTao* | $1.31\times 10^{-34}$ | 10.39 |  −4.896 | $ [−5.630, −4.356]$   |
>
>
> **Q4. What is the scope of this paper’s contribution?**
>
> **A4.**
> We have clarified the scope of our contribution more explicitly in the revised manuscript. Our work is motivated by the challenge of handling heterogeneous wearable data for personalized fitness assessment, and our contributions are twofold. On the methodological side, we propose a simple yet effective architecture that robustly integrates current and historical multi-channel signals under missing and non-uniform feature patterns. **All key design choices in our model are systematically evaluated through ablation and sensitivity studies on two datasets, demonstrating the effectiveness and robustness of the proposed framework.**
>
> On the data side, we construct and publicly release the *ParroTao* dataset, a large-scale, multi-sport, multi-device cohort specifically designed to study heterogeneity in real-world wearable data. Unlike typical benchmarks that homogenize inputs by keeping only the intersection of features across devices, *ParroTao* deliberately preserves device-specific feature sets and cross-user variation, providing a more realistic and rigorous testbed for models that aim to operate under real-world heterogeneity. We have revised the final paragraph of the introduction to clearly articulate these two aspects as the core contributions of the paper.
>
>
> **References**
>
> [1] Nie, Y. (2022). A time series is worth 64words: Long-term forecasting with transformers. arXiv preprint arXiv:2211.14730 .
>
> [2] Su, J., Ahmed, M., Lu, Y., Pan, S., Bo, W., & Liu, Y. (2024). Roformer: Enhanced transformer with rotary position embedding. Neurocomputing, 568 , 127063.
>
> [3] Zeng, A., Chen, M., Zhang, L., & Xu, Q. (2022). Are transformers effective for time series forecasting? Retrieved from https://arxiv.org/abs/2205.13504

---

> ### Comment · Reviewer_xP3V · 2025-11-20
> **Response to Authors**
>
> Thank you for the additional and comprehensive experimental results. After carefully reading these results and the revised manuscript, I believe most of my concerns are sufficiently addressed, including
> - Additional baseline added to comparison which better contextualize the performance of the proposed framework.
> - Very detailed analysis on the contribution of the input components, along with well-rounded discussion of the observation. The results are reasonable, for example, the model put more attention on the recent historical data.
>
> The paper now have clearer claim on the contribution (both the method and data aspect) and in-depth experimentation under varied settings. I have updated my score accordingly.
>
> ## Remaining Concern
> My last concern is mostly on the second question. I may have previously conceptualized the notion in a misplaced way, but I am particularly referring to Section 3.2 (in the revised version), the Time-aware Attention module. This module describes how multi-historical data is handled, while each past record is also time-series data. Since several prior works have addressed modeling data of this type, a comparison could sufficiently addressed this concern through a discussion of the differences and a complexity analysis (while empirical evidence definitely could help but not strictly expected) .
>
> For example, the most recent work, Chronos-2 [1], though it is too recent and not yet published, uses the exact same modeling logic (for intra-channel and inter-channel) as CBraMod [2] and Panda [3]. Another approach with lower complexity is proposed by NormWear [4], which uses a representative token embedding (e.g., [CLS] or mean of the intra-channel encoder output) for inter-channel processing. This is similar to what your paper proposes, where the embedding from the last time step of the intra-channel encoder is leveraged (if I understand correctly). A discussion of these methods, along with complexity analysis and trade-offs, would help better contextualize the proposed approach.
>
> ## Reference
> [1] Ansari, Abdul Fatir, et al. "Chronos-2: From Univariate to Universal Forecasting." arXiv preprint 2025
>
> [2] Wang, Jiquan, et al. "Cbramod: A criss-cross brain foundation model for eeg decoding." ICLR 2025
>
> [3] Lai, Jeffrey, Anthony Bao, and William Gilpin. "Panda: A pretrained forecast model for universal representation of chaotic dynamics." arXiv preprint 2025
>
> [4] Luo, Yunfei, et al. "Toward Foundation Model for Multivariate Wearable Sensing of Physiological Signals." arXiv preprint, 2024.

---

> > ### Author Response · Authors · 2025-11-28
> > **Reply to Reviewer xP3V: Part 7**
> >
> > **Q5. Comparison between our time-aware attention module and existing multi-historical time-series models.**
> >
> > **A5.**  We thank the reviewer for pointing us to Chronos-2 [1] and NormWear [2]. Below we (i) clarify the structural relationship between these models, (ii) provide a theoretical comparison of the inference-time complexity, and (iii) report an empirical FLOPs comparison on the FitRec dataset.
> >
> > Our time-aware attention module follows the same general "two-axis" idea as these works, but operates at a different granularity and is tailored to our personalized heart-rate prediction setting.
> >
> > Chronos-2 alternates time-attention and group-attention layers over patch tokens. Let $C$ denote the number of series in a group, $L$ the number of patches per series, and $d$ the hidden dimension. Then a single transformer block in Chronos-2 has a per-layer attention complexity
> > $$
> >   \mathcal{O}\bigl(d (C L^2 + L C^2)\bigr)
> >   \=\
> >   \mathcal{O}\bigl(d L C (L + C)\bigr).
> > $$
> >
> > NormWear adopts a conceptually similar representative-token strategy at the channel level. Each channel is first processed by a transformer backbone whose per-layer self-attention cost is $\mathcal{O}(C L^2 d)$. After the backbone produces a per-channel [CLS] token for each of the $C$ channels, the [CLS]-attention fusion performs self-attention only over these tokens, with complexity $\mathcal{O}(d C^2)$, independent of the sequence length $L$. A temporal fusion module (MSiTF) then aggregates patch-level information, adding a linear $\mathcal{O}(L C d)$ term. Collecting these terms, a forward pass of the full NormWear model can be upper-bounded by
> > $$
> >   \mathcal{O}(C L^2 d)
> >   \+\
> >   \mathcal{O}(d C^2)
> >   \+\
> >   \mathcal{O}(L C d)
> >   \=\
> >   \mathcal{O}\bigl(d\(C L^2 + C^2)\bigr),
> > $$
> > where the $C L^2 d$ term from the transformer backbone typically dominates.
> >
> > In contrast, our module first summarizes each historical workout into a single vector via two small BiLSTMs and a GRU, and then performs multi-head attention with a single query (the most recent context) over the resulting $N$ workout-level context vectors. Let $H$ denote the hidden size of the RNNs. The overall complexity of the time-aware attention module is
> > $$
> >   \mathcal{O}\left(\sum_{i=1}^{N} L_i H^2\right)
> >   \+\
> >   \mathcal{O}(N H^2),
> > $$
> > where $L_i$ is the length of the $i$-th historical workout. In our setting $N$ is bounded (10 most recent workouts) and $H$ is modest, so the cross-history attention part scales as $\mathcal{O}(N H^2)$, i.e., linearly in the number of historical sessions and independent of per-workout sequence length once the summaries are computed. Importantly, each workout-level summary vector can be computed once, cached for each user, and reused across future predictions. This explicit history-level cache is not present in Chronos-2 or NormWear, where representations are recomputed from raw inputs for each segment.
> >
> > To complement this theoretical analysis, we measured inference FLOPs and parameter counts on the FitRec dataset. The results are summarized in Table 8.
> >
> > | Model           | FLOPs (G) | Params (M) |
> > |----------------|-----------|-----------|
> > | NormWear       | 0.0802    | 0.5225    |
> > | Chronos        | 0.1130    | 0.4191    |
> > | Ours (Cached)  | 0.0492    | 0.3019    |
> > | Ours (Uncached)| 0.4786    | 0.3019    |
> >
> > Table 8: Comparison of model size and inference FLOPs on the FitRec dataset.
> >
> > The empirical numbers are consistent with the above complexity discussion. Without caching, the FLOPs of Ours (Uncached) are roughly $9.7\times$ higher than Ours (Cached), which matches the fact that re-encoding all 10 historical workouts costs about ten times as much work as encoding only the current one. Once workout-level summaries are cached, Ours (Cached) becomes the most efficient model while using fewer parameters than the Transformer-based baselines.
> >
> > Chronos remains quite efficient even without caching. This behaviour is consistent with its patching mechanism, which significantly shortens the effective sequence length before applying time- and group-attention. The NormWear baseline also achieves competitive efficiency. In principle, its encoder plus temporal aggregation could also benefit from per-workout caching, but we did not explore this option in order to keep the original design.
> >
> > Overall, for the concrete setting of robust heart-rate modeling from multi-session wearable data, our history-aware yet lightweight architecture offers a favorable trade-off between complexity and accuracy. At the same time, we stress that Chronos-2 and NormWear are designed as general-purpose foundation models. Their lack of explicit history caching is not a flaw, but reflects a different design goal aimed at broad applicability.
> >
> > **References**
> >
> > [1] Ansari, Abdul Fatir, et al. ``Chronos-2: From Univariate to Universal Forecasting.'' arXiv preprint, 2025.
> >
> > [2] Luo, Yunfei, et al. ``Toward Foundation Model for Multivariate Wearable Sensing of Physiological Signals.'' arXiv preprint, 2024.

---

### Official Review · Reviewer_xynF · 2025-10-31

**Soundness:** 2
**Presentation:** 3
**Contribution:** 2
**Rating:** 4
**Confidence:** 4

**Summary:**

This paper describes a novel method for representation learning using exercise data from heterogeneous sources (different fitness devices).   The training and downstream analysis uses two different data sets, FitRec and ParroTao— the former is publicly available on Kaggle; it is not clear whether the latter is publicly available.  The authors describe their method in technical detail and have shared sufficient code that it could (probably) be reproduced using another data set.

The method consists of a contrastive learning strategy that is designed to accommodate both subject heterogeneity (individuals can have different physiology, and also engage in systematically different exercise types) and source heterogeneity (devices from different manufacturers have different data streams).  The strategy relies upon ‘
‘Random Feature Dropout’ (to encourage invariance to input streams), a ‘Time-Aware Attention Module’ to integrate information drawn from past exercise sessions. The loss function used for end-to-end training consists of both a reconstruction (MSE) loss for the forecasted heart rate, as well as a contrastive loss applied to the user embeddings.

The authors’ model is evaluated and compared against a large number of baseline models based upon the forecasted HR accuracy (MSE and MAE).  The paper concludes with description of a potential downstream application (route recommendation).

**Strengths:**

## This paper represents strong contributions in the following areas:

### Originality:
The random feature dropout strategy addresses a fundamental challenge in consumer hardware: sensor suite variability across device manufacturers and products.   From the results it is also clear that the reported method produces better performance for HR forecasting compared to existing baseline approaches.

### Clarity
The training objective is clear (contrastive loss applied to subject embedding + MSE loss applied to forecasted HR), well-reasoned, and in line with scientific intuition about physiology and exercise.

### Significance
If publicly available, the ParroTao dataset (42,576 sessions from 113 users; 3 device manufacturers) represents a valuable dataset for related analysis and future comparisons.   I also commend the authors for sharing their code, which enables reproducibility of their findings.

### Quality
The authors’ comparison of their model against a large selection of other well-described baselines (Section D) is a significant strength of the paper.

**Weaknesses:**

# Brief Summary #
This paper has several important weaknesses that are listed briefly below, with more detailed comments and some suggested steps to address each weakness in the section following the list the list:

1. Concerns regarding the data split into train/val/test sets.
2. Concerns regarding use of non-causal input features
3. No evaluation of input feature importance (or conversely, input feature invariance) for the HR forecasting task
4. Minimally detailed evaluation of performance (HR forecasting accuracy) stratified by participant and device type
5. It lacks quantitative analysis supporting the statements made in the abstract (line 27) and section 4.5 (line 426) that “learned representations are highly discriminative across both users and sports”
6. [most concerning] No evaluation of the utility or informativeness of the learned subject representations.
7. Insufficient evaluation of downstream application(s)
8. [Covered in 'Question' section] Abstract includes a vague statement about performance relative to baselines (line 26; also repeated on line 108)
9. [Covered in 'Question' section] Ambiguous data availability (data is stated as publicly released, but with no information about how/where to access)



# Additional Discussion Details #

## Issue 1: Concerns regarding the data split into train/val/test sets.
The authors report this (lines 355-356) as “We split the dataset into 80%, 10%, and 10% for training, validation, and testing” however they provide no comments or details regarding how the split was stratified.  Is the 80/10/10 split based on random selection of # participant’s # (meaning that a participant’s data is present in one and only one of the splits), or is it based on randomly-selected exercise sessions (meaning a participant’s data may be present in multiple splits).  If one participant’s data is present in multiple splits, this is likely to influence the validation/test performance via data leakage.

### Suggestions to address Issue 1:
At a minimum the authors should explicitly state whether the data splits were performed randomly by session, or stratified by participant ID.  Ideally, the data split should be the latter (stratified by participant), but if this was not done then the authors should discuss the downstream impact as a potential limitation of the study.   It is very likely that random splits not stratified by participant will inflate the downstream performance numbers.


## Issue 2:  Concerns regarding use of non-causal input features
Looking at the full list of variables in Section C Table 6 I am concerned that some of these inputs are non-causal in nature. For example VO2max is likely calculated after the completion of the workout session (based on information collected during the session itself).  Therefore if the VO2max value is used for HR forecasting early in the workout session, this may represent the use of future/non-causal information.   I have similar concerns regarding Stress and ‘Body Battery’ (these are likely proprietary manufacturer-specific metrics, so obtaining detailed information about them may not be possible).

### Suggestions to address Issue 2:
I recommend that the authors carefully review the nature of the input variables, and consider removing any that are non-causal from the input streams.  This would also leave
those metrics available to use as prediction targets for additional downstream tasks.



## Issue 3::  No evaluation of input feature importance (or conversely, input feature invariance) for the HR forecasting task or other
Despite the input invariance encouraged in pre-training (random feature dropout), it is likely that some input streams are more informative than others for the HR estimation task.  Do some streams remain more essential than others?  For example running speed and elevation are likely more informative for predicting HR than stance time.

### Suggestions to address Issue 3:
Add some form of feature importance analysis (e.g. ablation study removing specific data streams at inference) to quantify the importance or ‘informativeness’ of each stream.   Consider doing the same for other downstream tasks (if any are added).



## Issue 4  Minimally detailed evaluation of performance (HR forecasting accuracy) stratified by participant and device type
Tables 8 and 9 provide a detailed breakdown of HR forecasting accuracy by exercise type.  However there is no breakdown according to device type (manufacturer).  Additionally, there is no information regarding the distribution of accuracy over participants.  Having performance information stratified by participant and/or device type would be valuable to understand the distribution of algorithm performance.

## Suggestions to address Issue 4:
For a subset of better-performing models (e.g. ‘FitRec’ and ‘Ours’) tabulate the performance stratified by device type (Garmin, Coros, Huawei ).  It may make sense to also stratify these by exercise type (only for the most common exercise, such as running, biking, indoor cycling).

For a subset of better-performing models (e.g. ‘FitRec’ and ‘Ours’) it would be helpful to see the distribution of performance over participants (e.g. histogram of MAE over participants).  This will highlight whether there are outlier subjects for whom the modeling simply does not perform well.



## Issue 5: Lack of quantitative analysis supporting discriminative capability of learned representations
In the abstract (line 27) and section 4.5 (line 426) the authors state that “learned representations are highly discriminative across both users and sports”.   However, the supporting tSNE analysis in Fig. 3 is entirely visual/qualitative, and does not demonstrate (convincingly or quantitatively), that the learned representations effectively separate individuals and activities.  Can this be quantified in some manner?

### Suggestions to address Issue 5:
Consider adding some measure of unsupervised embedding quality such as Rankme (or similar), and/or a participant-level cluster evaluation metric (e.g. KNN accuracy, ARI, NMI or Davies-Bouldin Index)


## Issue 6:[most concerning] No evaluation of the utility or informativeness of the learned subject representations.
The authors state several times that an important characteristic of the user embedding is to capture personal physiologically meaningful information (lines 196, 260).   However, there are no experiments in the paper that quantify that (or even meaningfully test the hypothesis).  It is important to include these to demonstrate that the embeddings contain some relevant information regarding participant physiology, fitness or (ideally) health status, not merely exercise patterns.

### Suggestions to address Issue 6:
Add some analysis that quantifies the physiologically-relevant content of each user embedding.  If demographic information is available in the data set (e.g. participant age, sex or height/weight) those could be used as relevant proxy targets for evaluating physiological content.   An alternative targets that could be used as a proxy for health/fitness is VO2max (only if this is not used as a model input).


## Issue 7:  Insufficient evaluation of downstream application(s)
The choice of route recommendations based on forecasted heart rate does not seem particularly useful.  Assuming a run workout, if the route recommendation/forecast controls for user-chosen factors such as running pace then it seems like the recommendation would boil down to just a terrain-based predictor (hilly courses tend to cause higher heart rates, which I think everyone already knows).  If the route recommendation/forecast # doesn’t # control for user chosen-factors (which can be made in the moment), then it is not very useful because the user can influence the forecast accuracy simply through their choice of run pace.

Additionally, the evaluation metric(s) that would be used to quantify performance for this downstream application (MAE/MSE) are essentially identical to those used for evaluating the primary (upstream) HR forecasting algorithm.

### Suggestions to address Issue 7:
Is it possible to identify some other downstream application that could be useful in a real-world setting?

One example could be offline VO2max estimation for watches that lack this functionality (looking at Section C Table 6 the Huawei watch has this metric, but the Garmin and Coros watches do not).  Providing that estimate to a non-Huawei device user would be genuinely valuable.   Same idea for Effort Pace, Calories / Kilojoules, and Grade-Adjusted Speed (these are each available from one device maker, but not the other two).

A second example could be heart rate imputation for time periods when HR measurements from a device are missing (HR loss is common across many wrist-based HR).  The method described here would be able to impute missing values to produce a better-looking post-workout HR chart.

**Questions:**

## Issue 8:  Vague statement in abstract
In the abstract the authors state “Evaluations on both PARROTAO and the public FitRec dataset show that our model significantly out-performs existing baselines by 17.5% and 14.6%, respectively”.  However, they do not explain anything about the evaluation (e.g. what the numbers 17.% and 14.6% represent).
### Suggestions to address Issue 8:
Expand this sentence to make it clear that the performance comparison is based on HR forecasting MAE (I think this is the case, though I’m not certain).

## Issue 9:  Ambiguous data availability
Is the ParroTao data set publicly available?  The authors make this statement “we have constructed and publicly released PARROTAO—a large-scale, multi-device, multi-activity dataset” (lines 100-101) but do not provide any further information.   I was not able to locate it through google searches.
### Suggestions to address Issue 9:
Include a clear data availability statement at the end of the main section.  Link to the public dataset location if available.

---

> ### Author Response · Authors · 2025-11-20
> **Reply to Reviewer xynF: Part 1**
>
> We sincerely appreciate the valuable feedback provided by the reviewer. We have carefully considered the reviewer's comments and have made the necessary revisions to our manuscript. Below, we would like to address the reviewer's questions and provide additional information:
>
> **Q1. Concerns regarding the data split into train/val/test sets.**
>
> **A1.**
> We apologize for not describing the split strategy clearly in the original manuscript. In all experiments, the 80/10/10 train/validation/test split is applied at the **participant** level: all sessions from a given user are assigned to exactly one split, and no participant appears in more than one of the train, validation, or test sets. This ensures that there is no user-level data leakage across splits. Because users contribute different numbers of workouts, we construct the splits so that **the total number of workouts in each split is approximately in an 8:1:1 ratio for train, validation, and test, respectively**. We have clarified this procedure in the revised manuscript.
>
> **Q2. Concerns regarding use of non-causal input features.**
>
> **A2.**
> We thank the reviewer for raising this important point. Following the suggestion, we performed a systematic audit of all channels in the *ParroTao* dataset.
>
> First, we identified a set of variables that are effectively constant (all zeros) across all sessions, indicating that these metrics were never actually recorded by the contributing devices in our data collection. Concretely, they were removed from the released dataset and from our model inputs.
>
> Second, we isolated a set of composite metrics that vendors compute from heart rate, pace, and other within-session signals, and that are typically presented to users as high-level “wellness” or “performance” scores. All these variables are retained in the released dataset, but they are not used as model inputs.
>
> Table 6 in Appendix C has been updated to add an explicit "Role in *ParroTao*" column indicating which channels are used as inputs, which are derived-only, which are targets, and which are metadata.
>
> We have re-run all experiments on *ParroTao* using this cleaned set of causal input features. The main results and qualitative conclusions are unchanged within the reported confidence intervals.

---

> ### Author Response · Authors · 2025-11-20
> **Reply to Reviewer xynF: Part 2**
>
> **Q3. Evaluation of input feature importance.**
>
> **A3.**
> We analyse input feature importance at two levels: (i) global feature ablations on FitRec and *ParroTao* to see how much performance degrades when individual channels are removed, (ii) sport- and user-level heterogeneity of input channels on both datasets.
>
> **Feature ablation on FitRec.**
> On the FitRec dataset, the full model uses three sequential inputs: distance, altitude, and time elapsed. At inference time, we drop either the distance channel ("w/o distance") or the altitude channel ("w/o altitude"). We report MSE and MAE, averaged per sport and per user on the test set.
>
> Table 2 summarizes the global effect of these ablations. Removing distance leads to a substantial degradation in performance compared to the full model, while removing altitude has a smaller but still clearly detrimental effect.
>
> **Table 2. Global effect of dropping distance or altitude on FitRec.**
> We report test MSE and MAE (mean ± standard deviation over users and sports) for the full model and its ablated variants.
>
> | Model | Metric | Full model | w/o distance | w/o altitude |
> |-------|--------|-----------:|-------------:|-------------:|
> | Ours  | MSE    | **204.63 ± 3.41** | 522.11 ± 5.60 | 284.68 ± 3.96 |
> | Ours  | MAE    | **10.28 ± 0.05** | 17.67 ± 0.09 | 12.21 ± 0.07 |
>
> **Feature ablation on *ParroTao*.**
> On the *ParroTao* dataset, the full model uses 14 sequential input channels. For each feature, we evaluate the model after removing that single channel at inference time on the test set, and we report the resulting MSE and MAE.
>
> As shown in Table 3, dropping most features increases both MSE and MAE, indicating that the model consistently benefits from combining multiple heterogeneous signals. In particular, power, cadence, and speed produce the largest performance degradation when ablated, suggesting that they are globally the most influential features under our normalized representation.
>
> **Table 3. Overall effect of dropping each input feature on *ParroTao*.**
> For each feature, we report the test error of the ablated model ("$MSE_{drop}$" and “$MAE_{drop}$”) and the change relative to the full model (ΔMSE, ΔMAE). Positive Δ indicates worse performance (higher error).
>
> | Feature              | $MSE_{drop}$ |   ΔMSE | $MAE_{drop}$ |  ΔMAE |
> |----------------------|-------------:|-------:|-------------:|------:|
> | speed                |       164.13 |  39.09 |         8.76 |  1.24 |
> | cadence              |       172.30 |  47.25 |         9.09 |  1.56 |
> | power                |       312.26 | 187.22 |        13.89 |  6.36 |
> | stance time          |       127.20 |   2.16 |         7.70 |  0.17 |
> | temperature          |       136.30 |  11.26 |         7.72 |  0.20 |
> | enhanced altitude    |       130.60 |   5.56 |         7.68 |  0.16 |
> | position latitude    |       134.96 |   9.92 |         7.86 |  0.33 |
> | position longitude   |       128.25 |   3.21 |         7.63 |  0.10 |
> | step length          |       137.88 |  12.83 |         8.09 |  0.56 |
> | cycle length 16      |       124.70 |  -0.34 |         7.53 |  0.01 |
> | vertical oscillation |       125.67 |   0.62 |         7.58 |  0.05 |
> | vertical ratio       |       129.95 |   4.90 |         7.82 |  0.30 |
> | distance in meters   |       124.33 |  -0.72 |         7.52 | -0.01 |
> | elevation in meters  |       124.07 |  -0.97 |         7.51 | -0.01 |

---

> ### Author Response · Authors · 2025-11-20
> **Reply to Reviewer xynF: Part 3**
>
> **Heterogeneity of feature importance across sports and users.**
> Beyond these global averages, we also investigate how feature importance varies across sports and users. On FitRec, we compare the relative effect of dropping distance versus altitude at the sport and user levels (Table 4). For $21/25$ sports ($84.0%$), removing distance increases MSE more than removing altitude. In contrast, $3/25$ sports ($12.0%$) are more sensitive to altitude, and one sport ($4.0%$) is largely insensitive to both features. A similar pattern appears across users: for $72/91$ users ($79.1%$), distance is more important, whereas for $16/91$ users ($18.6%$) altitude is more critical, and only $3.3%$ of users are effectively insensitive to both.
>
> **Table 4. Heterogeneity of feature importance across sports and users on FitRec.**
> “Distance > altitude” counts sports/users for which dropping distance increases MSE more than dropping altitude, or where only the distance ablation increases MSE. Percentages are relative to 25 sports and 91 users in the test set.
>
> | Pattern                    | Sports | Users |
> |----------------------------|:------:|:-----:|
> | Distance > altitude        | 21 (84.0 %) | 72 (79.1 %) |
> | Altitude > distance        | 3 (12.0 %)  | 16 (18.6 %) |
> | Neither clearly important  | 1 (4.0 %)   | 3 (3.3 %) |
>
> On *ParroTao*, we summarize the feature importance across sports and users in Table 5. For each feature, we report the average change in MSE when it is dropped and the fraction of sports/users for which the ablation increases MSE. Feature importance is clearly not uniform across all sports and users. Across the 19 users in the test set, power is the most critical feature for the majority of users, but cadence, speed, stance time, temperature, and distance each dominate for at least one user. Overall, these results highlight that, the effective importance of each channel is strongly conditioned on sport type and user-specific characteristics.
>
> **Table 5. Heterogeneity of feature importance across sports and users on *ParroTao*.**
> For each feature, we report the average change in MSE when the feature is dropped (Sport/User mean ΔMSE) and the fraction of sports/users for which the ablation increases MSE (ΔMSE > 0). There are 13 sports and 19 users in the test set.
>
> | Feature               | Sport mean ΔMSE | Sport ΔMSE > 0 | User mean ΔMSE | User ΔMSE > 0 |
> |-----------------------|----------------:|---------------:|---------------:|--------------:|
> | speed                 | 149.1 | 84.6 % |  47.8 | 94.7 % |
> | cadence               | 100.3 | 76.9 % |  88.1 |100.0 % |
> | power                 | 166.4 | 84.6 % | 149.4 | 94.7 % |
> | stance time           |  −3.5 | 38.5 % |   7.4 | 31.6 % |
> | temperature           |   3.5 | 61.5 % |  13.3 | 57.9 % |
> | enhanced altitude     |   5.3 | 53.9 % |  13.8 | 73.7 % |
> | position latitude     |  31.6 | 61.5 % |   7.1 | 78.9 % |
> | position longitude    |   7.8 | 53.9 % |   1.1 | 57.9 % |
> | step length           |  13.7 | 69.2 % |  16.6 | 84.2 % |
> | cycle length 16       |  −0.8 | 38.5 % |   0.1 | 42.1 % |
> | vertical oscillation  |   5.4 | 69.2 % | −0.6 | 36.8 % |
> | vertical ratio        |  −2.3 | 38.5 % |   6.5 | 52.6 % |
> | distance in meters    |  −1.6 | 46.2 % |   8.8 | 52.6 % |
> | elevation in meters   |  −2.9 | 38.5 % | −0.7 | 42.1 % |

---

> ### Author Response · Authors · 2025-11-20
> **Reply to Reviewer xynF: Part 4**
>
> **Q4. Evaluation of performance stratified by participant and device type.**
>
> **A4.**
> We have added a stratified evaluation to better characterize how performance varies across device types, sports, and participants. Since the FitRec dataset does not contain device labels, we stratify it only by sport and participant. In contrast, the *ParroTao* dataset includes three device types (Huawei, Coros, Garmin), which allows us to analyse device- and sport-specific performance.
>
> Table 6 reports macro-averaged MSE and MAE over all sports on *ParroTao*, stratified by device type for the two best-performing models (FitRec and Ours). Our model consistently improves over the FitRec baseline across all three device types in both metrics.
>
> **Table 6. Device-stratified performance on the *ParroTao* dataset for the FitRec baseline and our model.**
> Values are macro-averaged test MSE and MAE over all sport IDs for each device (lower is better ↓).
>
> | Device | FitRec MSE ↓ | FitRec MAE ↓ | Ours MSE ↓ | Ours MAE ↓ |
> |--------|--------------:|-------------:|-----------:|-----------:|
> | Huawei | 148.44 | 8.18 | **120.31** | **7.28** |
> | Coros  | 142.48 | 8.40 | **132.71** | **7.83** |
> | Garmin | 113.54 | 7.50 | **90.19** | **6.28** |
>
>
> To examine variability across participants, Figures in appendix P show density plots of per-user test MAE for the FitRec baseline and our model on the FitRec and *ParroTao* datasets, respectively. For each dataset, we compute the per-participant MAE for both models and visualise the resulting distributions using normalised histograms with overlaid kernel density estimates. Across almost all users, our model either matches or improves upon the FitRec baseline, while the few extreme outliers in the right tail correspond to participants with very few recorded workouts, for whom both models exhibit higher error due to limited data.
>
> Finally, to study the interaction between device type and exercise type on *ParroTao*, we report device- and sport-stratified MSEs for four common sports that are recorded by all three devices in Table 7. For most device--sport combinations, our model achieves lower error than the FitRec baseline.
>
> **Table 7. Device- and sport-stratified test MSE on the *ParroTao* dataset for the FitRec baseline and our model.**
> We report results for four sports (sport IDs 0, 1, 2, and 4) that are present on all three device types.
>
> | Device | Sport Type | FitRec MSE | Ours MSE |
> |--------|----------------------|-----------:|-----------:|
> | Huawei | Running               | 115.10 | **95.69** |
> | Huawei | Cross-Country Run     | 149.62 | 167.98 |
> | Huawei | Cycling               | 316.84 | **238.11** |
> | Huawei | Treadmill Running     |  56.83 | **24.89** |
> | Coros  | Running               | 121.16 | **108.89** |
> | Coros  | Cross-Country Run     | 211.65 | **210.29** |
> | Coros  | Cycling               | **322.28** | 344.07 |
> | Coros  | Treadmill Running     | **288.09** | 392.96 |
> | Garmin | Running               | 104.53 | **80.14** |
> | Garmin | Cross-Country Run     | 335.75 | **312.56** |
> | Garmin | Cycling               | 216.24 | **211.05** |
> | Garmin | Treadmill Running     | **442.44** | 588.50 |

---

> ### Author Response · Authors · 2025-11-20
> **Reply to Reviewer xynF: Part 5**
>
> **Q5. Quantitative analysis supporting discriminative capability of learned representations.**
>
> **A5.**
> To quantitatively assess how well the learned representations separate different labels, we treat the latent vectors from the FitRec baseline and our model as embeddings and evaluate a battery of standard structure-aware metrics on both FitRec and *ParroTao*. For each representation and each label space, we measure (i) local label recovery via $1$- and $5$-NN classification accuracy (kNN@1, kNN@5), (ii) label--representation alignment via Adjusted Rand Index (ARI), Normalized Mutual Information (NMI), homogeneity, completeness, and V-measure, and (iii) cluster quality via Davies--Bouldin Index (DBI, lower is better), average silhouette width (ASW), Calinski--Harabasz score (CH, higher is better), and a global rank-based score (RankMe). We consider three label types: **sport type**, **user identity**, and the combined **user$\times$sport** label. The full results are reported in Tables 8-10.
>
> Across all label granularities and on both datasets, our model consistently achieves better kNN accuracy, stronger label–embedding alignment, and more compact, better-separated clusters than the FitRec baseline. For sport labels, the gains are modest but systematic. For user identity and user$\times$sport labels, the improvements are substantially larger despite the much higher number of classes, indicating that our embeddings differentiate individuals particularly well. This provides quantitative evidence that the proposed architecture learns representations that more cleanly organize sessions by both sport type and user-specific characteristics, which is crucial for downstream personalization and analysis.
>
> **Table 8. Representation quality with respect to sport type labels.**
> Higher is better for RankMe, kNN, ARI, NMI, ASW, CH, homogeneity, completeness, and V-measure. Lower is better for DBI.
>
> | Dataset | Model | RankMe | kNN@1 | kNN@5 | ARI | NMI | DBI ↓ | ASW | CH | Hom. | Compl. | V-meas. |
> |---------|-------|-------:|------:|------:|----:|----:|------:|----:|----:|-----:|-------:|--------:|
> | FitRec | FitRec | 40.91 | 0.931 | 0.920 | 0.043 | 0.110 | 1.924 | 0.089 | 2259.19 | 0.193 | 0.077 | 0.110 |
> | FitRec | **Ours** | **61.17** | **0.945** | **0.934** | **0.051** | **0.140** | **1.508** | **0.153** | **3595.47** | **0.245** | **0.098** | **0.140** |
> | *ParroTao* | FitRec | 37.81 | 0.929 | **0.934** | 0.020 | 0.084 | 1.628 | **0.204** | 967.65 | 0.234 | 0.051 | 0.084 |
> | *ParroTao* | **Ours** | **52.33** | **0.934** | 0.926 | **0.024** | **0.089** | **1.420** | 0.168 | **1136.27** | **0.243** | **0.054** | **0.089** |
>
> **Table 9. Representation quality with respect to user labels.**
> Higher is better for RankMe, kNN, ARI, NMI, ASW, CH, homogeneity, completeness, and V-measure. Lower is better for DBI.
>
> | Dataset | Model | RankMe | kNN@1 | kNN@5 | ARI | NMI | DBI ↓ | ASW | CH | Hom. | Compl. | V-meas. |
> |---------|-------|-------:|------:|------:|----:|----:|------:|----:|----:|-----:|-------:|--------:|
> | FitRec  | FitRec | 40.91 | 0.495 | 0.427 | 0.077 | 0.164 | 1.714 | 0.078 | 164.33  | 0.174 | 0.155 | 0.164 |
> | FitRec  | **Ours** | **61.17** | **0.713** | **0.681** | **0.091** | **0.617** | **1.462** | **0.116** | **872.16** | **0.616** | **0.618** | **0.617** |
> | *ParroTao* | FitRec | 37.81 | 0.369 | 0.319 | **0.097** | 0.170 | 1.614 | 0.149 | 371.51 | 0.181 | 0.160 | 0.170 |
> | *ParroTao* | **Ours** | **52.33** | **0.697** | **0.669** | 0.054 | **0.585** | **1.450** | **0.203** | **792.76** | **0.579** | **0.592** | **0.585** |
>
> **Table 10. Representation quality with respect to joint user×sport labels.**
> Higher is better for RankMe, kNN, ARI, NMI, ASW, CH, homogeneity, completeness, and V-measure. Lower is better for DBI.
>
> | Dataset | Model | RankMe | kNN@1 | kNN@5 | ARI | NMI | DBI ↓ | ASW | CH | Hom. | Compl. | V-meas. |
> |---------|-------|-------:|------:|------:|----:|----:|------:|----:|----:|-----:|-------:|--------:|
> | FitRec  | FitRec | 40.91 | 0.367 | 0.318 | 0.082 | 0.313 | 1.699 | 0.084 | 116.68 | 0.373 | 0.270 | 0.313 |
> | FitRec  | **Ours** | **61.17** | **0.492** | **0.425** | **0.115** | **0.694** | **1.525** | **0.136** | **377.73** | **0.698** | **0.689** | **0.694** |
> | *ParroTao* | FitRec | 37.81 | 0.437 | 0.383 | **0.088** | 0.326 | 1.570 | 0.116 | 276.04 | 0.388 | 0.280 | 0.326 |
> | *ParroTao* | **Ours** | **52.33** | **0.697** | **0.658** | 0.065 | **0.659** | **1.323** | **0.173** | **376.82** | **0.656** | **0.661** | **0.659** |

---

> ### Author Response · Authors · 2025-11-20
> **Reply to Reviewer xynF: Part 6**
>
> **Q6. Evaluation of the utility or informativeness of the learned subject representations.**
>
> **A6.**
> To assess whether our learned user representations encode meaningful subject-specific information, we perform a downstream gender prediction task on both the FitRec and *ParroTao* datasets. Concretely, we freeze the user embeddings produced by either the FitRec baseline or our model, and train a lightweight classifier on top of these embeddings to predict binary gender. We report accuracy and F1-score on held-out test sets.
>
> As shown in Table 11, both sets of embeddings support strong gender classification, confirming that user identity information is indeed present in the learned representations. Moreover, our model consistently achieves higher accuracy and F1-score on both datasets, indicating that it produces more informative and discriminative user embeddings. Together with the downstream tasks in A7, where we successfully predict user-related physiological quantities, these results provide converging evidence that the proposed framework captures stable, user-specific characteristics that are useful for personalization.
>
> **Table 11. Gender prediction from learned user representations on *ParroTao* and FitRec.**
> We report accuracy and F1-score on held-out test sets.
>
> | Dataset      | Model  | Accuracy ↑ | F1-score ↑ |
> |--------------|--------|-----------:|-----------:|
> | *ParroTao*   | FitRec | 0.8959 | 0.8949 |
> | *ParroTao*   | **Ours** | **0.9765** | **0.9881** |
> | FitRec       | FitRec | 0.9346 | 0.9666 |
> | FitRec       | **Ours** | **0.9580** | **0.9785** |
>
> **Q7. More evaluation of downstream applications.**
>
> **A7.**
> We have expanded the downstream evaluation beyond route recommendation to further demonstrate the practical utility of the learned representations and the proposed architecture. In addition to the original route-planning scenario, we now consider two additional families of tasks: cross-device feature imputation and heart-rate imputation under missing segments.
>
> **Cross-device feature imputation on Huawei devices.**
> We first conduct a cross-device experiment on the *ParroTao* subset collected using Huawei wearables, mimicking a deployment scenario where a target device does not expose certain sensor channels that are available on other devices. We train models on Huawei workouts to reconstruct three Huawei-specific targets (calories, stance time, vertical oscillation) from the remaining inputs, and evaluate on a held-out test split.
>
> Table 12 compares the FitRec baseline with our model in terms of MSE and MAE. Across all three targets, our approach substantially reduces both error metrics, showing that it can serve as an effective cross-device imputation module for recovering missing sensor channels.
>
> **Table 12. Cross-device feature imputation on the *ParroTao*–Huawei subset.**
> We predict three target channels (calories, stance time, vertical oscillation) from the remaining inputs.
> For each metric we report mean ± standard deviation. Lower is better for both MSE and MAE.
>
> | Target | Model | MSE ↓ | MAE ↓ |
> |--------|-------|-------|-------|
> | Calories | FitRec | 0.03408 ± 1.20×10 E−4 | 0.12439 ± 6.80×10 E−6 |
> |  | **Ours** | **0.01275 ± 3.80×10 E−7** | **0.01607 ± 1.84×10 E−5** |
> | Stance time | FitRec | 0.00929 ± 2.79×10 E−6 | 0.09491 ± 2.22×10 E−5 |
> |  | **Ours** | **3.78×10 E−4 ± 3.63×10 E−7** | **0.01284 ± 1.53×10 E−5** |
> | Vertical oscillation | FitRec | 0.00394 ± 4.59×10 E−7 | 0.05856 ± 2.65×10 E−6 |
> |  | **Ours** | **8.29×10 E−4 ± 8.20×10 E−7** | **0.02495 ± 1.70×10 E−5** |

---

> ### Author Response · Authors · 2025-11-20
> **Reply to Reviewer xynF: Part 7**
>
> **Cross-device feature imputation on Garmin devices.**
> We further evaluate cross-device imputation on the *ParroTao* subset collected with Garmin wearables. Here we reconstruct three Garmin-specific targets (body battery, performance condition, vertical ratio) from the remaining inputs, again training on a subset and evaluating on a held-out test split.
>
> As summarised in Table 13, our model is competitive or superior to the FitRec baseline across these targets, confirming that the proposed architecture can generalise as a flexible imputation module for different device-specific channels.
>
> **Table 13. Cross-device feature imputation on the *ParroTao*–Garmin subset.**
> We predict three target channels (body battery, performance condition, vertical ratio) from the remaining inputs.
> For each metric we report mean ± standard deviation. Lower is better for both MSE and MAE.
>
> | Target | Model | MSE ↓ | MAE ↓ |
> |--------|-------|-------|-------|
> | body_battery | FitRec | 0.00518 ± 0.00235 | 0.02208 ± 0.00146 |
> |             | **Ours** | **7.44 × 10 E−4 ± 8.62 × 10 E−4** | **0.00143 ± 5.26 × 10 E−4** |
> | performance_condition | FitRec | 1.83 × 10 E−5 ± 6.35 × 10 E−6 | 2.34 × 10 E−3 ± 7.61 × 10 E−5 |
> |                       | **Ours** | **1.85 × 10 E−5 ± 2.14 × 10 E−5** | **8.23 × 10 E−4 ± 8.11 × 10 E−5** |
> | vertical_ratio | FitRec | 0.0837 ± 0.0562 | 0.0147 ± 0.00629 |
> |                | **Ours** | **0.0574 ± 0.0492** | **0.0187 ± 0.00493** |
>
> **Heart-rate imputation.**
> Finally, we evaluate whether the framework can also support imputing missing heart-rate (HR) values, a common issue for wrist-based sensors. We construct masked variants of the FitRec and *ParroTao* datasets (denoted \texttt{fitrec-mask} and \texttt{parrotao-mask}) by randomly masking short HR segments so that roughly $20%$ of time steps are missing in each sequence. Given the original covariates, the partially masked HR sequence, and a binary mask indicating missing positions, each model is trained to reconstruct the full HR trajectory, with the loss computed only on masked time steps.
>
> We compare our model against four baselines: (i) the FitRec model, (ii) a Kalman filter, (iii) linear interpolation, and (iv) last-observation-carried-forward (LOCF). Table 14 reports MSE on masked positions for both datasets. Our method achieves the lowest error across all settings, substantially outperforming classical interpolation and filtering methods as well as the FitRec baseline. This shows that the learned representations are not only useful for forecasting, but also for high-quality HR imputation, enabling smoother and more informative post-workout HR profiles.
>
> **Table 14. Heart-rate imputation on masked HR sequences (FitRec and *ParroTao*).**
> We report mean ± standard deviation of MSE over masked positions. Lower is better.
>
> | Dataset | Method | MSE ↓ |
> |---------|--------|-------|
> | FitRec | FitRec | 69.78 ± 0.33 |
> | | Kalman | 832.61 ± 1.77 |
> | | Linear interpolation | 91.11 ± 0.15 |
> | | LOCF | 174.00 ± 0.44 |
> | | **Ours** | **40.84 ± 0.17** |
> | *ParroTao* | FitRec | 19.72 ± 0.66 |
> | | Kalman | 167.41 ± 0.48 |
> | | Linear interpolation | 34.81 ± 0.15 |
> | | LOCF | 81.92 ± 0.64 |
> | | **Ours** | **7.54 ± 0.08** |

---

> ### Author Response · Authors · 2025-11-20
> **Reply to Reviewer xynF: Part 8**
>
> **Q8. Vague statement in abstract.**
>
> **A8.**
> We thank the reviewer for pointing out this ambiguity. In the original abstract, the reported improvements of 17.5% and 14.6% referred to relative reductions in heart-rate forecasting MSE on the FitRec and *ParroTao* datasets compared to the best-performing baseline. In the revised manuscript, we now state this explicitly by clarifying that these percentages quantify the reduction in **HR forecasting MSE** over the strongest baseline on each dataset. In addition, after updating the input feature set for *ParroTao* to exclude potentially non-causal variables and re-running all experiments, the corresponding improvement was revised from 14.6% to 10.4% in both the abstract and the introduction.
>
> **Q9. Data availability clarification.**
>
> **A9.**
> We appreciate the reviewer’s comment and apologize for the ambiguity. At this stage, we do not provide a publicly searchable link to *ParroTao* in order to preserve double-blind anonymity for ICLR. Because ICLR reviews will remain publicly visible, we would like to state explicitly that, **if the paper is accepted, we will immediately release the dataset and provide a permanent access link in the camera-ready version and the associated project page.**

---

> ### Author Response · Authors · 2025-11-28
>
> Dear Reviewer xynF,
>
> Thank you again for your thoughtful and constructive comments. We have carried out additional experiments and analyses to address your points in detail, and we have summarized these results in our responses above.
>
> As the discussion period is nearing its end, we would like to kindly ask whether you have any **remaining concerns or suggestions** that you would like us to address. If everything now looks satisfactory, we would also be grateful if you could briefly indicate this in the discussion.
>
> We sincerely appreciate your time and effort in reviewing our paper and your valuable feedback in helping us improve this work.

---

### Official Review · Reviewer_gRNk · 2025-11-01

**Soundness:** 3
**Presentation:** 3
**Contribution:** 2
**Rating:** 6
**Confidence:** 2

**Summary:**

This paper proposes a framework for heart rate prediction that learns data representations robust to source and user heterogeneity. The proposed method claims it explicitly models the fragmented nature of data from wearable devices and individual physiological variance, which are common challenges in real-world deployment.

**Strengths:**

The work presents a formulation of the data heterogeneity problem in heart rate modeling, separating it into source and user dimensions. The proposed method to address these challenges is reasonable. The introduction of random feature dropout to handle source heterogeneity and the use of a time-aware attention module and contrastive learning to manage user heterogeneity is grounded in existing representation learning techniques. A contribution is the creation and release of the PARROTAO dataset, which captures multi-device and multi-activity variations.

**Weaknesses:**

The rationale for certain architectural choices could be further substantiated. For instance, the selection of BiLSTMs and GRUs within the time-aware attention module is presented without a comparison to alternative sequential models. While random feature dropout is intended to create robustness to missing features, its interaction with features that have different units or scales across devices is not explored. The paper states that unobserved features are set to zero, but the impact of this imputation choice is not discussed. The contrastive loss is applied using either user ID or activity type as labels, but the effect of this choice is not analyzed.

**Questions:**

The PARROTAO dataset is one of the main contribution, but the paper provides limited detail on the demographics of the 113 athletes (e.g., age, fitness level distribution). How might the specific characteristics of this cohort affect the generalizability of a model trained on it, particularly to populations like older adults or individuals with different health conditions?

For the route recommendation application, how were the planned route features, such as topographical data and intended pace, acquired to be fed into the model for prediction?

The time-aware attention module considers the 10 most recent workouts. Was a sensitivity analysis performed on this number? It seems plausible that for some users or activities, a longer or shorter history might be more predictive. Furthermore, how does the model handle new users with fewer than 10 historical sessions?

---

> ### Author Response · Authors · 2025-11-20
> **Reply to Reviewer gRNk: Part 1**
>
> We sincerely appreciate the valuable feedback provided by the reviewer. We have carefully considered the reviewer's comments and have made the necessary revisions to our manuscript. Below, we would like to address the reviewer's questions and provide additional information:
>
> **Q0. Potential Ethical Concern.**
>
> **A0.**
> All wearable data in this study were contributed voluntarily by users who provided written informed consent. The consent form specified what signals would be collected, how they would be anonymized and stored, and participants’ rights to inspect or withdraw their data at any time. No direct personal identifiers or communication records were collected. Device and account IDs were replaced by anonymized codes, and access to raw logs was limited to authorized project members under strict access control.
>
> The data are used solely for non-commercial research. Raw device-level logs are deleted after project completion, and only de-identified aggregate statistics are retained for reproducibility. All procedures were designed to comply with established human-subjects research principles and applicable data-protection regulations.
>
> **Q1. What is the rationale for certain architectural choices?**
>
> **A1.**
> To assess the impact of our architectural choices in the time-aware attention module, we conducted additional experiments in which (i) we kept the GRU fixed and replaced the BiLSTM with a GRU, a unidirectional LSTM, or a
> Transformer encoder, and (ii) we kept the BiLSTM fixed and replaced the GRU with a BiLSTM, a unidirectional LSTM, or a
> Transformer encoder. We evaluated all variants on the FitRec and ParroTao datasets, and report the results in Table 1.
> Across both datasets, these substitutions lead to only small fluctuations in MSE and MAE, suggesting that the
> performance gains primarily come from the overall design of the time-aware attention module rather than from any
> particular choice of recurrent encoder.
>
> **Table 1. Encoder choice in the time-aware attention module.
> We report mean ± std of MSE and MAE on the FitRec and
> ParroTao**
>
> | Dataset      | Modification         | MSE ↓             | MAE ↓            |
> |--------------|----------------------|-------------------|------------------|
> | **FitRec**   | Original             | **204.63 ± 3.41** | 10.28 ± 0.05     |
> |              | BiLSTM → GRU         | 248.84 ± 3.16     | 11.78 ± 0.06     |
> |              | BiLSTM → LSTM        | 206.20 ± 4.09     | **10.27 ± 0.06** |
> |              | BiLSTM → Transformer | 206.30 ± 3.48     | 10.30 ± 0.06     |
> |              | GRU → BiLSTM         | 212.42 ± 3.49     | 10.56 ± 0.06     |
> |              | GRU → LSTM           | 210.42 ± 2.88     | 10.63 ± 0.06     |
> |              | GRU → Transformer    | 208.89 ± 3.50     | 10.34 ± 0.06     |
> | **ParroTao** | Original             | 125.04 ± 4.57     | **7.53 ± 0.11**  |
> |              | BiLSTM → GRU         | **124.01 ± 4.49** | 7.63 ± 0.11      |
> |              | BiLSTM → LSTM        | 128.53 ± 4.72     | 7.80 ± 0.12      |
> |              | BiLSTM → Transformer | 127.06 ± 4.84     | 7.77 ± 0.11      |
> |              | GRU → BiLSTM         | 125.57 ± 4.79     | 7.65 ± 0.11      |
> |              | GRU → LSTM           | 125.29 ± 4.86     | 7.54 ± 0.12      |
> |              | GRU → Transformer    | 127.71 ± 4.80     | 7.79 ± 0.11      |

---

> ### Author Response · Authors · 2025-11-20
> **Reply to Reviewer gRNk: Part 2**
>
> **Q2. How does random feature dropout interact with heterogeneous features with different units and scales?**
>
> **A2.**
> All input channels from different devices are first standardized using z-score normalization. Random feature dropout is
> therefore applied in a **common, unitless feature space**, which mitigates direct dependence on the original physical
> units and scales. To further examine how dropout interacts with heterogeneous features, we performed **additional
> feature ablation studies** on both the FitRec and ParroTao datasets at inference time: we masked each individual
> feature channel and quantified the resulting degradation in prediction performance (MSE and MAE) as a proxy for feature
> importance.
>
> **Feature ablation on FitRec.**
> On the FitRec dataset, the full model uses three sequential inputs: distance, altitude, and time elapsed. At inference
> time, we drop either the distance channel ("w/o distance") or the altitude channel ("w/o altitude"). We report MSE and
> MAE, averaged per sport and per user on the test set.
>
> Table 2 summarizes the global effect of these ablations. Removing distance leads to a substantial degradation in
> performance compared to the full model, while removing altitude has a smaller but still clearly detrimental effect.
>
> **Table 2. Global effect of dropping distance or altitude on FitRec.
> We report test MSE and MAE (mean ± standard deviation over users and sports)
> for the full model and its ablated variants.**
>
> | Model | Metric | Full model            | w/o distance  | w/o altitude  |
> |-------|--------|-----------------------|---------------|---------------|
> | Ours  | MSE    | **204.63** ± **3.41** | 522.11 ± 5.60 | 284.68 ± 3.96 |
> | Ours  | MAE    | **10.28** ± **0.05**  | 17.67 ± 0.09  | 12.21 ± 0.07  |
>
> **Feature ablation on ParroTao.**
> On the ParroTao dataset, the full model uses 14 sequential input channels. For each feature, we evaluate the model after
> removing that single channel at inference time on the test set, and we report the resulting MSE and MAE.
>
> As shown in Table 3, dropping most features increases both MSE and MAE, indicating that the model consistently benefits
> from combining multiple heterogeneous signals. In particular, power, cadence, and speed produce the largest performance
> degradation when ablated, suggesting that they are globally the most influential features under our normalized
> representation.
>
> **Table 3. Overall effect of dropping each input feature on ParroTao**
>
> | Feature              | MSE$_{drop}$ | $Δ$ MSE | MAE$_{drop}$ | $Δ$ MAE |
> |----------------------|--------------|---------|--------------|---------|
> | speed                | 164.13       | 39.09   | 8.76         | 1.24    |
> | cadence              | 172.30       | 47.25   | 9.09         | 1.56    |
> | power                | 312.26       | 187.22  | 13.89        | 6.36    |
> | stance time          | 127.20       | 2.16    | 7.70         | 0.17    |
> | temperature          | 136.30       | 11.26   | 7.72         | 0.20    |
> | enhanced altitude    | 130.60       | 5.56    | 7.68         | 0.16    |
> | position latitude    | 134.96       | 9.92    | 7.86         | 0.33    |
> | position longitude   | 128.25       | 3.21    | 7.63         | 0.10    |
> | step length          | 137.88       | 12.83   | 8.09         | 0.56    |
> | cycle length 16      | 124.70       | -0.34   | 7.53         | 0.01    |
> | vertical oscillation | 125.67       | 0.62    | 7.58         | 0.05    |
> | vertical ratio       | 129.95       | 4.90    | 7.82         | 0.30    |
> | distance in meters   | 124.33       | -0.72   | 7.52         | -0.01   |
> | elevation in meters  | 124.07       | -0.97   | 7.51         | -0.01   |

---

> ### Author Response · Authors · 2025-11-20
> **Reply to Reviewer gRNk: Part 3**
>
> **Heterogeneity of feature importance across sports and users.**
> Beyond these global averages, we also investigate how feature importance varies across sports and users. On FitRec, we
> compare the relative effect of dropping distance versus altitude at the sport and user levels (Table 4). For $21/25$
> sports ($84.0$\%), removing distance increases MSE more than removing altitude. In contrast, $3/25$ sports ($12.0$\%)
> are more sensitive to altitude, and one sport ($4.0$\%) is insensitive to both features. A similar pattern appears
> across users: for $72/91$ users ($79.1$\%), distance is more important, whereas for $16/91$ users ($18.6$\%) altitude is
> more critical, and only $3.3$\% of users are relatively insensitive to both.
>
> **Table 4. Heterogeneity of feature importance across sports and users on FitRec. "Distance $>$ altitude" counts
> sports/users for which dropping distance increases MSE more than dropping altitude, or where only the distance ablation
> increases MSE. Percentages are relative to 25 sports and 91 users on the test set.**
>
> | Pattern                   | Sports          | Users           |
> |---------------------------|-----------------|-----------------|
> | Distance $>$ altitude     | $21$ ($84.0$\%) | $72$ ($79.1$\%) |
> | Altitude $>$ distance     | $3$ ($12.0$\%)  | $16$ ($18.6$\%) |
> | Neither clearly important | $1$ ($4.0$\%)   | $3$ ($3.3$\%)   |
>
> On ParroTao, we summarize the feature importance across sports and users in Table 5. For each feature, we report the
> average change in MSE when it is dropped and the fraction of sports/users for which the ablation increases MSE. Feature
> importance is clearly not uniform across all sports and users. Across the 19 users in the test set, power is the most
> critical feature for the majority of users, but cadence, speed, stance time, temperature, and distance each dominate for
> at least one user. Overall, these results highlight that, the effective importance of each channel is strongly
> conditioned on sport type and user-specific characteristics.
>
> **Table 5. Heterogeneity of feature importance across sports and users on ParroTao**
>
> | Feature              | Sport mean $Δ$MSE | Sport $Δ$MSE$>0$ | User mean $Δ$MSE | User $Δ$MSE$>0$ |
> |----------------------|-------------------|------------------|------------------|-----------------|
> | speed                | 149.1             | 84.6\%           | 47.8             | 94.7\%          |
> | cadence              | 100.3             | 76.9\%           | 88.1             | 100.0\%         |
> | power                | 166.4             | 84.6\%           | 149.4            | 94.7\%          |
> | stance time          | -3.5              | 38.5\%           | 7.4              | 31.6\%          |
> | temperature          | 3.5               | 61.5\%           | 13.3             | 57.9\%          |
> | enhanced altitude    | 5.3               | 53.9\%           | 13.8             | 73.7\%          |
> | position latitude    | 31.6              | 61.5\%           | 7.1              | 78.9\%          |
> | position longitude   | 7.8               | 53.9\%           | 1.1              | 57.9\%          |
> | step length          | 13.7              | 69.2\%           | 16.6             | 84.2\%          |
> | cycle length 16      | -0.8              | 38.5\%           | 0.1              | 42.1\%          |
> | vertical oscillation | 5.4               | 69.2\%           | -0.6             | 36.8\%          |
> | vertical ratio       | -2.3              | 38.5\%           | 6.5              | 52.6\%          |
> | distance in meters   | -1.6              | 46.2\%           | 8.8              | 52.6\%          |
> | elevation in meters  | -2.9              | 38.5\%           | -0.7             | 42.1\%          |

---

> ### Author Response · Authors · 2025-11-20
> **Reply to Reviewer gRNk: Part 4**
>
> **Q3. How does zero-imputation affect model behavior compared with alternative imputation strategies?**
>
> **A3.**
> We conducted additional experiments in which unobserved features were imputed using the feature-wise mean, median, or
> majority value estimated from the training, validation, and test sets, and we compared these variants with our original
> zero-imputation scheme. The result is shown in Table 6. Zero-Imputation consistently yields better predictive performance than these three alternatives. A key reason is that some features are only observed for a relatively small subset of devices or activities. Filling the missing entries of such sparse features with their mean, median, or majority value induces a noticeable distribution shift between genuinely observed and imputed values.
>
> **Table 6. Heart-rate forecasting performance (MSE and MAE) on the *ParroTao* dataset under different imputation strategies.**
>
> | Imputation method | MSE ↓ | MAE ↓ |
> |-------------------|---------------|--------------|
> | Avg              | 429.95 ± 10.17 | 16.34 ± 0.18 |
> | Majority         | 400.15 ± 10.83 | 15.48 ± 0.20 |
> | Median           | 389.15 ± 10.78 | 15.15 ± 0.19 |
> | Zero-Imputation  | **125.04 ± 4.57** | **7.53 ± 0.11** |
>
> **Q4. What is the effect of using user IDs versus activity types as contrastive labels?**
>
> **A4.**
> We have compared three variants: using only user IDs as labels, using only activity types as labels, and using both user IDs and activity types jointly (our original implementation). The experimental results are presented in Table 7. The joint-label setting consistently achieves the best performance on both the FitRec and ParroTao dataset. Intuitively, user-ID supervision encourages the encoder to capture user-specific preferences and physiological patterns across sessions, while activity-type supervision promotes better separation between different sports. Combining both sources of supervision therefore more accurate and robust downstream predictions.
>
> **Table 7. Ablation study of contrastive labels on FitRec and ParroTao datasets.**
>
> | Dataset   | Contrastive labels        | MSE (mean ± std)        | MAE (mean ± std)        |
> | ---       | ---                       | ---                      | ---                      |
> | FitRec    | User ID only              | 211.16 ± 3.44            | 10.50 ± 0.05            |
> |           | Sport type only           | 227.79 ± 3.26            | 11.10 ± 0.06            |
> |           | User ID and Sport type    | **204.63 ± 3.41**        | **10.28 ± 0.05**        |
> | ParroTao  | User ID only              | 128.62 ± 4.72            | 7.79 ± 0.12             |
> |           | Sport type only           | 136.79 ± 4.98            | 8.19 ± 0.11             |
> |           | User ID and Sport type    | **125.04 ± 4.57**        | **7.53 ± 0.11**         |

---

> ### Author Response · Authors · 2025-11-20
> **Reply to Reviewer gRNk: Part 5**
>
> **Q5. What are the demographic and fitness characteristics of the ParroTao cohort, and how might these attributes influence the model’s generalizability to broader populations?**
>
> **A5.**
> The demographic and fitness characteristics of the ParroTao cohort are detailed in Table 8. The dataset comprises 113 athletes, with comprehensive metadata on their age, gender, and fitness level.
>
> In a significant enhancement over previous datasets like FitRec, which include only gender and sport type, our dataset incorporates detailed age and fitness level metadata. However, the ParroTao cohort is still primarily composed of young and physically active individuals. Consequently, the model's generalizability to broader populations, such as elderly users or patients, has not yet been established due to current data constraints.
>
> **Table 8. Demographic and Fitness Characteristics of the ParroTao Dataset.**
>
> | Characteristic                 | Count (Athletes) | Percentage |
> | ---                           | ---              | ---        |
> | Age Distribution              |                  |            |
> |  18–20 years                | 30               | 26.5%      |
> |  21–23 years                | 37               | 32.7%      |
> |  24–26 years                | 26               | 23.0%      |
> |  27–30 years                | 20               | 17.7%      |
> | Gender Distribution           |                  |            |
> |  Male                       | 79               | 69.9%      |
> |  Female                     | 34               | 30.1%      |
> | Fitness Level Distribution    |                  |            |
> |  Amateur                    | 52               | 46.0%      |
> |  Semi-Professional          | 39               | 34.5%      |
> |  Professional               | 22               | 19.5%      |
> | Total Athletes                | 113              | 100.0%     |
>
>
> **Q6. For the route recommendation application, how were the planned route features obtained?**
>
> **A6.**
> In the route recommendation application, the model uses only *planned* information that is available before the training
> session. Concretely, we construct three types of planned features for each candidate route: (i) route geometry and
> topography, (ii) environmental context at the planned training time, and (iii) the planned pace profile.
>
> For the *route geometry and topography*, we start from a candidate route chosen by the coach (e.g., a polyline on a
> digital map or an exported GPX route). We sample a sequence of latitude–longitude points along this route at fixed
> distance intervals and attach elevation to each point, thus obtaining the planned topographical profile.
>
> For the *environmental context*, we retrieve the forecast temperature (and, when available, related variables such as
> humidity) for the training location and time from standard weather services, and align these values with the sampled
> time steps.
>
> For the *planned pace profile*, the coach (or athlete) specifies the intended pace as a schedule over the route (e.g.,
> desired splits or segment-wise target paces). We interpolate this schedule and discretize it into a planned speed
> sequence that is time-aligned with the route samples. These planned route, context, and pace features are then fed into
> the model to predict the heart-rate trajectory for each candidate route, which allows the coach to compare the expected
> training load across alternatives.

---

> ### Author Response · Authors · 2025-11-20
> **Reply to Reviewer gRNk: Part 6**
>
> **Q7. What is the effect of the number of historical workouts provided to the time-aware attention module across
> different users or sport types, and how does the model handle cases where a user has fewer than 10 past sessions?**
>
> **A7.**
> We evaluate history
> lengths $K ∈ \{1,5,10,20,30\} $ on the FitRec dataset and report overall MSE and MAE in Table 9. Moving from $K{=}1$ to $
> K{=}5$ has almost no effect on the aggregate error. In contrast, increasing the history length to $
> K{=}10$ reduces MSE by about $3.3\%$ and MAE by $0.6\%$ compared to $K{=}5$. Using $
> K{=}20$ workouts yields a very similar performance (slightly lower MSE and MAE), while $
> K{=}30$ further improves MAE but starts to increase MSE again. Overall, the range $K≈10$ - $
> 20$ appears to be near-optimal at the dataset level, and $
> K{=}10$ offers a convenient operating point that clearly outperforms short histories ($
> K{=}1,5$) while avoiding the additional computational cost of $K{=}20$ or $30$.
>
> To examine heterogeneity across sports and users, Table 10 reports, for each $K$, how many sports and users achieve their
> lowest per-entity MSE at that history length. At the sport level, $8/25$ sports reach their best performance with a
> short history ($K{=}1$ or $5$), $3/25$ prefer $K{=}10$, and $14/25$ benefit from longer histories ($K{=}20$ or $30$). At
> the user level, no single value of $K$ dominates: different users prefer short, intermediate, or long histories,
> and $K{=}10$ is competitive (it is the best setting for $22.0\%$ of users) without being extreme in either direction.
>
> Taken together, these results suggest that: (i) using roughly ten past workouts is already sufficient to capture most of
> the useful historical structure when averaged over the whole dataset, (ii) there is genuine variability across sports
> and users, with a minority favouring very short or very long histories, and (iii) choosing $K{=}10$ strikes a reasonable
> balance between accuracy and computational cost.
>
> Regarding users with fewer than 10 past sessions, the time-aware attention module naturally supports variable-length
> histories. Each available workout is first encoded into a workout summary, and the sequence of summaries is then fed to
> the GRU-based aggregator. Users with shorter histories are therefore represented by shorter sequences, without any ad
> hoc padding or special-case handling.
>
> **Table 9. Effect of history length $K$ (number of past workouts) on overall
> prediction error on the FitRec**
>
> | $K$ (past workouts) | MSE    | MAE   | $Δ$ MSE vs. $K{=}10$ | $Δ$ MAE vs. $K{=}10$ |
> |---------------------|--------|-------|----------------------|-----------------------|
> | 1                   | 206.47 | 10.52 | +6.82                | +0.26                 |
> | 5                   | 206.40 | 10.32 | +6.75                | +0.06                 |
> | 10                  | 199.65 | 10.26 | 0.00                 | 0.00                  |
> | 20                  | 199.50 | 10.24 | -0.15                | -0.02                 |
> | 30                  | 203.24 | 10.17 | +3.59                | -0.09                 |
>
> **Table 10. Number of sports and users for which a given history length $K$
> achieves the lowest per-entity MSE on {FitRec**
>
> | $K$ | \#sports best | \% sports | \#users best | \% users |
> |-----|---------------|-----------|--------------|----------|
> | 1   | 3             | 12.0\%    | 7            | 7.69\%   |
> | 5   | 5             | 20.0\%    | 23           | 25.3\%   |
> | 10  | 3             | 12.0\%    | 20           | 22.0\%   |
> | 20  | 9             | 36.0\%    | 17           | 18.7\%   |
> | 30  | 5             | 20.0\%    | 24           | 26.4\%   |

---

> ### Author Response · Authors · 2025-11-28
>
> Dear Reviewer gRNk,
>
> Thank you again for your thoughtful and constructive comments. We have carried out additional experiments and analyses to address your points in detail, and we have summarized these results in our responses above.
>
> As the discussion period is nearing its end, we would like to kindly ask whether you have any **remaining concerns or suggestions** that you would like us to address. If everything now looks satisfactory, we would also be grateful if you could briefly indicate this in the discussion.
>
> We sincerely appreciate your time and effort in reviewing our paper and your valuable feedback in helping us improve this work.

---

### Author Response · Authors · 2025-11-20
**General Response**

We extend our sincere appreciation to reviewers
**gRNk, xP3V, and xynF** for dedicating
their time and expertise during the rebuttal phase. Their constructive
comments are invaluable to the enhancement of our manuscript:

**Problem formulation and motivation.** Reviewers
**gRNk** and
**xP3V** appreciated that we formulate
heart-rate prediction under source and user heterogeneity as a
practically meaningful and well-motivated problem for real-world
wearable deployment.

**Methodology.** Reviewers **gRNk**,
**xP3V**, and
**xynF** noted that our combination of
random feature dropout, time-aware attention, and contrastive learning
provides a reasonable and well-grounded approach to handling
heterogeneous wearable data and learning informative user
representations.

**Dataset and reproducibility.** Reviewers
**gRNk** and
**xynF** highlighted the value of the
ParroTao dataset as a realistic multi-device, multi-activity benchmark
for studying heterogeneous wearable data.

**Experimental evaluation and baselines.** Reviewer
**xynF** praised the breadth of baselines,
the clear experimental setup, and the consistently strong empirical
performance of our model.

**Downstream applications and translational potential.** Reviewer
**xP3V** emphasized that the downstream
applications convincingly demonstrate how our framework can be used for
interactive, personalized heart-rate planning in practice.

---
To address the reviewers’ concerns, we added several new analyses and clarified existing results, while our main conclusions remain **unchanged**. We systematically evaluated **all** key design choices through ablation and sensitivity studies on two datasets, demonstrating the effectiveness and robustness of our proposed framework. During the rebuttal phase, reviewer **xP3V** provided highly positive feedback on these additions. The key results added during the rebuttal phase are summarized below.

1.  **Ethics and data governance.** We clarified that all participants
    provided written informed consent, that collected signals are
    anonymized and accessible only to authorized researchers for
    non-commercial use, addressing potential ethical concerns about our
    wearable data collection and handling.

2.  **Dataset splits and cohort characteristics.** We clarified the
    participant-level train/validation/test splitting strategy and
    reported key demographics of the ParroTao cohort to better
    contextualize the scope and generalizability of our results.

3.  **Feature set inspection and imputation strategy.** We audited all
    channels in ParroTao, removed constant inputs, updated the variable
    table to clarify each channel’s role, and showed that
    zero-imputation is the most effective strategy on our dataset.

4.  **Architecture and historical modeling.** We systematically compared
    alternative encoders, ablated the historical stream, varied the
    history length, and inspected attention weights, confirming the
    effectiveness of our proposed architecture and indicating that our
    time-aware attention module with $K \approx 10$ past workouts offers
    a good trade-off between performance and computational cost.

5.  **Feature importance and heterogeneity across sports and users.**
    Through feature ablations on FitRec and ParroTao, we quantified
    which channels are globally influential and showed that effective
    feature importance varies substantially across sports and
    individuals.

6.  **Representation quality and subject-specific information.** Using
    structure-aware embedding metrics and downstream gender/physiology
    prediction tasks, we showed that our learned representations capture
    informative, stable sport-specific and user-specific
    characteristics.

7.  **More downstream utility cases.** We extended our evaluation to
    cross-device feature imputation and heart-rate imputation, where our
    framework consistently outperforms classical interpolation/filtering
    methods and the strongest baseline model, highlighting its
    versatility for practical deployment scenarios.

8.  **Stronger baselines, statistical tests, and clarified claims.** We
    added recent transformer-based/self-supervised baselines and compared them in terms of both predictive performance and computational complexity, conducted Wilcoxon signed-rank tests for key ablations, and clarified our
    post-acceptance plan for releasing the ParroTao dataset.

All textual updates in the revised manuscript are clearly
marked in **blue**. We believe these
revisions substantially strengthen the paper and align it more closely
with the standards of the conference, and we hope the revised version
will be considered a valuable contribution to the field.

---

### Note · Authors · 2026-01-27

I have read and agree with the venue's withdrawal policy on behalf of myself and my co-authors.

---

### Meta-Review · Area_Chair_Wi5x · 2026-01-05

**Summary:**

This paper proposes a framework for heart rate prediction from heterogeneous wearable data that addresses source heterogeneity (different devices) and user heterogeneity (individual physiological differences). While the authors demonstrate improvements over baselines and provide extensive rebuttals addressing reviewer concerns, several fundamental weaknesses remain unaddressed, particularly regarding the limited novelty of the methodology and incomplete evaluation of the learned representations' utility.

**Reviewer Concerns:**

**Addressed concerns:**
- The authors adequately addressed data split methodology by clarifying participant-level stratification (80/10/10 split with no user appearing in multiple splits) [Reviewers gRNk, xynF]
- Non-causal feature concerns were addressed by auditing all channels and removing potentially non-causal variables like VO2max from inputs [Reviewer xynF]
- Feature importance analysis was added through ablation studies showing power, cadence, and speed as most influential features [Reviewers xP3V, xynF]
- History length sensitivity analysis was conducted, showing K≈10 past workouts offers optimal trade-off [Reviewer gRNk]
- Additional transformer baselines (PatchTST, Rotary TS Transformer) were compared, with the proposed method outperforming both [Reviewer xP3V]

**Outstanding concerns:**
- The methodology relies on relatively standard techniques (random dropout, contrastive learning, attention) without demonstrating clear novelty over existing approaches for handling heterogeneous time-series data [Reviewers xP3V, xynF]
- The downstream applications remain limited: route recommendation essentially reduces to terrain-based prediction, and the evaluation metrics mirror the primary HR forecasting task [Reviewer xynF]
- The ParroTao cohort consists primarily of young, physically active athletes (ages 18-30), limiting generalizability claims to broader populations including elderly or clinical populations [Reviewer gRNk]
- Statistical significance of ablation improvements is marginal in some cases, with the no-dropout ablation on ParroTao showing only 1.87% improvement [Reviewer xP3V]

**Reviewer Scores:**

Reviewer gRNk: 6. Justification: The reviewer's concerns about architectural justification and demographic limitations were partially addressed through encoder ablations and cohort characterization, but generalizability to broader populations remains undemonstrated.

Reviewer xP3V: 6. Justification: While the reviewer acknowledged improvements after rebuttal (updating from 4), the core concern about limited methodological novelty, combining standard techniques without comparison to recent modality-agnostic architectures, was only partially addressed through complexity analysis rather than empirical comparison.

Reviewer xynF: 4. Justification: Despite extensive additional experiments, the fundamental weakness regarding insufficient evaluation of representation utility persists: gender prediction alone does not convincingly demonstrate physiologically meaningful information capture, and downstream applications lack distinct evaluation frameworks.

---

### Decision · Program_Chairs · 2026-01-26

Reject